# Ocean carbon cycle feedbacks in CMIP6 models: contributions from different basins

Anna Katavouta[1,2] and Richard G. Williams[1]

[1]Department of Earth, Ocean and Ecological Sciences, School of Environmental Sciences, University of Liverpool, Liverpool, UK
[2]National Oceanography Centre, Liverpool, UK

**Correspondence:** Anna Katavouta (a.katavouta@liverpool.ac.uk)

**Abstract.** The ocean response to carbon emissions involves the combined effect of an increase in atmospheric $CO_2$, acting to enhance the ocean carbon storage, and climate change, acting to decrease the ocean carbon storage. This ocean response can be characterised in terms of a carbon-concentration feedback and a carbon-climate feedback. The contribution from different ocean basins to these feedbacks is explored using diagnostics of ocean carbonate chemistry, physical ventilation and biological processes in 11 CMIP6 Earth system models. To gain mechanistic insight, the dependence of these feedbacks to the Atlantic Meridional Overturning Circulation (AMOC) is also investigated in an idealised climate model and the CMIP6 models. For the carbon-concentration feedback, the Atlantic, Pacific and Southern Oceans provide comparable contributions when estimated in terms of the volume integrated carbon storage. This large contribution from the Atlantic Ocean relative to its size is due to a strong local physical ventilation and an influx of carbon transported from the Southern Ocean. The Southern Ocean has a large anthropogenic carbon uptake from the atmosphere, but its contribution to the carbon storage is relatively small due to a large carbon transport to the other basins. For the carbon-climate feedback estimated in terms of carbon storage, the Atlantic and Arctic Oceans provide the largest contributions relative to their size. In the Atlantic, this large contribution is primarily due to climate change acting to reduce the physical ventilation. In the Arctic, this large contribution is associated with a large warming per unit volume. The Southern Ocean provides a relatively small contribution to the carbon-climate feedback, due to a competition between the climate effects of a decrease in solubility and physical ventilation, and an increase in accumulation of regenerated carbon. The more poorly-ventilated Indo-Pacific Ocean provides a small contribution to the carbon cycle feedbacks relative to its size. In the Atlantic Ocean, the carbon cycle feedbacks strongly depend on the AMOC strength and its weakening with warming. In the Arctic, there is a moderate correlation between the AMOC weakening and the carbon-climate feedback that is related to changes in carbonate chemistry. In the Pacific, Indian and Southern Oceans, there is no clear correlation between AMOC and the carbon cycle feedbacks, suggesting that other processes control the ocean ventilation and carbon storage there.

## 1 Introduction

Carbon emissions drive an Earth system response via direct changes in the biogeochemical carbon cycle and the physical climate. These changes in the biogeochemical carbon cycle and physical climate further amplify or dampen the Earth system

response, with this amplification often referred to as a feedback (Sherwood et al., 2015). The physical climate feedback involves the combined effect from changes in atmospheric water vapour, tropospheric lapse rate, surface albedo and clouds (Ceppi and Gregory, 2017), and from a shift in the regional patterns of ocean heat uptake due to changes in the ocean circulation (Winton et al., 2013). For the carbon cycle, an initial increase in atmospheric $CO_2$ leads to a carbon uptake and storage in land and ocean reservoirs. This response of the carbon cycle to the increase in atmospheric $CO_2$ is characterised by the carbon-concentration feedback. At the same time, the carbon cycle is further modified by changes in the physical climate, such as warming and an increase in ocean stratification leading to an amplification of the initial increase in atmospheric $CO_2$. This response of the carbon cycle to changes in the physical climate is characterised by the carbon-climate feedback. These two carbon cycle feedbacks have been extensively used to understand and quantify the response of the global carbon cycle to carbon emissions (Friedlingstein et al., 2003, 2006; Gregory et al., 2009; Boer and Arora, 2009; Arora et al., 2013; Schwinger et al., 2014; Schwinger and Tjiputra, 2018; Williams et al., 2019; Arora et al., 2020). A regional extension of the carbon cycle feedbacks have been also used to explore their geographical distribution and the mechanisms that control the land and ocean carbon uptake and storage in difference regions (Yoshikawa et al., 2008; Boer and Arora, 2010; Tjiputra et al., 2010; Roy et al., 2011).

On a global scale, the carbon-concentration feedback is of comparable strength over the land and ocean, while the carbon-climate feedback is about three times stronger over the land than the ocean on centennial timescales in the CMIP6 Earth system models (Arora et al., 2020). However, there is a substantial geographical variation in the ocean carbon-climate feedback (Tjiputra et al., 2010; Roy et al., 2011) as a result of an interplay between the effect of carbonate chemistry, physical ventilation and biological processes. In the tropics, the carbonate chemistry and the decrease in solubility with warming drives a reduction in the ocean carbon uptake with climate change (Roy et al., 2011; Rodgers et al., 2020). In the North Atlantic, the physical ventilation and its weakening with warming acts to further reduce the ocean carbon uptake with climate change (Yoshikawa et al., 2008; Tjiputra et al., 2010; Roy et al., 2011). In the Southern Ocean, changes in the cycling of biological material with climate change can partly counteract the reduction in the ocean carbon uptake due to the decrease in solubility and physical ventilation with warming (Sarmiento et al., 1998; Bernardello et al., 2014).

The ocean carbon cycle feedbacks can be defined in terms of either the cumulative ocean carbon uptake or the ocean carbon storage (Schwinger et al., 2014; Arora et al., 2020). For the global ocean, these two definition are almost equivalent apart from a small contribution from the land-to-ocean carbon flux from river runoff and the carbon burial in ocean sediments (Arora et al., 2020). However, on a regional scale these two definitions are different, as the ocean carbon storage explicitly includes the convergence of transport of carbon by the ocean circulation. This transport effect leads to different geographical patterns for the ocean carbon storage and the ocean cumulative carbon uptake (Frölicher et al., 2015). This transport effect also leads to a broadly similar geographical distribution for ocean carbon and heat storage, with the "redistribution" of the pre-industrial carbon and heat by changes in the circulation with warming driving a second order asymmetry between the regional patterns of heat and carbon storage (Winton et al., 2013; Bronselaer and Zanna, 2020; Williams et al., 2021). The combined air-sea transfer and transport effect leads to the Atlantic, Pacific and Southern Oceans each storing about 25-30% of the additional heat and carbon in CMIP6 models for a quadrupling of atmospheric $CO_2$ (Fig. 1a and b), despite their different size (see supplement Fig. C1 for the basins definition). The Atlantic and Arctic Oceans have the largest increase in carbon and heat per unit volume, as

given by the dissolved inorganic carbon and temperature (Fig. 1c and d). The Pacific Ocean has the smallest increase in carbon and heat per unit volume (Fig. 1c and d). Our motivation is to explore the mechanisms that lead to these regional variations in carbon storage and carbon cycle feedbacks in the different ocean basins in CMIP6 models.

A mechanism that can affect the regional carbon storage is the Atlantic Meridional Overturning Circulation (AMOC). The projected weakening in AMOC with climate change (Cheng et al., 2013) weakens the ocean physical ventilation and transport of carbon into the ocean interior, which acts to reduce the ocean carbon uptake and storage (Sarmiento and Le Quéré, 1996; Crueger et al., 2008). The weakening in AMOC with climate change also increases the residence time in the ocean interior and the accumulation of remineralised carbon at depth, which acts to increase the ocean carbon uptake and storage (Sarmiento and Le Quéré, 1996; Joos et al., 1999; Schwinger et al., 2014; Bernardello et al., 2014). Previous studies suggest that the combined effect of these two competing processes leads to a modest reduction in ocean carbon uptake and storage with AMOC weakening, and an ocean carbon-climate feedback that amplifies the increase in atmospheric $CO_2$ (Sarmiento and Le Quéré, 1996; Joos et al., 1999; Crueger et al., 2008; Schwinger et al., 2014). However, the net effect of AMOC weakening with climate change on the carbon storage is highly uncertain and sensitive to the representation of the vertical carbon gradient and ocean biological processes in Earth system models. This uncertainty motivate us to explore the control of AMOC on the carbon cycle feedbacks in CMIP6 models, and the relative importance of changes in biological processes and physical ventilation for the carbon storage in different ocean basins.

Our aim is to provide insight into the relative contribution of different ocean basins to the ocean carbon cycle feedbacks and the processes that drive this regional partitioning in the CMIP6 models. In section 2, we provide the framework for the carbon cycle feedbacks and explore their geographical distribution in 11 CMIP6 Earth system models (Table 1). In section 3, the ocean carbon cycle feedbacks are separated into contribution from carbonate chemistry, physical ventilation and biological processes, and the controls of the global and regional feedbacks are investigated in diagnostics of CMIP6 models. In section 4, the effect of the AMOC on the global and basin-scale carbon cycle feedbacks is investigated, firstly using an idealised climate model that provides a mechanistic insight, and then in diagnostics of CMIP6 models. Section 5 summarises our conclusions and discusses the wider context of our analysis .

## 2   Ocean carbon cycle feedbacks in CMIP6 models

### 2.1   Global ocean

In the carbon cycle feedback framework introduced by Friedlingstein et al. (2003, 2006) the ocean carbon gain due to anthropogenic carbon emissions, $\Delta I_{ocean}$, is expressed as a function, $F$, of changes in the atmospheric $CO_2$ and the physical climate,

$$\Delta I_{ocean} = F(CO_{2,0} + \Delta CO_2, T_0 + \Delta T) - F(CO_{2,0}, T_0), \tag{1}$$

where the surface air temperature, $T$, is used as a proxy for the physical climate and subscript $_0$ denotes the pre-industrial state. By expanding the function $F$ into a Taylor series (Schwinger et al., 2014; Williams et al., 2019), the ocean carbon gain relative to the pre-industrial, $\Delta I_{ocean}$, is expressed as

$$\Delta I_{ocean} = \left.\frac{\partial F}{\partial CO_2}\right|_0 \Delta CO_2 + \left.\frac{\partial F}{\partial T}\right|_0 \Delta T + \left.\frac{\partial^2 F}{\partial CO_2 \partial T}\right|_0 \Delta CO_2 \Delta T \tag{2}$$

$$+ \left.\frac{\partial^2 F}{\partial CO_2^2}\right|_0 \Delta CO_2^2 + \left.\frac{\partial^2 F}{\partial T^2}\right|_0 \Delta T^2 + R^3, \tag{3}$$

where $R^3$ contains the third order and higher derivatives. By ignoring the second and higher order terms, but keeping the terms for the non-linear relationship between atmospheric $CO_2$ and climate change, Eq. (3) is rewritten as

$$\Delta I_{ocean} = \beta \Delta CO_2 + \gamma \Delta T + \mathcal{N} \Delta CO_2 \Delta T, \tag{4}$$

where the ocean carbon-concentration feedback parameter is defined as $\beta = \left.\frac{\partial F}{\partial CO_2}\right|_0$, the ocean carbon-climate feedback parameter is defined as $\gamma = \left.\frac{\partial F}{\partial T}\right|_0$ and the non-linearity of the ocean carbon cycle feedbacks is defined as $\mathcal{N} = \left.\frac{\partial^2 F}{\partial CO_2 \partial T}\right|_0$.

The carbon cycle feedback parameters, $\beta$ and $\gamma$, are traditionally estimated using Earth system model simulations with the couplings between the carbon cycle and radiative forcing either switched on or off: a fully coupled simulation, a radiatively coupled simulation and a biogeochemically coupled simulation (Friedlingstein et al., 2006; Arora et al., 2013; Jones et al., 2016; Arora et al., 2020). Any combination of these three simulations can be used to estimate the carbon cycle feedback parameters, however, each combination yields somewhat different results due to the non-linearity of the system (Gregory et al., 2009; Zickfeld et al., 2011; Schwinger et al., 2014; Arora et al., 2020). Here, we estimate the carbon cycle feedback parameters using the fully coupled simulation (COU) and the biogeochemically coupled simulation (BGC) under the 1% yr$^{-1}$ increasing $CO_2$ experiment, in which the atmospheric $CO_2$ concentration increases from its pre-industrial value of around 285 ppm until it quadruples over a 140-year period, following the recommended C$^4$MIP protocol of experiments (Jones et al., 2016; Arora et al., 2020). To remove the effect of model drift and reduce model biases, the pre-industrial control simulation (piControl) was used to estimate the pre-industrial state in the Earth system models. For simplicity, we ignore the effect of the air temperature increase in the BGC simulation to the feedbacks, which has a contribution of less than 5% (Arora et al., 2020), such that

$$\begin{aligned} \beta &= \frac{\Delta I_{ocean}^{BGC}}{\Delta CO_2}, \\ \gamma &= \frac{\Delta I_{ocean}^{COU} - \Delta I_{ocean}^{BGC}}{\Delta T}, \end{aligned} \tag{5}$$

where $\Delta CO_2$ is the increase in atmospheric $CO_2$ and $\Delta T$ the increase in air surface temperature in the fully coupled Earth system (i.e. COU simulation) relative to the pre-industrial state. The carbon-climate feedback parameter, $\gamma$, in Eq. (5) corresponds to the effect of climate change under rising atmospheric $CO_2$ and hence includes the effect of the non-linearity, $\mathcal{N} \Delta CO_2$, in Eq. (4) (Schwinger et al., 2014).

The ocean carbon-concentration feedback parameter, $\beta$, is positive in all the CMIP6 models (Table 2). The ocean carbon-climate feedback parameter, $\gamma$, is negative in all the Earth system models, indicating that the ocean takes up less carbon in response to climate change (Table 2). The variability in $\beta$ amongst the Earth system models, as described by the coefficient of variation, CV, is relatively small on the global scale (CV=0.09) when compared with the variability in $\gamma$ (CV=0.43) (Table 2). However, for the uncertainty in the ocean carbon gain due to carbon emissions, the carbon-concentration feedback contributes to a spread of 62 PgC, while the carbon-climate feedback contributes only to a spread of 25 PgC amongst the CMIP6 Earth system models on a global scale and for a quadrupling of atmospheric $CO_2$; where the spread corresponds to one standard deviation.

## 2.2 Regional ocean

The carbon cycle feedbacks for the global ocean in Eq. (4) can be further separated into contributions from different ocean regions such that

$$\Delta I_{ocean} = \sum_{n=1}^{global} \beta_n \Delta CO_2 + \sum_{n=1}^{global} \gamma_n \Delta T + \sum_{n=1}^{global} \mathcal{N}_n \Delta CO_2 \Delta T, \tag{6}$$

where $n$ denotes the different ocean regions, $\Delta CO_2$ and $\Delta T$ are the global changes in atmospheric $CO_2$ and the surface air temperature, respectively, and $\beta_n$, $\gamma_n$ and $\mathcal{N}_n$ are the carbon cycle feedback parameters and their non-linearity for each ocean region $n$.

Traditionally, the carbon cycle feedbacks are defined based on the cumulative carbon uptake from the atmosphere, $\Delta\mathcal{S}$, (Tjiputra et al., 2010; Roy et al., 2011) such that the carbon cycle feedback parameters for each ocean region $n$ are expressed as

$$\begin{aligned} \beta_n &= \frac{\Delta\mathcal{S}_n^{BGC}}{\Delta CO_2}, \\ \gamma_n &= \frac{\Delta\mathcal{S}_n^{COU} - \Delta S_n^{BGC}}{\Delta T}. \end{aligned} \tag{7}$$

where the carbon-climate feedback parameter, $\gamma_n$, corresponds to the effect of climate change under rising atmospheric $CO_2$ and hence includes the effect of the non-linearity, $\mathcal{N}_n \Delta CO_2$, in Eq. (6) .

Alternatively, the carbon cycle feedbacks can be defined based on the carbon storage, such that the regional carbon cycle feedback parameters for each ocean region $n$ are expressed as

$$\begin{aligned} \beta_n^* &= \frac{\Delta I_n^{BGC}}{\Delta CO_2}, \\ \gamma_n^* &= \frac{\Delta I_n^{COU} - \Delta I_n^{BGC}}{\Delta T}, \end{aligned} \tag{8}$$

where $\Delta I_n$ is the change in the carbon inventory in region $n$. The carbon storage includes the combined effect from the local air-sea carbon exchange and the transport of carbon by the ocean circulation, such that

$$\Delta I_n = \Delta\mathcal{S}_n + \Delta\mathcal{G}_n, \tag{9}$$

where $\Delta \mathcal{S}_n$ is the regional cumulative ocean carbon uptake from the atmosphere relative to the pre-industrial, and $\Delta \mathcal{G}_n$ is the regional cumulative carbon gain relative to the pre-industrial due to the ocean transport.

By substituting Eqns (9) and (7) in Eq. (8) the two definitions for the carbon cycle feedback parameters are related by

$$
\begin{aligned}
\beta_n^* &= \beta_n + \frac{\Delta \mathcal{G}_n^{BGC}}{\Delta CO_2}, \\
\gamma_n^* &= \gamma_n + \left( \frac{\Delta \mathcal{G}_n^{COU} - \Delta \mathcal{G}_n^{BGC}}{\Delta T} \right).
\end{aligned} \tag{10}
$$

Expression (10) shows that the feedback parameters defined by the regional carbon storage, $\beta^*$ and $\gamma^*$, are proportional to the feedback parameters defined by the regional cumulative carbon uptake, $\gamma$ and $\beta$, but further modified by the ocean carbon transport.

### 2.2.1 Estimates based on carbon storage versus carbon uptake

On a global scale, the transport effect on the carbon storage integrates to zero, $\sum_n^{global} \Delta \mathcal{G}_n = 0$, such that the feedback parameters estimated from the carbon storage, $\beta^*$ and $\gamma^*$, are equivalent to the feedback parameters estimated from the cumulative carbon uptake, $\beta$ and $\gamma$, when ignoring the small carbon exchange between land and ocean. However, on regional scales the effect of the ocean transport on carbon storage leads to different spatial patterns in these carbon cycle feedback parameters, $\beta$ and $\beta^*$, and $\gamma$ and $\gamma^*$ (Fig. 2).

The carbon-concentration feedback parameter estimated from the cumulative carbon uptake, $\beta$, is largest and has more inter-model variability in (i) the Southern Ocean, (ii) the eastern boundary upwelling regions, (iii) the Gulf Stream and its extension into North Atlantic Current, and (iv) the Kuroshio Extension (Fig 2b). The inter-model variability in $\beta$ is also significant along the equatorial Pacific, with this variability related to the inter-model spread in the trade winds and equatorial upwelling. In contrast, in the subtropical gyres, the ocean anthropogenic carbon uptake is limited and $\beta$ is small (Fig. 2b).

The carbon-concentration feedback parameter estimated from ocean carbon storage, $\beta^*$, is again large in the North Atlantic, but instead large in the Southern hemisphere subtropical gyres and small in the Southern Ocean south of $50^o$S relative to $\beta$ (Fig. 2a). This difference between $\beta$ and $\beta^*$ (supplement Fig. C2) is due to the northward transport of anthropogenic carbon from the Southern Ocean associated with subduction and transport of mode and intermediate waters. The variability in $\beta^*$ amongst the models is large in the North Atlantic and extends south along the Atlantic western boundary (Fig. 2a).

The carbon-climate feedback parameter estimated from the cumulative carbon uptake, $\gamma$, is large and negative in the North Atlantic and in the Southern Ocean from $50^o$S to $65^o$S (Fig 2b). In contrast, $\gamma$ is large and positive in a narrow band between $40^o$S and $45^o$S, in the Southern hemisphere eastern boundary upwelling regions and in polar regions with sea ice. The regions of large ocean carbon loss or uptake from the atmosphere due to climate change, as shown by the large $\gamma$, also experience the largest variability in $\gamma$ amongst the CMIP6 Earth system models (Fig 2b).

The effect of carbon transport on $\gamma^*$ is of opposite sign to the effect of the cumulative carbon uptake in most regions (supplement Fig C2). This transport effect leads to a $\gamma^*$ that is overall less negative in the Southern Ocean and in the high latitudes of the North Atlantic, but more negative in the Arctic, the equatorial Pacific and along the Atlantic western boundary

relative to $\gamma$ (Fig. 2.a). The spread in $\gamma^*$ amongst the models is largest in the North Atlantic, in the Arctic, along the Atlantic western boundary and in the Southern Ocean (Fig. 2.a).

The carbon cycle feedbacks estimated from the cumulative carbon uptake better describe the atmosphere-ocean interaction. The carbon cycle feedbacks estimated from the ocean carbon storage instead better describe the response of the ocean carbon budget to carbon emissions. Here, we focus on the carbon cycle feedbacks estimated from the regional ocean carbon storage to enable diagnostics in terms of the preformed and regenerated carbon pools and gain more mechanistic insight.

### 2.2.2 Basin-scale $\beta^*$ and $\gamma^*$

We define the Southern Ocean as south of $35^o$S and the Arctic Ocean as north of $65^o$N, and exclude semi-enclosed seas from our basins definition (see supplement Fig. C1 for a map of the regions). The Pacific, Southern and Atlantic Oceans contribute equally to the ocean carbon-concentration feedback parameter, as estimated in terms of carbon storage, with an inter-model mean $\beta^*$ of 0.24, 0.22 and 0.22 PgC ppm$^{-1}$, respectively (Table 2). The Indian Ocean contributes less than half than the other three basins to $\beta^*$, with an inter-model mean of 0.10 PgC ppm$^{-1}$ (Table 2). The Arctic Ocean has a $\beta^*$ of only 0.03 PgC ppm$^{-1}$. The Pacific, Southern, Atlantic, Indian and Arctic Oceans have an inter-model mean carbon-climate feedback parameter, defined in terms of carbon storage, $\gamma^*$, of -4.8, -3.2, -4.8, -1.2 and -1.2 PgC K$^{-1}$, respectively (Table 2). The basin-scale variability in $\beta^*$ amongst the models, as described by the coefficient of variation, CV, is less than 0.15 (Table 2). The basin-scale variability in $\gamma^*$ amongst the models varies from CV=0.56 in the Pacific Ocean to CV=0.85 in the Southern Ocean (Table 2). The inter-model variability in $\beta^*$ and $\gamma^*$ for each basin is larger than that of the global ocean (Table 2), which suggests that variability in different basins compensate for each other. For diagnostics of the separate contribution of the ocean carbon uptake and transport on the basin-scale carbon storage, and feedback parameters see appendix A.

## 3 Processes controlling the carbon cycle feedbacks in CMIP6 models

To gain insight into the driving mechanisms of the carbon cycle feedbacks and their uncertainty amongst Earth system models, $\beta^*$ and $\gamma^*$ may be separated into contribution from the regenerated, the saturated and the disequilibrium ocean carbon pools following the methodology of Williams et al. (2019) and Arora et al. (2020). The ocean dissolved inorganic carbon, $DIC$, may be defined in terms of these separate carbon pools (Ito and Follows, 2005; Williams and Follows, 2011; Lauderdale et al., 2013; Bernardello et al., 2014),

$$DIC = DIC_{pref} + DIC_{reg} = DIC_{sat} + DIC_{dis} + DIC_{reg}, \tag{11}$$

where $DIC_{pref}$ is the part of the $DIC$ transferred from the surface into the ocean interior due to the physical ventilation, involving the circulation; and $DIC_{reg}$ is the part of the $DIC$ accumulated into the ocean interior due to biological regeneration of organic carbon. Similarly, the $DIC_{pre}$ can be viewed as the part of the $DIC$ associated with the solubility pump, and the $DIC_{reg}$ as the part of the $DIC$ associated with the biological pump. The $DIC_{pref}$ can be further split into two idealised carbon pools: (i) the $DIC_{sat}$ representing the amount of $DIC$ that the ocean would have if the whole ocean reached a full

chemical equilibrium with the contemporaneous atmospheric $CO_2$ concentration; and $DIC_{dis}$ representing the extent that the ocean departs from a full chemical equilibrium with the contemporaneous atmospheric $CO_2$. Assuming the changes in the biological organic carbon inventory are small, the changes in the ocean carbon inventory relative to the pre-industrial era, $\Delta I_{ocean}$ in PgC, are related to the volume integral of the changes in each of the $DIC$ pools, $\Delta DIC$ in mol m$^{-3}$, as

$$\Delta I_{ocean} = m \int_V (\Delta DIC_{sat} + \Delta DIC_{dis} + \Delta DIC_{reg}) \, dV = \Delta I_{sat} + \Delta I_{dis} + \Delta I_{reg}, \tag{12}$$

where $m = 12.01 \cdot 10^{-15}$ PgC mol$^{-1}$ is a unit conversion from moles to Pg of carbon.

By substituting Eq. (12) into Eq. (5), for the global ocean, or Eq. (8), for the regional ocean, the carbon cycle feedback parameters may be diagnosed in terms of these different ocean carbon pools (Williams et al., 2019; Arora et al., 2020),

$$
\begin{aligned}
\beta^* &= \beta_{sat} + \beta_{dis} + \beta_{reg} = \frac{\Delta I_{sat}^{BGC}}{\Delta CO_2} + \frac{\Delta I_{dis}^{BGC}}{\Delta CO_2} + \frac{\Delta I_{reg}^{BGC}}{\Delta CO_2}, \\
\gamma^* &= \gamma_{sat} + \gamma_{dis} + \gamma_{reg} = \frac{\Delta I_{sat}^{COU} - \Delta I_{sat}^{BGC}}{\Delta T} + \frac{\Delta I_{dis}^{COU} - \Delta I_{dis}^{BGC}}{\Delta T} + \frac{\Delta I_{reg}^{COU} - \Delta I_{reg}^{BGC}}{\Delta T}.
\end{aligned} \tag{13}
$$

### 3.1 Contribution from the saturated carbon pool to $\beta^*$ and $\gamma^*$

The saturated part of $\beta^*$ and $\gamma^*$ in Eq (13) is expressed as

$$
\begin{aligned}
\beta_{sat} &= \frac{\Delta I_{sat}^{BGC}}{\Delta CO_2}, \\
\gamma_{sat} &= \frac{\Delta I_{sat}^{COU} - \Delta I_{sat}^{BGC}}{\Delta T},
\end{aligned} \tag{14}
$$

The changes in the saturated carbon pool relative to the pre-industrial in Eq. (14) are diagnosed as

$$\Delta I_{sat} = m \int_V \Delta DIC_{sat} dV = m \int_V \Delta f (CO_2, T_{ocean}, S, P, Si, Alk_{pre}) \, dV, \tag{15}$$

where $\Delta$ is the change relative to the pre-industrial, $T_{ocean}$ is the ocean temperature , $S$ is the ocean salinity, $P$ is the ocean phosphate concentration, $Si$ is the ocean silicate concentration, $Alk_{pre}$ is the preformed alkalinity, and $f$ is a non-linear function representing the solution to the ocean carbonate chemistry which provides $DIC_{sat}$ for the contemporaneous atmospheric $CO_2$. Here, $f$ is estimated following the iterative solution for the ocean carbonate chemistry of Follows et al. (2006) and by
230 considering the small contribution of minor species (borate, phosphate, silicate) to the preformed alkalinity. In the limit that the ocean hydrogen ion concentration at a chemical equilibrium with the given atmospheric $CO_2$, $[H^+]_{sat}$, is known, or the preformed alkalinity is assumed equal to the carbonate alkalinity, $f$ function corresponds to the usual solution of the carbonate system that provides $DIC_{sat}$ based on two knowns: atmospheric $CO_2$ and either $[H^+]_{sat}$ (see Eq. (18)) or carbonate alkalinity. The preformed alkalinity is estimated from a multiple linear regression using salinity and the conservative tracer $PO$ (Gruber
et al., 1996), with the coefficients of this regression estimated based on the surface (first 10 m) alkalinity, salinity, oxygen, and phosphate in each of the Earth system models.

To understand how the ocean carbonate chemistry operates and the mechanisms that control $\beta_{sat}$, consider an ocean buffer factor, $B$, where the fractional change in the atmospheric $CO_2$ and saturated carbon inventory is defined relative to the pre-industrial (Katavouta et al., 2018),

$$B = \frac{\Delta CO_2 / CO_{2,0}}{\Delta I_{sat} / I_{sat,0}}, \tag{16}$$

where subscript $_0$ denotes the pre-industrial era.

Substituting Eq. (16) into Eq. (14) the saturated part of $\beta^*$ can be expressed as

$$\beta_{sat} = \frac{1}{B(CO_2, T_{ocean,0}, S_0, Alk_{pre,0})} \frac{I_{sat,0}}{CO_{2,0}}, \tag{17}$$

where $B(CO_2, T_{ocean,0}, S_0, Alk_0)$ is the ocean buffer factor for the increasing atmospheric $CO_2$, but with no climate change (i.e. for the pre-industrial ocean temperature, salinity and alkalinity), as is the case in the BGC run.

Expression (17) shows that $\beta_{sat}$ is proportional to the ocean capacity to buffer changes in atmospheric $CO_2$ with no changes in the physical climate, $B(CO_2, T_{ocean,0}, S_0, Alk_{pre,0})^{-1}$. The rise in atmospheric $CO_2$ leads to an increase in the saturated ocean carbon inventory, $\Delta I_{sat}$, (Fig. 3a, red shade), but also to a decrease in the ocean capacity to buffer changes in atmospheric $CO_2$ as the ocean acidifies. Accordingly, the buffer factor, $B$, increases and $\beta_{sat}$ decreases with the rise in atmospheric $CO_2$ at a global and basin scale in all the Earth system models (Fig. 3b, red shade). The buffer factor, $B$, and so $\beta_{sat}$ also depend on the pre-industrial ocean state due to the non-linearity of the ocean carbonate system. Hence, there is a spread in $\beta_{sat}$ amongst the Earth system models forced by the same increase in atmospheric $CO_2$ (Fig. 3b, red shade) related to their different pre-industrial ocean temperature, salinity and alkalinity.

To understand the mechanisms that control $\gamma_{sat}$ consider the solution to the saturated part of $DIC$ as dictated by the carbonate chemistry,

$$
\begin{aligned}
DIC_{sat} &= [CO_2]_{sat} + [HCO_3^-]_{sat} + [CO_3^{2-}]_{sat} \\
&= CO_2 \left( K_o + \frac{K_o K_1}{[H^+]_{sat}} + \frac{K_o K_1 K_2}{[H^+]_{sat}^2} \right),
\end{aligned} \tag{18}
$$

where

$$[CO_2]_{sat} = K_o CO_2,$$
$$[HCO_3^-]_{sat} = CO_2 K_o \frac{K_1}{[H^+]_{sat}},$$
$$[CO_3^{2-}]_{sat} = CO_2 K_o \frac{K_1 K_2}{[H^+]_{sat}^2},$$

and $K_o$ is the solubility, $K_1$ and $K_2$ are the ocean carbon dissociation constants, and $[H^+]_{sat}$ is the ocean hydrogen ion concentration at a chemical equilibrium with the contemporaneous atmospheric $CO_2$. $K_o$, $K_1$ and $K_2$ are a function of the ocean temperature and salinity and so depend on the physical climate change, while $[H^+]_{sat}$ depends primary on the changes

in atmospheric $CO_2$. Combining Eq. (18) with Eq. (14) and Eq. (15), and assuming that the ocean temperature and salinity remain at their pre-industrial value in the BGC run with no climate change, $\gamma_{sat}$ can be expressed as

$$\gamma_{sat} = \frac{m}{\Delta T} \int_V CO_2 \left( \Delta K_o + \frac{\Delta \left( K_o K_1 \right)}{[H^+]_{sat}} + \frac{\Delta \left( K_o K_1 K_2 \right)}{[H^+]_{sat}^2} \right) dV. \tag{19}$$

   By expanding $[H^+]_{sat} = [H^+]_{sat,0} + \Delta [H^+]_{sat}$, Eq. (19) can be written as

$$\gamma_{sat} = \frac{m}{\Delta T} CO_2 \int_V \left( \underbrace{\left\{ \Delta K_o + \frac{\Delta \left( K_o K_1 \right)}{[H^+]_{0,sat}} + \frac{\Delta \left( K_o K_1 K_2 \right)}{[H^+]_{0,sat}^2} \right\}}_{\text{effect of ocean warming under } pH_0} - \underbrace{\left\{ \frac{\Delta \left( K_o K_1 \right)}{[H^+]_{0,sat}} \frac{\Delta [H^+]_{sat}}{[H^+]_{sat}} + \frac{\Delta \left( K_o K_1 K_2 \right)}{[H^+]_{0,sat}^2} \frac{\Delta \left( [H^+]_{sat}^2 \right)}{[H^+]_{sat}^2} \right\}}_{\text{effect of ocean warming under } \Delta pH} \right) dV.$$

$$\tag{20}$$

The first term in curly brackets in Eq. (20) contributes to the linear part of $\gamma_{sat}$ and is controlled by changes in climate only; specifically by the effect of changes in the ocean temperature to the solubility and ocean carbon dissociation constants. Under warming due to changes in climate this term is negative. The second term in curly brackets in Eq. (20) contributes to the non-linear part of $\gamma_{sat}$, and it depends on changes in pH, due to the increase in atmospheric $CO_2$, and changes in climate. Under rising atmospheric $CO_2$ and warming, this term is negative as $[H^+]_{sat}$ increases and $K_o$ decreases, and is smaller

than the linear term as $\frac{\Delta [H^+]_{sat}}{[H^+]_{sat}} < 1$. The adjustment of $\gamma_{sat}$ due to changes in the pH, represented by this non-linear term, depends on the increase in atmospheric $CO_2$; for example, the non-linear term is about 30% and 50% of the linear term for a doubling of atmospheric $CO_2$ and for a quadrupling of atmospheric $CO_2$, respectively. Hence, the non-linearity of the carbonate chemistry acts to reduce the magnitude of the negative $\gamma_{sat}$ consistent with the carbonate system being less sensitive to change in temperature under higher ocean $DIC$ (Schwinger et al., 2014).

Expression (20) shows that $\gamma_{sat}$ is proportional to the changes in solubility due to climate change, and is further modified by changes in the ocean carbon dissociation constants with warming and by the non-linearity of the carbonate chemistry. Ocean warming due to climate change leads to a decrease in the saturated carbon inventory, $\Delta I_{sat}$, in all basins (Fig. 4a, red shade) primary driven by a decrease in solubility. This decrease in $\Delta I_{sat}$ with warming drives a nearly constant negative $\gamma_{sat}$ (Fig. 4b, red shade) with the deviations from a constant value being associated to the non-linearity of the carbonate system. The

spread of $\gamma_{sat}$ amongst the Earth system models (Fig. 4b, red shade) is relatively small at a global and basin scale, except in the Arctic, and is associated with different pre-industrial ocean states in the models.

### 3.2   Contribution from the regenerated carbon pool to $\beta^*$ and $\gamma^*$

The regenerated part of $\beta$ and $\gamma$ in Eq (13) is expressed as

$$\beta_{reg} = \frac{\Delta I_{reg}^{BGC}}{\Delta CO_2},$$

$$\gamma_{reg} = \frac{\Delta I_{reg}^{COU} - \Delta I_{reg}^{BGC}}{\Delta T}, \tag{21}$$

Assuming that the oxygen concentration is close to saturation at the surface, $DIC_{reg}$ can be estimated from the apparent oxygen utilization, $AOU$, and the contribution of biological calcification to alkalinity, $Alk$, (Ito and Follows, 2005; Williams and Follows, 2011; Lauderdale et al., 2013), such that $\Delta I_{reg}$ in Eq. (21) is diagnosed as

$$\Delta I_{reg} = m \int_V \Delta DIC_{reg} dV = m \int_V \left( R_{CO} \Delta AOU + \frac{1}{2} \left( \Delta Alk - \Delta Alk_{pre} - R_{NO} \Delta AOU \right) \right) dV, \tag{22}$$

where $R_{CO}$ and $R_{NO}$ are constant stoichiometric ratios, and $Alk_{pre}$ is the preformed alkalinity such that $Alk - Alk_{pre}$ gives the contribution to alkalinity from biological calcification.

The regenerated part of $\beta^*$ is associated with changes in ocean biological processes due to the atmospheric $CO_2$ increase. $\Delta I_{reg}^{BGC}$ and $\beta_{reg}$ are effectively negligible in the Earth system models (Fig. 3a and b, green shade) as these models do not include an explicit dependence of biological production to an increase in carbon availability or decrease in seawater pH. The regenerated part of $\gamma^*$ is associated with changes in ocean biological processes due to changes in climate, including the effect of changes in the circulation on the sinking rate of particles, the effect of warming on the solubility of oxygen and the effect of changes in alkalinity on the dissolution of the calcium carbonate shells of calcifying phytoplankton. $\Delta I_{reg}^{COU} - \Delta I_{reg}^{BGC}$ and $\gamma_{reg}$ are positive and increase in time on a global and basin scale (Fig. 4a and b, green shade), indicating that $\gamma_{reg}$ is dominated by the weakening in the ocean physical ventilation due to climate change. This weakening in the ocean physical ventilation leads to a longer residence time of water masses in the ocean interior, and so to an increase in the accumulation of carbon from the regeneration of biologically cycled carbon in the deep ocean (Schwinger et al., 2014; Bernardello et al., 2014).

### 3.3 Contribution from the disequilibrium carbon pool to $\beta^*$ and $\gamma^*$

The disequilibrium part of $\beta^*$ and $\gamma^*$ are diagnosed using Eq. (13) as

$$\begin{aligned} \beta_{dis} &= \beta^* - \beta_{sat} - \beta_{reg}, \\ \gamma_{dis} &= \gamma^* - \gamma_{sat} - \gamma_{reg}. \end{aligned} \tag{23}$$

The disequilibrium part of the carbon cycle feedback parameters is controlled by the ocean physical ventilation. Specifically, $\beta_{dis}$ is a function of the pre-industrial ocean physical ventilation and the rate of transfer of the anthropogenic carbon from the ocean surface into the ocean interior. The rise in atmospheric $CO_2$ leads to an increase in the magnitude of the negative disequilibrium ocean carbon inventory, $\Delta I_{dis}$, at a global and basin scale (Fig. 3a, blue shade), as the ocean physical ventilation is relatively slow and the ocean carbon transfer over the ocean interior cannot keep up with the rate of increase in atmospheric $CO_2$. Hence, $\beta_{dis}$ is negative at a global and basin scale in all the Earth system models (Fig. 3b, blue shade). However, the rate of increase in the magnitude of $\Delta I_{dis}$ slows down in time (Fig. 3a, blue shade) and $\beta_{dis}$ becomes less negative in time (Fig. 3b, blue shade), as more anthropogenic carbon is transferred into the ocean interior while the buffer capacity of the ocean decreases, which brings the ocean closer to an equilibrium with the contemporaneous atmospheric $CO_2$.

The disequilibrium part of $\gamma^*$ depends on the weakening of the ocean physical ventilation with climate change. Here, $\gamma_{dis}$ is defined based on the climate change impact under rising atmospheric $CO_2$ (i.e., COU-BGC runs) and so includes: (i) the

effect of a weakening ventilation on the pre-industrial ocean carbon gradient involving the natural carbon; (ii) the effect of a weakening ventilation on the anthopogenic carbon; and (iii) the effect from a decreasing sea-ice coverage leading to an increase in the ocean in direct contact with the atmosphere. Overall, the effect of the weakening ventilation on the combined

anthopogenic and natural carbon leads to a negative $\Delta I_{dis}$ and a negative $\gamma_{dis}$ on a global scale and in the Atlantic, Indian, Pacific and Southern Oceans after year 40 (Fig. 4a and b, blue shade). In the Arctic, the effect of the decreasing sea-ice coverage drives a slightly positive $\gamma_{dis}$ over the first 80 years, while the effect of the weakening ventilation dominates and drivers a negative $\gamma_{dis}$ after year 80. The disequilibrium part of $\gamma^*$ becomes more negative in time on global and basin scale as the ocean ventilation weakens with warming.

### 3.4 Combined effect of saturated, disequilibrium and regenerated carbon pools to $\beta^*$ and $\gamma^*$

On a global and basin scale, the ocean carbon-concentration feedback parameter, $\beta^*$, is positive in all the Earth system models. This positive $\beta^*$ is explained by the chemical response involving the rise in ocean saturation, $\beta_{sat}$, opposed by the effect of the relatively slow ocean ventilation, such that the physical uptake of carbon within the ocean is unable to keep pace with the rise in atmospheric $CO_2$, $\beta_{dis}$ (Fig. 3b). There is no significant contribution from biological changes to $\beta^*$ in the Earth system

models. The spread in $\beta_{sat}$ amongst the Earth system models is small on a global and basin scale (Fig. 5a), reflecting the use of similar carbonate chemistry schemes and bulk parameterizations of air–sea $CO_2$ fluxes across marine biogeochemical models in CMIP6 (Séférian et al., 2020). The spread in $\beta_{dis}$ is also small on a global and basin scale (Fig. 5a) as all the Earth system models have a broadly similar general circulation and physical ventilation at the pre-industrial era.

The decrease in solubility and in the physical ventilation with warming reduces the ocean carbon uptake leading to negative

$\gamma_{sat}$ and $\gamma_{dis}$, respectively (Fig. 4b). However, the decrease in the ventilation with warming also acts to increase the residence time in the ocean interior leading to an increase in the regenerated carbon and a positive $\gamma_{reg}$ (Fig. 4b, green shading). The combined $\gamma_{sat}$ and $\gamma_{dis}$ dominate over the opposing $\gamma_{reg}$ leading to an overall negative $\gamma^*$ on a global and basin scale. On a global scale, $\gamma_{sat}$, $\gamma_{dis}$ and $\gamma_{reg}$ are of a similar magnitude (Fig. 5b, black circles). The inter-model spread in the global $\gamma^*$ is mainly driven by the spread in $\gamma_{dis}$ and $\gamma_{reg}$ (Fig. 5b, black circles), and is associated with the different response of the ocean

ventilation to warming in these models. The inter-model spread in the global $\gamma_{reg}$ is larger than the spread in the global $\gamma_{dis}$ due to the different parametrizations of ocean biogeochemical processes in the models.

In the Southern Ocean, the contributions from the saturated, disequilibrium and regenerated carbon pools to $\gamma^*$ are of a similar magnitude (Fig. 5b, blue circles), such that the decrease in solubility, the reduction in the physical ventilation and the increase in the regenerated carbon accumulation in the ocean interior due to climate change are equally important. The inter-

350 model spread in $\gamma^*$ in the Southern Ocean is dominated by the spread in $\gamma_{dis}$ and $\gamma_{reg}$. In the Pacific and Indian Oceans, the magnitude of $\gamma^*$ is primarily controlled by the saturated carbon pool and the decrease in carbon solubility due to warming (Fig. 5b, purple and yellow circles). However, the inter-model spread in $\gamma^*$ in the Pacific and Indian Oceans is dominated by the response of the regenerated carbon pool to climate change, $\gamma_{reg}$ (Fig. 5b, purple and yellow circles). In the Atlantic Ocean, $\gamma^*$ is dominated by the disequilibrium carbon pool (Fig. 5b, red circles) and the reduction in the physical ventilation due to climate

change, with the contribution from the the saturated carbon pool being relatively small. In the Arctic Ocean, $\gamma^*$ is primarily

controlled by the saturated carbon pool and the decrease in solubility, with the contribution from the regenerated carbon pool being negligible (Fig. 5b, green circles).

On regional scales the contributions from the saturated, disequilibrium and regenerated carbon pools to $\beta^*$ and $\gamma^*$ are further modified by local upwelling, changes in alkalinity and the conversion of regenerated to disequilibrium carbon at the ocean surface, as discussed in Appendix B.

### 3.5 Processes controlling the contribution from different basins to $\beta^*$ and $\gamma^*$

The Southern and Indian Oceans contribute 27% and 12%, respectively, to the ocean carbon-concentration feedback parameter, $\beta^*$, following their fractional volumes of the global ocean (Fig. 6a). However, the Atlantic and Arctic Oceans contribute 26% and 3% to $\beta^*$, respectively, significantly more than their fractional volume of 18% and 1% of the global ocean. In contrast, the Pacific Ocean contributes only 30% to $\beta^*$ despite its fractional volume of 42% of the global ocean. By definition, the contribution of each basin to $\beta_{sat}$ is approximately proportional to the ocean volume contained in each basin (Fig. 6a). However, $\beta_{dis}$ is relatively low in the Atlantic and Arctic Oceans and high in the Pacific Ocean compared with their respective volumes (Fig. 6a), which may be understood by the Atlantic and Arctic Oceans interior being more ventilated and the Pacific Ocean interior being less ventilated than the rest of the ocean. Specifically, the low $\beta_{dis}$ and the relative large contribution from the Atlantic Ocean to $\beta^*$ are due to a large transfer of anthropogenic carbon into the ocean interior from a strong local physical ventilation and a transport of carbon from the Southern Ocean. In the Arctic Ocean, the transfer of anthropogenic carbon from the well ventilated Atlantic Ocean contributes towards a decrease in $\beta_{dis}$ and to a relatively large $\beta^*$. The Southern Ocean has a large anthropogenic carbon uptake from the atmosphere, but its contribution to $\beta^*$, as estimated from carbon storage, is relatively small due to a large carbon transport to the other basins (Fig. A1).

The Pacific and Indian Oceans contributions to $\gamma_{sat}$ are slightly smaller than expected from their fractional volumes (Fig. 6b), consistent with a low warming per unit volume in these basins (Fig. 1d). The Pacific and Indian Oceans contributions to $\gamma_{dis}$ and $\gamma_{reg}$ are significantly smaller than expected from their fractional volumes (Fig. 6b), indicating that there is no significant effect from changes in ventilation in these basins. This absence of any significant effect from changes in the ventilation with warming in the Pacific and the Indian Oceans leads to their much smaller contribution to $\gamma^*$ relative to their volumes (Fig. 6b).

The Atlantic Ocean has a contribution of 31% to $\gamma^*$, much larger than expected from its fractional volume of 18% of the global ocean. This large contribution from the Atlantic Ocean is primarily due to $\gamma_{dis}$ (Fig. 6b), and the reduction in the physical ventilation due to climate change. The Atlantic Ocean has a smaller contribution to $\gamma_{sat}$ than expected from its fractional volume (Fig. 6b), despite experiencing large warming (Fig. 1d), which suggests that the non-linearity of the carbonate system is important in this basin. Specifically, the Atlantic Ocean has a large increase in $DIC$ (Fig 1c) which acts to significantly reduce the magnitude of the negative $\gamma_{sat}$ driven solely by the effect of warming on solubility (see Eq. (20)). The Arctic Ocean has a contribution of 8% to $\gamma^*$, much larger than expected from its fractional volume of 1% of the global ocean. This large contribution from the Arctic Ocean is primarily associated with the saturated part of $\gamma^*$ (Fig. 6b) and a very large warming per unit volume in this basin (Fig 1d).

The Southern Ocean has a contribution of 38% to $\gamma_{dis}$, and of 53% to $\gamma_{reg}$, much larger than expected from its fractional volume (Fig. 6b). These large contributions indicate that changes in the physical ventilation and the accumulation of regenerated carbon due to climate change have a large effect on the Southern Ocean carbon storage. The Southern Ocean contribution to $\gamma_{sat}$ is also larger than expected from its fractional volume (Fig. 6b), consistent with a large warming per unit volume in this basin (Fig. 1d). Hence, the apparent small contribution of the Southern Ocean to $\gamma^*$, of 21%, is due to a compensation between (i) the large decrease in carbon storage associated with the combined decrease in solubility and physical ventilation, and (ii) the large increase in carbon storage associated with a longer residence time and accumulation of regenerated carbon in the Southern Ocean interior.

## 4    Dependence of the carbon cycle feedbacks on the Atlantic Meridional Overturning Circulation

The ocean carbon cycle feedbacks are controlled by ocean ventilation and the transfer of carbon from the mixed layer to the thermocline and deep ocean. The ocean ventilation involves the seasonal cycle of the mixed layer, the subduction process, and the effects of the eddy, gyre and overturning circulations. Despite this complexity the strength of the Atlantic Meridional Overturning Circulation (AMOC) and its weakening due to climate change is often used as a proxy for the large scale ocean ventilation. Here we investigate the dependence of the carbon cycle feedbacks to the AMOC strength and its weakening with climate change.

### 4.1    Insight from an idealised climate model with a meridional overturning

The idealised climate model of Katavouta et al. (2019) is used to investigate the control of the carbon cycle feedbacks by the AMOC. This idealised model consists of a slab atmosphere, two upper ocean boxes for the southern and northern high latitudes, two boxes for the mixed layer and the thermocline in the low and mid latitudes, and one box for the deep ocean (Fig. 7a). The model solves for the thermocline thickness from a volumetric balance between the surface cooling conversion of light to dense waters in the North Atlantic, the diapycnic transfer of dense to light waters in low and mid latitudes, and the conversion of dense to light waters in the Southern Ocean involving an Ekman transport partially compensated by a polewards mesoscale eddy transport (Gnanadesikan, 1999; Johnson et al., 2007; Marshall and Zanna, 2014). The model also accounts for the rate of subduction occurring in the Southern Ocean versus the tropics and subtropics through an isolation fraction for water remaining below the mixed layer and spreading northwards in the thermocline. The model solves for the ocean carbon cycle including physical and chemical transfers, but ignores biological transfers, and sediment and weathering interactions involving changes in the cycling of organic carbon or calcium carbonate. The ocean carbonate system is solved using the iterative algorithm of Follows et al. (2006). For the model closures and an explicit description of the model budgets and equations see Katavouta et al. (2019).

The model is first integrated to a pre-industrial steady state, with the distribution of temperature and $DIC$ depending on the pre-industrial strength of the overturning. The model is then forced by a 1% yr$^{-1}$ increase in atmospheric $CO_2$ concentration from a pre-industrial value of 280 ppm until atmospheric $CO_2$ quadruples over a 140-year period. This increase in atmospheric

$CO_2$ drives a radiative forcing: $R = a \ln(CO_2/CO_{2,0})$, where a = 5.35 W m$^{-2}$ (Myhre et al., 1998) and subscript $_0$ denotes the pre-industrial state. This radiative forcing then drives a radiative response, $\lambda \Delta T_{air}$, and a net planetary heat uptake given by the downward heat flux entering the system at the top of the atmosphere, $N_{TOA}$, (Gregory et al., 2004) such that $R = \lambda \Delta T_{air} + N_{TOA}$; where $T_{air}$ is the temperature of the slab atmosphere and $\lambda$ is the climate feedback parameter that is assumed constant and equal to 1 W m$^{-2}$K$^{-1}$ for simplicity. The ocean heat uptake, $N$, is estimated as the planetary heat uptake minus the atmospheric heat uptake, $N = N_{TOA} - c(\Delta T_{surf} - \Delta T_{air})$, where $c$=50 W m$^{-2}$ K$^{-1}$ is an air–sea heat transfer parameter and $T_{surf}$ is the ocean temperature at the surface. The ocean heat uptake is distributed equally over the ocean surface. For this model closure, the ocean heat uptake is more than 95% of the net planetary heat uptake.

This additional ocean heat uptake reduces the conversion of light to dense waters in the North Atlantic, $q_{NA}$, and leads to an overturning weakening, following

$$\Delta q_{NA} = -\frac{AN}{\rho C_p T_{contrast}}, \tag{24}$$

where $A$ is the model area covered by the low and mid latitudes, $\rho$ is a referenced ocean density, $C_p$ is the specific heat capacity for the ocean, and $T_{contrast}$ is the temperature contrast between light waters in the low and mid latitudes, involving the mixed layer and thermocline, and the dense waters in the deep ocean.

Three experiments were conducted using this idealised model forced by 1% yr$^{-1}$ increase in atmospheric $CO_2$: (i) a control experiment with a strong pre-industrial AMOC that experiences a weakening with warming as described by Eq. (24) (solid line in Fig. 7b); (ii) an experiment with a weak pre-industrial AMOC that experiences the same weakening with warming as in the control experiment (dotted line in Fig. 7b); and (iii) an experiment with a strong pre-industrial AMOC that experiences a doubled weakening with warming than in the control experiment, such that the right hand of Eq. (24) is multiplied by a factor of 2 (dashed line in Fig. 7b). In all the experiments, a fully coupled simulation (COU) and a biogeochemically coupled simulation (BGC) are used to estimate the carbon-cycle feedback parameters $\beta$ and $\gamma$.

In the idealised model, a weaker pre-industrial meridional overturning leads to a smaller carbon-concentration feedback parameter, $\beta$, during the centennial transient response to the increase in atmospheric $CO_2$ (Fig. 7c, black lines). The saturated part of $\beta$ does not directly depend on the AMOC strength (Fig. 7c, red lines). In contrast, a weaker pre-industrial AMOC leads to a more negative disequilibrium part of $\beta$ (Fig. 7c, blue lines) due to a weaker and slower transfer of anthropogenic carbon below the ocean surface. Hence, the pre-industrial AMOC controls $\beta$ through its effect on the physical ventilation via the disequilibrium carbon pool.

The strength of the pre-industrial meridional overturning has only a small impact on the carbon-climate feedback parameter, $\gamma$, that is associated with the saturated carbon pool (Fig. 7d, solid and dotted lines). In the idealised model, a different pre-industrial overturning is associated with a different pre-industrial ocean state, in terms of temperature and $DIC$, which has a small effect in the the saturated part of $\gamma$ due to the non-linearity of the carbonate chemistry. The carbon-climate feedback parameter, $\gamma$, is primarily controlled by the changes in AMOC with warming, and specifically by the dependence of the negative $\gamma_{dis}$ to the AMOC weakening with warming (Fig. 7d, solid and dashed lines). Hence, a more pronounced weakening in AMOC

with warming drives a more negative $\gamma$, as there is a more pronounced decrease in the in the physical transfer of carbon into the ocean interior.

## 4.2 Results from CMIP6 Earth system models

The strength and weakening of the AMOC are diagnosed as the maximum pre-industrial AMOC between $30^o$N and $50^o$N (Fig. 8) and the maximum AMOC change between $30^o$N and $50^o$N (Fig. 9), respectively, in 11 CMIP6 Earth system models (Table 3). There is a large spread in the pre-industrial strength of AMOC amongst the Earth system models (Fig. 8) with a range of 11.3 Sv in IPSL-CM6A-LR to 23.2 Sv in NorESM2-LM (Table 3). There is also a large spread in the response of the AMOC to climate change amongst the models (Fig. 9), with the magnitude of the AMOC weakening ranging from -5.6 Sv in IPSL-CM6A-LR to -18.7 Sv in MRI-ESM2 (Table 3). Generally, the models with a stronger pre-industrial AMOC simulate a larger weakening in AMOC with warming. Linear correlations are used to relate the AMOC variability amongst the models with the variability in the carbon-cycle feedbacks for the global ocean and the different ocean basins (Table 4).

For the global ocean, there is a moderate positive correlation between the pre-industrial AMOC and the ocean carbon-concentration feedback parameter, $\beta$, ($r$=0.72, Table 4), such that models with stronger pre-industrial AMOC have a more positive $\beta$. However, there is no significant correlation between the pre-industrial AMOC and the saturated or disequilibrium part of $\beta$ (Table 4) on the global scale. In the Atlantic Ocean, $\beta^*$ is strongly correlated with the pre-industrial AMOC in the Earth system models ($r$=0.87) and this correlation is due to the contribution from $\beta_{dis}$ (Table 4 and Fig. 10a), which suggests that a stronger pre-industrial AMOC leads to a larger $\beta^*$ via a stronger physical transfer of anthopogenic carbon below the ocean surface there, similar to the behaviour of the idealised model (Fig. 7c).

In the Southern Ocean, there is a statistically significant correlation between the pre-industrial AMOC and $\beta^*$ ($r$=0.82), that is not associated with either the saturated, the disequilibrium or the regenerated carbon pools (Table 4). A stronger pre-industrial AMOC is associated with a stronger return flow of interior water into the surface in the Southern Ocean upwelling branch of the overturning circulation (Marshall and Speer, 2012), and hence with a stronger local anthropogenic carbon uptake from the atmosphere in the BGC run there. However, in the Southern Ocean both the anthopogenic carbon uptake from the atmosphere and its transport to the other basins are also controlled by subduction (Sabine et al., 2004; Sallée et al., 2012, 2013) and the residual circulation involving the wind-driven Ekman transport and eddy fluxes (Lauderdale et al., 2013). Hence, there is no significant correlation between the pre-industrial AMOC and $\beta_{dis}$ in the Southern Ocean as other processes involving subduction of mode and intermediate waters, and the residual circulation are important in setting the pre-industrial physical ventilation and transport of carbon to other basins. There is also no significant correlation between the pre-industrial AMOC and $\beta^*$ in the Pacific and Indian Oceans, which suggests that other processes dominate the ocean ventilation there.

For the global ocean, the carbon-climate feedback parameter, $\gamma$, and its disequilibrium part, $\gamma_{dis}$, are positively correlated with the AMOC weakening due to climate change in the Earth system models ($r$=0.62, and $r$=0.71, respectively, Table 4). This correlation for the global ocean is to first order driven by the Atlantic Ocean, as there is no significant correlation between AMOC weakening and $\gamma^*$ in the Pacific, Indian and Southern Oceans or between AMOC weakening and $\gamma_{dis}$ in the Arctic Ocean (Table 4). In the Atlantic Ocean, a more pronounced AMOC weakening due to climate change is associated with a

larger reduction in the physical transfer of carbon below the ocean surface and so with a more negative $\gamma^*$ and $\gamma_{dis}$ (Fig. 10b), consistent with the inferences for the idealised model (Fig. 7d). In the Arctic Ocean, there is a moderate correlation between $\gamma^*$ and AMOC weakening ($r$=0.67, Table 4). This correlation between $\gamma^*$ and AMOC weakening in the Arctic Ocean is associated with the carbonate chemistry and the saturated carbon pool, $\gamma_{sat}$, (Table 4) and it appears to be related with changes in alkalinity (supplement Fig. C3).

In the Atlantic Ocean, the correlation between $\gamma_{reg}$ and AMOC weakening is negative ($r$=-0.64, Table 4), when excluding the CNRM-ESM2-1 model which has a much larger $\gamma_{reg}$ than the rest of the Earth system models (Table 3). A more pronounced AMOC weakening leads to a longer residence time in the Atlantic Ocean interior and so to a larger accumulation of regenerated carbon there and a more positive $\gamma_{reg}$ (Fig. 10b). The substantially larger change in the regenerated carbon pool with warming in CNRM-ESM2-1 than the rest of the models is probably due to a revised parameterization for organic mater remineralization (in PISCESv2-gas), its parametrisation of sedimentation, and its interactive riverine input; see Séférian et al. (2020) for a discussion of differences in ocean biogeochemistry amongst the Earth system models.

## 5    Discussion and Summary

Our study reveals the contribution of the Atlantic, Pacific, Southern, Indian and Arctic Oceans to the ocean carbon cycle feedbacks in a set of CMIP6 Earth system models. This basin-scale contribution is explained in terms of the effect of the carbonate chemistry, physical ventilation and biological processes. Experiments using an idealised climate model and diagnostics of the CMIP6 models suggest a dependence of the ocean carbon cycle feedbacks on the strength of the pre-industrial Atlantic Meridional Overturning Circulation (AMOC) and on the magnitudes of the AMOC weakening with warming.

The global ocean carbon-concentration feedback parameter, $\beta$, in CMIP6 models is controlled by a competition between opposing contributions from the saturated and the disequilibrium carbon pools, such that (i) the carbonate chemistry drives a positive $\beta$ that decreases with an increase in atmospheric $CO_2$ as the ocean acidifies and its capacity to buffer atmospheric $CO_2$ decreases, and (ii) the physical ventilation drives a negative $\beta$ that becomes less negative in time as the anthropogenic carbon is transferred from the ocean surface into the ocean interior. The global ocean carbon-climate feedback parameter, $\gamma$, in CMIP6 models is controlled by a competition between the combined decrease in the saturated and disequilibrium carbon pools and the increase in the regenerated carbon pool, such that (i) the decrease in solubility and weakening in physical ventilation with warming drive a negative $\gamma$, and (ii) the increase in the accumulation of regenerated carbon, associated with a longer residence time in the ocean interior with climate change, drives a positive $\gamma$.

### 5.1    Regional ocean carbon cycle feedbacks

The regional ocean carbon storage is controlled by the local air-sea carbon exchange and the transport of carbon by the ocean circulation. Here, the regional carbon cycle feedbacks are estimated from the regional ocean carbon storage, so that the effect of the ocean transport of carbon is included in the feedbacks. This transport effect acts to decrease the carbon-concentration feedback parameter, $\beta^*$, in the Southern Ocean, and increase $\beta^*$ in the Southern hemisphere subtropical gyres and the western

boundary of the Atlantic Ocean in CMIP6 models. The transport effect on $\beta^*$ is consistent with the view that while almost half of the anthopogenic carbon enters through the Southern Ocean, much of this carbon is transferred and stored in the Atlantic, Pacific and Indian Oceans (Sabine et al., 2004; Khatiwala et al., 2009; Frölicher et al., 2015). The transport effect also leads to a more spatially uniform and negative carbon-climate feedback parameter estimated in terms of carbon storage, $\gamma^*$, compared with the carbon-climate feedback parameter estimated in terms of cumulative carbon uptake.

The Atlantic, Pacific and Southern Oceans provide comparable contributions, of between 26% and 30%, to the global ocean carbon-concentration feedback, as estimated from the carbon storage, despite their different size. This large contribution from the Atlantic Ocean relative to its volume is associated with an enhanced physical transfer of anthropogenic carbon into the ocean interior due to a strong local ventilation and a transport of carbon from the Southern Ocean. The Arctic Ocean also provides a larger contribution to $\beta^*$ relative to its volume, consistent with observational based estimated of anthropogenic carbon storage

(Tanhua et al., 2009). The inter-model variability in the carbon-concentration feedback parameter, $\beta^*$, as estimated by the coefficient of variation, is relatively small in all the ocean basins, reflecting the use of a similar carbonate chemistry scheme in these models (Séférian et al., 2020) and a similar general large-scale circulation.

  The Atlantic Ocean has a large contribution to the ocean carbon-climate feedback relative to its size, as estimated from the carbon storage, in CMIP6 models, which is mainly due to the disequilibrium carbon pool. This result indicates that the

535 Atlantic Ocean experiences a strong weakening in physical ventilation with climate change. Specifically, the negative $\gamma^*$ per unit area is largest in the high latitudes of the North Atlantic in CMIP6 models, consistent with $\gamma$ diagnosed from the air-sea carbon fluxes in previous generation Earth system models (Roy et al., 2011; Ciais et al., 2013). This response is in accord with previous studies suggesting the effect of climate-driven changes in the circulation to the carbon storage is significant in the North Atlantic (Sarmiento et al., 1998; Winton et al., 2013; Bernardello et al., 2014). The Arctic Ocean also has a large

contribution to $\gamma^*$ relative to its size, primarily attributed to the saturated carbon pool and the large warming per unit volume in this basin.

  The relative small contribution from the Southern Ocean to $\gamma^*$ in CMIP6 is due to a partial compensation between the combined decrease in solubility and physical ventilation with warming, driving a negative $\gamma^*$, and the large increase in accumulation of regenerated carbon in the ocean interior with climate change, driving a positive $\gamma^*$. This compensation between

545 the decrease in solubility and physical ventilation with warming and the increase in accumulation of regenerated carbon with climate change in the Southern Ocean is also supported by sensitivity model experiments (Sarmiento et al., 1998; Bernardello et al., 2014). A large increase in the regenerated carbon pool of the Southern Ocean with climate change was also found in CMIP5 models (Schwinger et al., 2014; Ito et al., 2015). In the Indo-Pacific Ocean, the carbon-climate feedback in CMIP6 is primarily attributed to the saturated carbon pool and the decrease in solubility with warming. This response in the Indo-Pacific

Ocean is consistent with a feedback triggered by warming and the reduction in solubility dominating in low and mid-latitudes, as revealed in a CMIP5 model (Rodgers et al., 2020), and diagnostics in previous generation Earth system models (Roy et al., 2011; Ciais et al., 2013). The inter-model variability in $\gamma^*$ is mainly driven by the spread in the response of physical ventilation and circulation to climate change, and in the parametrization of ocean biogeochemical processes amongst the models.

## 5.2 Effect of the Altantic Meridional Overturning Circulation

Our sensitivity experiments with an idealised climate model reveal that a weaker pre-industrial Atlantic Meridional Overturning Circulation (AMOC) leads to a smaller ocean carbon-concentration feedback parameter, $\beta$. This dependence of $\beta$ to the pre-industrial AMOC is controlled by the disequilibrium carbon pool, and the physical transfer of anthropogenic carbon from the surface to the ocean interior. The ocean carbon-climate feedback parameter, $\gamma$, is instead primarily controlled by the weakening in the AMOC with warming, which reduces the physical ventilation and leads to the disequilibrium part of $\gamma$ becoming more negative in the idealised model.

Turning to the CMIP6 models, in the Atlantic Ocean, the carbon-concentration feedback parameter, $\beta^*$, is strongly correlated to the strength of the pre-industrial AMOC. This correlations of $\beta^*$ to AMOC is associated with the disequilibrium carbon pool and the effect of physical ventilation, such that Earth system models with stronger AMOC provide a more efficient transport of anthopogenic carbon into the ocean interior in the Atlantic Ocean, in accord with the behaviour in the idealised climate model. However, there is only a moderate correlation between AMOC strength and $\beta$ on a global scale, which suggests that other processes related to ocean ventilation, involving the wind-driven gyre circulation, mode water formation and subduction, and the Southern Ocean residual circulation, are also important for controlling the ocean carbon-concentration feedback. In contrast with our results based on the idealised model and the CMIP6 models, Roy et al. (2011) found no direct dependence of $\beta$ to the AMOC strength in older generation Earth system models, which suggests that either: (i) the sample of 4 models used in Roy et al. (2011) is too small to reveal the link between AMOC and $\beta$; or (ii) the dependence of $\beta$ to AMOC strength is more pronounced when considering estimates at the quadrupling of atmospheric $CO_2$ rather than estimates from years 2010 to 2100 in a high-emission scenario as in Roy et al. (2011).

In the Atlantic Ocean, the carbon-climate feedback parameter, $\gamma^*$, is strongly correlated to the AMOC weakening with climate change in the CMIP6 models, similarly to the behaviour of the idealised climate model. This correlation of $\gamma^*$ to AMOC weakening is: (i) primarily due to the disequilibrium carbon pool, with a more pronounced AMOC weakening driving a larger reduction in the physical transfer of carbon into the ocean interior and a more negative $\gamma^*$; and (ii) to a less extent due to the regenerated carbon pool, with a more pronounced AMOC weakening driving a larger accumulation of regenerated carbon in the ocean interior and a more positive $\gamma^*$. The dependence of the carbon-climate feedback to the AMOC weakening in the Atlantic Ocean is consistent with experiments using an older generation Earth system model (Crueger et al., 2008; Yoshikawa et al., 2008). In the Arctic Ocean, there is moderate correlation between $\gamma^*$ and AMOC weakening in the CMIP6 models, with this correlation being associated with the saturated carbon pool and changes in carbonate chemistry. There is no significant correlation between $\gamma^*$ and AMOC weakening in the Pacific, Indian and Southern Oceans. Hence on a global scale the weakening in the AMOC due to climate change has only a modest impact on ocean carbon uptake on the global and centennial scale, consistent with previous studies (Sarmiento et al., 1998; Joos et al., 1999; Zickfeld et al., 2008).

A caveat in our analysis is that it does not reveal the control of the carbon cycle feedbacks by other ventilation processes beyond the AMOC, such as the formation and subduction of mode and intermediate waters involving the seasonal mixed-layer cycle, and the horizontal and vertical circulation. An analysis for the ocean carbon uptake and storage in dynamically based

density space (Sallée et al., 2013; Meijers, 2014; Iudicone et al., 2016) is necessary to reveal the contribution of different water masses to $\beta$ and $\gamma$ (Roy et al., 2020, personal communication); and (ii) to better constrain the effect of local versus remote processes on the regional ocean carbon uptake and storage. Although our diagnostics reveal the relative contribution from the air-sea fluxes and the ocean transport of carbon to the regional carbon storage, further analysis is necessary to identify the effect of the ocean transport of carbon and other traces (e.g., temperature, salinity and nutrients) to the local air-sea fluxes. Furthermore, for a quadrupling of atmospheric $CO_2$ and on centennial time scale, as considered in this study, the carbon-concentration feedback is substantially larger than the carbon-climate feedback; however, this will not necessarily be the case after the emissions cease and the system adjusts towards equilibrium.

In summary, the Atlantic and Arctic Oceans provide a large contribution to the carbon cycle feedbacks, as estimated in terms of carbon storage, relative to their size in CMIP6. This large contribution from the Atlantic Ocean is associated with the physical ventilation. The Southern Ocean has a relatively small contribution to the carbon-concentration feedback as estimated in terms of carbon storage, despite its large anthropogenic carbon uptake, due to a large carbon transport to the other basins. The Southern Ocean also has a relatively small contribution to the carbon-climate feedback due to competing processes affecting the carbon storage in this basin. The more poorly-ventilated Indo-Pacific Ocean has a small contribution to the carbon cycle feedbacks relative to its size. On a global scale, the carbon cycle feedbacks depend on the strength of the pre-industrial AMOC and its weakening with warming, such that (i) a stronger pre-industrial AMOC leads to a more positive carbon-concentration feedback, and (ii) a more pronounced AMOC weakening due to climate change leads to a more negative carbon-climate feedback. This dependence of the carbon cycle feedbacks to AMOC is attributed to the Atlantic Ocean in the CMIP6 models.

**Appendix A:  Contribution from ocean carbon uptake and transport to the basin-scale $\beta^*$ and $\gamma^*$**

The Southern Ocean accounts on average for about half of the ocean anthopogenic carbon uptake (Fig. A1a red line) in CMIP6 Earth system model despite covering only 27% of the global ocean consistent with results from CMIP5 model during the historical period (Frölicher et al., 2015), as well as observational based estimates (Mikaloff-Fletcher et al., 2006). However, a large portion of the anthropogenic carbon entering through the Southern Ocean is transported into the Atlantic, Indian and Pacific Oceans, with the Atlantic Ocean receiving most of this carbon (Fig. A1a) in CMIP6 models, consistent with observational-based estimates (Sabine et al., 2004; Khatiwala et al., 2009). Consequently, the carbon-concentration feedback parameter estimated based on the regional cumulative carbon uptake, $\beta$, (Fig. A1b, red lines) is substantially larger in the Southern Ocean, but smaller in the Atlantic, Pacific and Indian Oceans than when estimated based on the regional carbon storage, $\beta^*$ (Fig. A1b, black lines). The feedback parameter $\beta$ is also smaller than $\beta^*$ in the Arctic Ocean indicating that ocean transport supply anthropogenic carbon to the Arctic in CMIP6 models.

The net effect of climate change is a carbon loss from the ocean to the atmosphere and a negative $\gamma$ in the Atlantic, Indian, Pacific and Southern Oceans (Fig. A1c, red lines). The ocean transport acts to reduce the negative $\gamma$ in the Atlantic, Indian and Southern Oceans, but to increase the negative $\gamma$ in the Pacific Ocean (Fig. A1c, compare the red with the black lines). In contrast, in the Arctic Ocean the decrease in sea-ice coverage due to climate change leads to an increase in the ocean carbon

uptake from the atmosphere and a positive $\gamma$. The ocean transport opposes this ocean gain in carbon from the atmosphere in the Arctic and leads to a reduction in the ocean carbon inventory and a negative $\gamma^*$ (Fig. A1c, black lines).

## Appendix B: Contribution from saturated, disequilibrium and regenerated carbon pools to the regional $\beta^*$ and $\gamma^*$

The saturated part of $\beta^*$ is positive everywhere in the ocean (Fig. B1a). By definition, $\beta_{sat}$ depends on the ocean volume, so is larger in deeper regions. The disequilibrium part of $\beta^*$ is small on the shelves and in the well-ventilated North Atlantic, and large in the more poorly-ventilated Pacific Ocean and in deep ocean regions (Fig. B1a). This vertically integrated regional view for the carbon storage reflects that in the BGC simulation $\Delta DIC_{dis}$ is small in the upper ocean, but large and negative in deeper water outside the North Atlantic (see Figs 10 and 11 in Arora et al., 2020). The regenerated part of $\beta^*$ is negligible in the CMIP6 models.

The saturated part of $\gamma^*$ is negative and large in the Arctic Ocean (Fig. B1b), reflecting a large decrease in solubility due to large warming per unit volume (Fig. 1d). Along the western boundary of the North Atlantic Ocean $\gamma_{sat}$ is slightly positive (Fig. B1b), which is related to a large regional gain in carbon (Fig 2a), acting to reduce the carbonate system sensitivity to changes in temperature (see Eq. (20)), combined with a large regional increase in alkalinity due to climate change. The disequilibrium part of $\gamma^*$ is negative and large in the North Atlantic, along the western boundary of the Atlantic Ocean and in the Southern Ocean, indicating that the effect of weakening in the ventilation is significant in these regions. In the Arctic Ocean, the decrease in sea-ice coverage drives an increase in the ocean carbon uptake and leads to areas with a large positive $\gamma_{dis}$. The regenerated part of $\gamma^*$ has a similar spatial pattern but of opposite sign to $\gamma_{dis}$ (Fig. B1b), which suggests that $\gamma_{reg}$ is dominated by changes in the ocean ventilation. Along the equator and in the eastern boundary upwelling regions, $\gamma_{dis}$ and $\gamma_{reg}$ are slightly positive and negative, respectively. This negative $\gamma_{reg}$ and positive $\gamma_{dis}$ in regions of upwelling is consistent with a coupling between the regenerated and the disequilibrium carbon pools, such that a positive $\Delta DIC_{reg}$ brought into the surface from depth leads to an excess of carbon at the surface that is converted into a positive $\Delta DIC_{dis}$ (i.e. oversaturation) (Ito and Follows, 2013).

*Acknowledgements.* The authors acknowledge the World Climate Research Programme, which, through its Working Group on Coupled Modelling, coordinated and promoted CMIP6, the climate modelling groups for producing and making available their model output, and the Earth System Grid Federation for archiving the data and providing access. This work was supported by the UK Natural Environmental Research Council grants NE/T007788/1 and NE/T010657/1. We thank Dr Jörg Schwinger and an anonymous reviewer for their constructive comments.

*Data availability.* The data used here are from the CMIP6 simulations performed by the various modelling groups and available from the CMIP6 archive (https://esgf-node.llnl.gov/search/cmip6).

*Author contributions.* Anna Katavouta conducted the analysis, and led the interpretation of the results and writing of the manuscript. Richard
G. Williams contributed to the design of the analysis, the interpretation of the results, and the writing of the manuscript.

*Competing interests.* The authors declare that they have no conflict of interest.

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

**Table 1.** List of the 11 CMIP6 Earth system models used in this study along with references for the model description.

| Earth system model | Reference |
| --- | --- |
| ACCESS-ESM1.5 | Ziehn et al. (2020) |
| CanESM5 | Swart et al. (2019) |
| CanESM5-CanOE | Swart et al. (2019) |
| CNRM-ESM2-1 | Séférian et al. (2019) |
| GFDL-ESM4 | Dunne et al. (2020) |
| IPSL-CM6A-LR | Boucher et al. (2020) |
| MIROC-ES2L | Hajima et al. (2020) |
| MPI-ESM1.2-LR | Mauritsen et al. (2019) |
| MRI-ESM2 | Yukimoto et al. (2019) |
| NorESM2-LM | Seland et al. (2020) |
| UKESM1-0-LL | Sellar et al. (2019) |

**Table 2.** Carbon-concentration feedback parameter based on carbon storage, $\beta^*$ (PgC ppm$^{-1}$), and carbon-climate feedback parameter based on carbon storage, $\gamma^*$ (PgC K$^{-1}$), for the global ocean and different ocean basins in 11 CMIP6 Earth system models, along with the inter-model mean, standard deviation and coefficient of variation (CV), estimated as the standard deviation divided by the mean. The estimates are based on the fully coupled simulation (COU) and the biogeochemically coupled simulation (BGC) under the 1% yr$^{-1}$ increasing $CO_2$ experiment. Diagnostics are from years 121 to 140 (the 20 years up to quadrupling of atmospheric $CO_2$). Note that for the global ocean $\beta^*$ and $\gamma^*$ are equivalent to $\beta$ and $\gamma$.

| Parameter $\beta^*$ | | | | | | |
|---|---|---|---|---|---|---|
| model | Global ocean | Pacific | Southern | Atlantic | Indian | Arctic |
| ACCESS-ESM1.5 | 0.901 | 0.247 | 0.277 | 0.241 | 0.099 | 0.029 |
| CanESM5 | 0.794 | 0.256 | 0.210 | 0.192 | 0.098 | 0.027 |
| CanESM5-CanOE | 0.750 | 0.241 | 0.202 | 0.178 | 0.094 | 0.024 |
| CNRM-ESM2-1 | 0.794 | 0.251 | 0.191 | 0.210 | 0.103 | 0.027 |
| GFDL-ESM4 | 0.933 | 0.278 | 0.250 | 0.257 | 0.107 | 0.028 |
| IPSL-CM6A-LR | 0.777 | 0.249 | 0.201 | 0.192 | 0.101 | 0.023 |
| MIROC-ES2L | 0.762 | 0.220 | 0.214 | 0.196 | 0.083 | 0.034 |
| MPI-ESM1.2-LR | 0.803 | 0.237 | 0.235 | 0.211 | 0.085 | 0.022 |
| MRI-ESM2 | 0.966 | 0.306 | 0.235 | 0.258 | 0.121 | 0.030 |
| NorESM2-LM | 0.815 | 0.197 | 0.252 | 0.236 | 0.088 | 0.024 |
| UKESM1-0-LL | 0.736 | 0.198 | 0.192 | 0.203 | 0.102 | 0.029 |
| mean | 0.821 | 0.244 | 0.224 | 0.216 | 0.098 | 0.027 |
| std | 0.077 | 0.032 | 0.028 | 0.028 | 0.011 | 0.004 |
| CV | 0.094 | 0.131 | 0.125 | 0.130 | 0.112 | 0.148 |
| Parameter $\gamma^*$ | | | | | | |
| model | Global ocean | Pacific | Southern | Atlantic | Indian | Arctic |
| ACCESS-ESM1.5 | -20.56 | -8.79 | -6.40 | -3.34 | -1.15 | -0.64 |
| CanESM5 | -13.94 | -5.41 | -2.59 | -2.65 | -1.94 | -0.87 |
| CanESM5-CanOE | -11.21 | -4.40 | -2.22 | -2.16 | -1.26 | -0.72 |
| CNRM-ESM2-1 | -1.54 | -2.16 | 0.90 | -0.04 | 0.61 | -0.61 |
| GFDL-ESM4 | -18.41 | -5.70 | -2.58 | -6.35 | -2.06 | -1.69 |
| IPSL-CM6A-LR | -11.82 | -4.76 | -2.28 | -3.07 | -1.03 | -0.48 |
| MIROC-ES2L | -19.08 | -3.06 | -4.35 | -7.40 | -1.84 | -2.54 |
| MPI-ESM1.2-LR | -16.70 | -2.77 | -5.39 | -5.99 | -1.19 | -0.81 |
| MRI-ESM2 | -27.64 | -10.42 | 0.19 | -12.19 | -1.64 | -2.61 |
| NorESM2-LM | -18.00 | -2.43 | -8.3 | -6.13 | -0.13 | -0.87 |
| UKESM1-0-LL | -11.94 | -2.84 | -2.23 | -3.28 | -2.05 | -1.02 |
| mean | -15.53 | -4.80 | -3.21 | -4.78 | -1.24 | -1.17 |
| std | 6.66 | 2.69 | 2.73 | 3.30 | 0.84 | 0.76 |
| CV | 0.43 | 0.56 | 0.85 | 0.69 | 0.68 | 0.65 |

**Table 3.** Carbon-concentration feedback parameter based on carbon storage, $\beta^*$ (PgC ppm$^{-1}$), and carbon-climate feedback parameter based on carbon storage, $\gamma^*$ (PgC K$^{-1}$), in the Atlantic Ocean, and the contribution from the saturated, disequilibrium and regenerated carbon pools, along with the strength of the Atlantic Meridional Overturning Circulation, AMOC in $Sv$, at the pre-industrial and its weakening relative to the pre-industrial due to climate change. The strength and weakening of the AMOC are estimated as the maximum pre-industrial AMOC and maximum AMOC change between $30^o$N and $50^o$N, respectively (Figs 8 and 9). Results are shown for 11 CMIP6 Earth system models, along with the inter-model mean, standard deviation and coefficient of variation (CV), estimated as the standard deviation divided by the mean. The estimates are based on the fully coupled simulation (COU) and the biogeochemically coupled simulation (BGC) under the 1% yr$^{-1}$ increasing $CO_2$ experiment. Diagnostics are from years 121 to 140 (the 20 years up to quadrupling of atmospheric $CO_2$).

| Parameter $\beta^*$ | | | | | |
|---|---|---|---|---|---|
| model | pre-industrial AMOC | $\beta^*$ | $\beta_{sat}$ | $\beta_{dis}$ | $\beta_{reg}$ |
| ACCESS-ESM1.5 | 22.24 | 0.241 | 0.723 | -0.482 | 0.000 |
| CanESM5 | 13.16 | 0.192 | 0.763 | -0.571 | 0.000 |
| CanESM5-CanOE | 12.23 | 0.178 | 0.721 | -0.542 | 0.000 |
| CNRM-ESM2-1 | 16.41 | 0.210 | 0.741 | -0.543 | 0.012 |
| GFDL-ESM4 | 21.74 | 0.257 | 0.756 | -0.513 | 0.014 |
| IPSL-CM6A-LR | 11.32 | 0.192 | 0.731 | -0.539 | 0.000 |
| MIROC-ES2L | 14.58 | 0.196 | 0.744 | -0.549 | 0.001 |
| MPI-ESM1.2-LR | 22.36 | 0.211 | 0.730 | -0.519 | 0.000 |
| MRI-ESM2 | 22.50 | 0.258 | 0.743 | -0.485 | 0.000 |
| NorESM2-LM | 23.18 | 0.236 | 0.751 | -0.515 | 0.000 |
| UKESM1-0-LL | 15.61 | 0.203 | 0.725 | -0.526 | 0.004 |
| mean | 17.76 | 0.216 | 0.739 | -0.526 | 0.003 |
| std | 4.68 | 0.028 | 0.014 | 0.027 | 0.005 |
| CV | 0.26 | 0.130 | 0.019 | 0.051 | 1.667 |
| Parameter $\gamma^*$ | | | | | |
| model | AMOC weakening | $\gamma^*$ | $\gamma_{sat}$ | $\gamma_{dis}$ | $\gamma_{reg}$ |
| ACCESS-ESM1.5 | -9.73 | -3.34 | -0.17 | -4.49 | 1.32 |
| CanESM5 | -6.86 | -2.65 | -0.32 | -2.79 | 0.47 |
| CanESM5-CanOE | -5.81 | -2.16 | -0.66 | -2.56 | 1.06 |
| CNRM-ESM2-1 | -10.53 | -0.04 | -2.57 | -4.94 | 7.46 |
| GFDL-ESM4 | -12.77 | -6.35 | -1.11 | -7.70 | 2.47 |
| IPSL-CM6A-LR | -5.63 | -3.07 | -2.00 | -4.18 | 3.11 |
| MIROC-ES2L | -11.54 | -7.40 | -1.89 | -9.04 | 3.53 |
| MPI-ESM1.2-LR | -10.27 | -5.99 | -2.17 | -5.73 | 1.91 |
| MRI-ESM2 | -18.74 | -12.19 | -1.84 | -14.25 | 3.90 |
| NorESM2-LM | -15.26 | -6.13 | -1.93 | -7.06 | 2.86 |
| UKESM1-0-LL | -9.66 | -3.28 | -0.80 | -3.56 | 1.08 |
| mean | -10.62 | -4.78 | -1.41 | -6.03 | 2.65 |
| std | 3.96 | 3.30 | 0.82 | 3.41 | 1.94 |
| CV | 0.37 | 0.69 | 0.58 | 0.57 | 0.73 |

**Table 4.** Correlation between the ocean carbon-concentration feedback parameter based on carbon storage, $\beta^*$, and the pre-industrial strength of the Atlantic Meridional Overturning Circulation (AMOC), and correlation between the ocean carbon-climate feedback parameter based on carbon storage, $\gamma^*$, and the AMOC weakening due to climate change, along with the contribution from the saturated, disequilibrium and regenerated carbon pools based on 11 CMIP6 models (Table 1). For the regenerated part of $\gamma^*$, correlations estimated excluding the CNRM-ESM2-1 model, which has substantially larger changes in the regenerated carbon with warming than the rest of the Earth system models, are presented as $\widehat{\gamma}_{reg}$. The correlation is expressed as a correlation coefficient, $r$, and a $p$-value. The level of significance is assumed as 0.05 and the statistically significant correlations with $p<0.05$ are highlighted by bold text. Diagnostics are from years 121 to 140 (the 20 years up to quadrupling of atmospheric $CO_2$). Note that for the global ocean $\beta^*$ and $\gamma^*$ are equivalent to $\beta$ and $\gamma$.

| $r$ and $p$-value for correlation between $\beta^*$ and pre-industrial AMOC strength | | | | | |
|---|---|---|---|---|---|
| ocean basin | $\beta^*$ | $\beta_{sat}$ | $\beta_{dis}$ | $\beta_{reg}$ | |
| Global Ocean | $r$=**0.72**, $p$=**0.013** | $r$=-0.10, $p$=0.773 | $r$=0.51, $p$=0.107 | $r$=0.23, $p$=0.493 | |
| Pacific Ocean | $r$=0.17, $p$=0.621 | $r$=-0.06, $p$=0.859 | $r$=0.13, $p$=0.703 | $r$=0.24, $p$=0.484 | |
| Southern Ocean | $r$=**0.82**, $p$=**0.002** | $r$=-0.16, $p$=0.646 | $r$=0.59, $p$=0.058 | $r$=0.31, $p$=0.360 | |
| Atlantic Ocean | $r$=**0.87**, $p$**<0.001** | $r$=0.14, $p$=0.674 | $r$=**0.80**, $p$=**0.003** | $r$=0.12, $p$=0.723 | |
| Indian Ocean | $r$=0.13, $p$=0.712 | $r$=-0.25, $p$=0.457 | $r$=0.30, $p$=0.378 | $r$=-0.04, $p$=0.910 | |
| Arctic Ocean | $r$=-0.03, $p$=0.933 | $r$=-0.11, $p$=0.749 | $r$=0.05, $p$=0.880 | $r$=0.17, $p$=0.625 | |
| $r$ and $p$-value for correlation between $\gamma^*$ and AMOC weakening with climate change | | | | | |
| ocean basin | $\gamma^*$ | $\gamma_{sat}$ | $\gamma_{dis}$ | $\gamma_{reg}$ | $\widehat{\gamma}_{reg}$ |
| Global Ocean | $r$=**0.62**, $p$=**0.041** | $r$=0.54, $p$=0.088 | $r$=**0.71**, $p$=**0.015** | $r$=-0.17, $p$=0.608 | $r$=-0.28 $p$=0.428 |
| Pacific Ocean | $r$=0.32, $p$=0.338 | $r$=0.44, $p$=0.176 | $r$=-0.03, $p$=0.920 | $r$=0.35, $p$=0.292 | $r$=0.41 $p$=0.234 |
| Southern Ocean | $r$=0.05, $p$=0.884 | $r$=0.14, $p$=0.675 | $r$=0.35, $p$=0.293 | $r$=-0.30, $p$=0.376 | $r$=-0.37 $p$=0.294 |
| Atlantic Ocean | $r$=**0.80**, $p$=**0.003** | $r$=0.36, $p$=0.271 | $r$=**0.89**, $p$**<0.001** | $r$=-0.36, $p$=0.279 | $r$=**-0.64** $p$=**0.048** |
| Indian Ocean | $r$=-0.05, $p$=0.875 | $r$=0.16, $p$=0.636 | $r$=-0.13, $p$=0.697 | $r$=-0.06, $p$=0.851 | $r$=-0.14 $p$=0.707 |
| Arctic Ocean | $r$=**0.67**, $p$=**0.025** | $r$=**0.69**, $p$=**0.019** | $r$=0.43, $p$=0.191 | $r$=-0.10, $p$=0.779 | $r$=-0.10 $p$=0.779 |

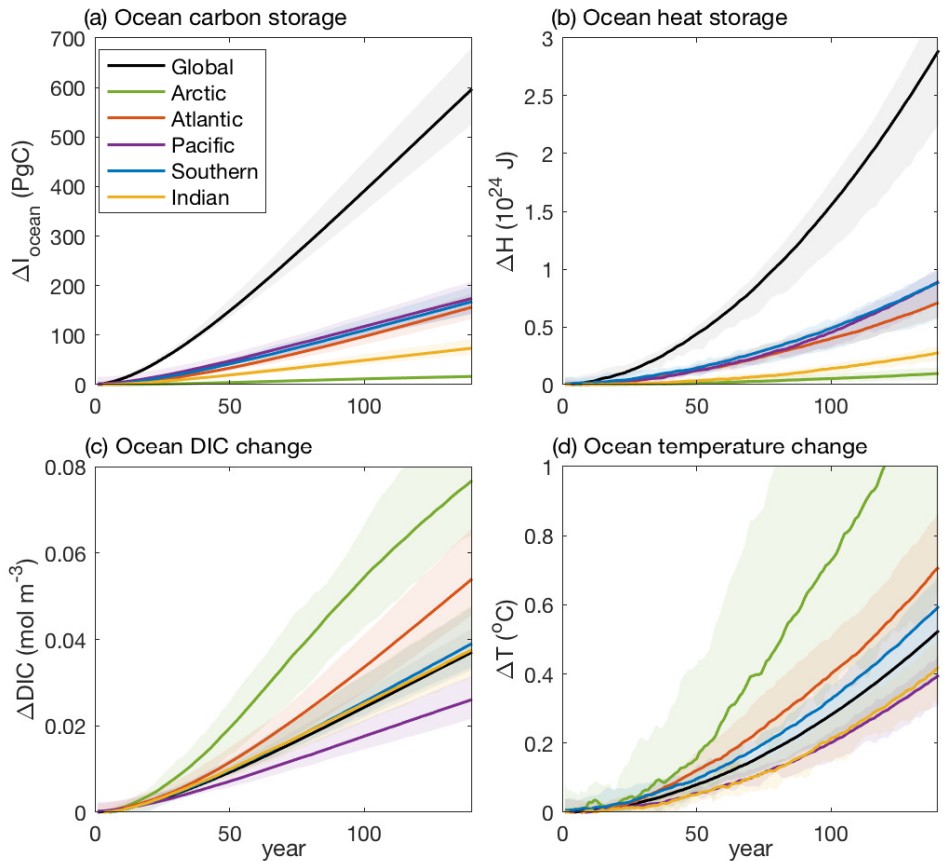

**Figure 1.** Carbon and heat storage for the global ocean and different ocean basins in CMIP6 Earth system models: (a) ocean carbon content changes relative to the pre-industrial in PgC, (b) ocean heat content changes relative to the pre-industrial in J, (c) changes in the ocean dissolved inorganic carbon relative to the pre-industrial in mol m$^{-3}$, expressing the ocean carbon storage changes per volume; and (d) changes in ocean temperature relative to the pre-industrial in $^o$C, expressing the ocean heat changes per volume. The solid lines show the model mean and the shading the model range based on the the 1 % yr$^{-1}$ increasing $CO_2$ experiment over 140 years in 11 CMIP6 models (Table 1). For the definition of the ocean basins see supplement Fig. C1.

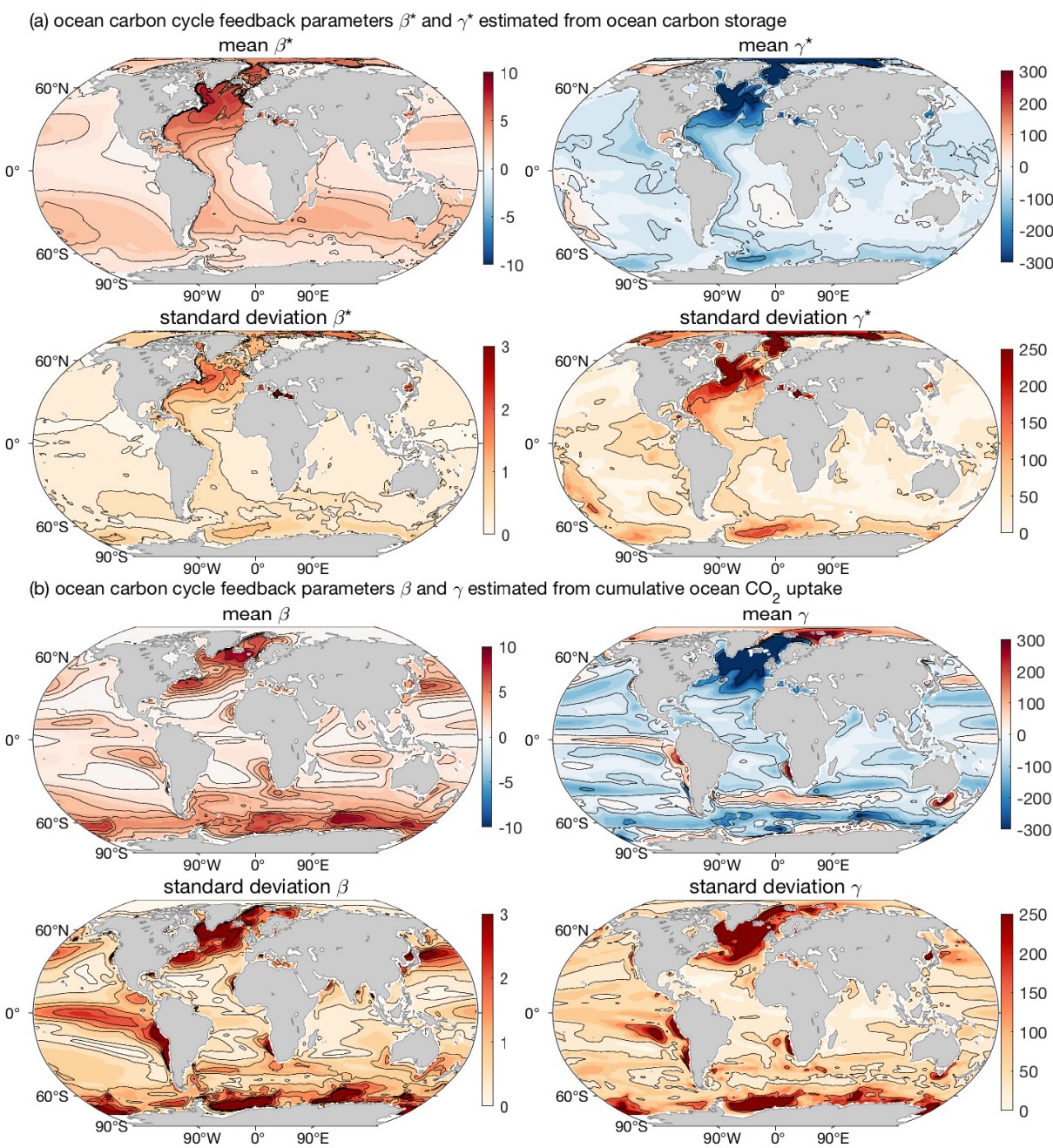

**Figure 2.** Geographical distribution of the carbon cycle feedback parameters normalised by area: (a) $\beta^*$ (gC ppm$^{-1}$ m$^{-2}$) and $\gamma^*$ (gC K$^{-1}$ m$^{-2}$), estimated based on the regional ocean carbon storage; and (b) $\beta$ (gC ppm$^{-1}$ m$^{-2}$) and $\gamma$ (gC K$^{-1}$ m$^{-2}$), estimated based on the regional cumulative ocean carbon uptake. Results are shown as the inter-model mean and standard deviation based on 11 CMIP6 Earth system models (Table 1). The estimates are based on the fully coupled simulation (COU) and the biogeochemically coupled simulation (BGC) under the 1% yr$^{-1}$ increasing CO$_2$ experiment. Diagnostics are from years 121 to 140 (the 20 years up to quadrupling of atmospheric CO$_2$).

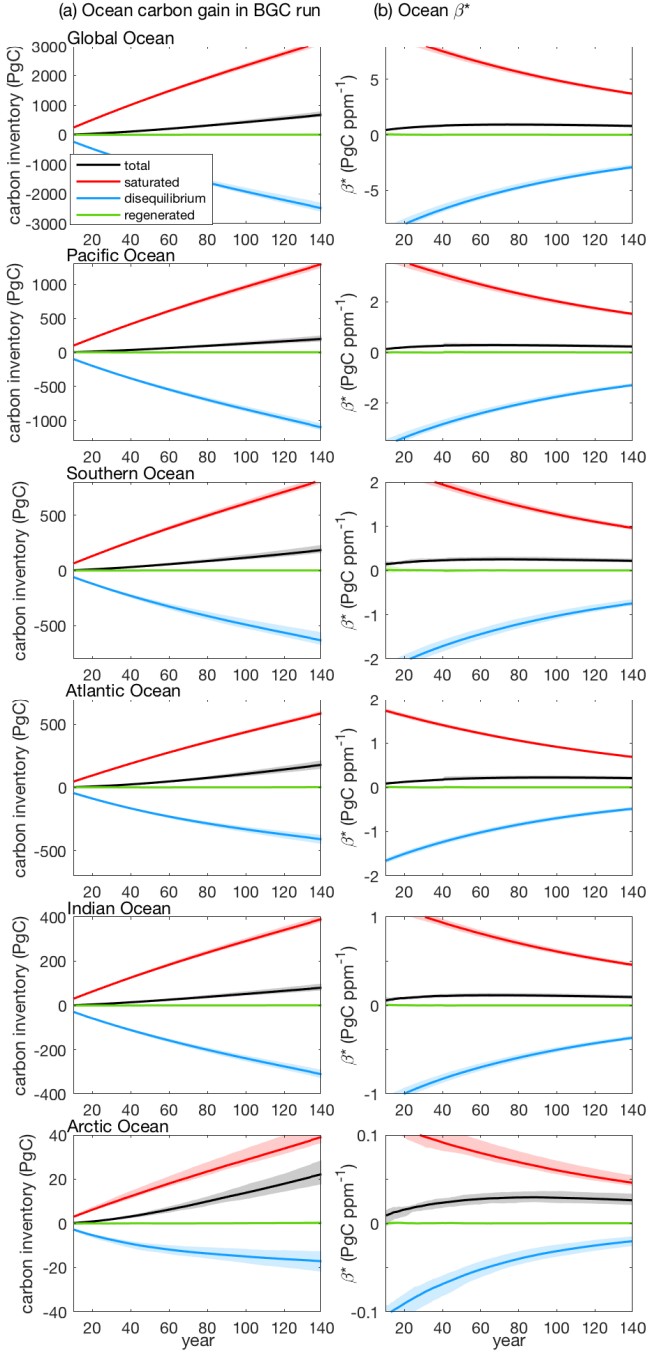

**Figure 3.** Ocean carbon storage and ocean carbon-concentration feedback parameter, $\beta^*$, for the global ocean and different ocean basins, along with the contribution from the saturated, disequilibrium and regenerated carbon pools in CMIP6 Earth system models: (a) ocean carbon inventory changes relative to the pre-industrial in the biogeochemically coupled simulation (BGC); and (b) ocean carbon-concentration feedback parameter based on carbon storage, $\beta^*$ (PgC ppm$^{-1}$). The solid lines show the model mean and the shading the model range based on the 1 % yr$^{-1}$ increasing $CO_2$ experiment over 140 years in 11 CMIP6 models (Table 1). Note that for the global ocean $\beta^*$ is equivalent to $\beta$.

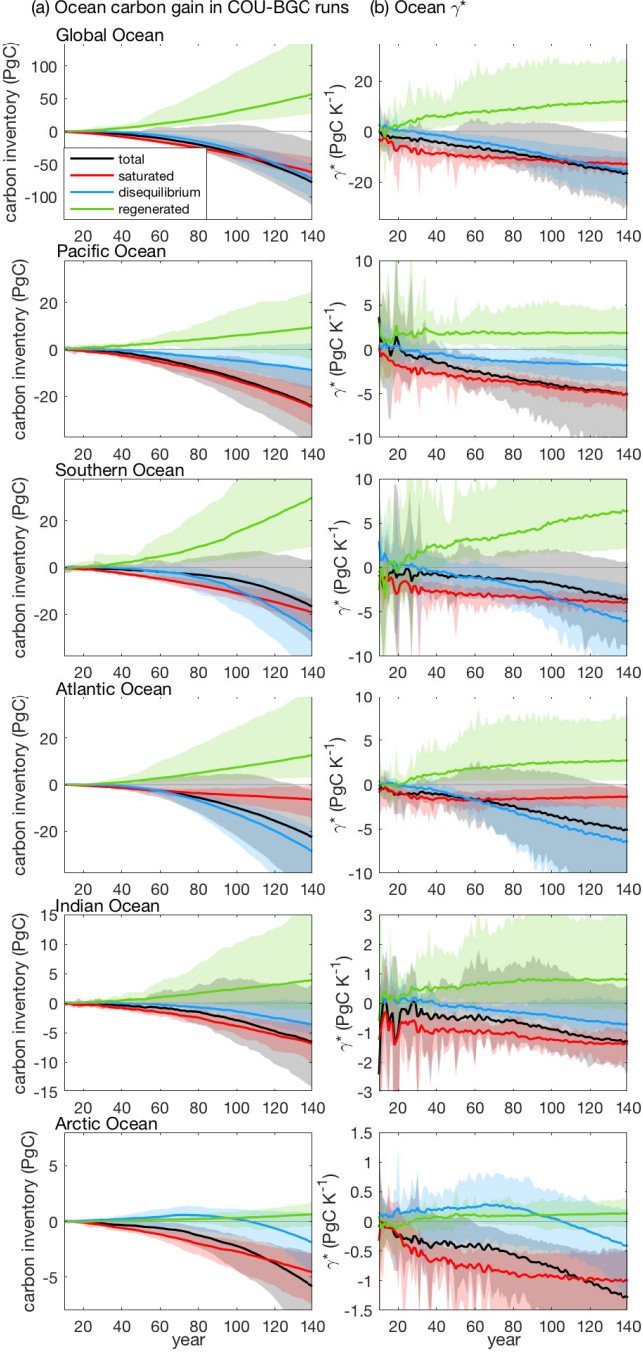

**Figure 4.** Ocean carbon storage and ocean carbon-climate feedback parameters, $\gamma^*$, for the global ocean and different ocean basins, along with the contribution from the saturated, disequilibrium and regenerated carbon pools in CMIP6 Earth system models: (a) ocean carbon inventory changes in the fully coupled simulation (COU) minus the biogeochemically coupled simulation (BGC); and (b) ocean carbon-climate feedback parameter based on carbon storage, $\gamma^*$ (PgC K$^{-1}$). The solid lines show the model mean and the shading the model range based on the the 1 % yr$^{-1}$ increasing CO$_2$ experiment over 140 years in 11 CMIP6 models (Table 1). Note that for the global ocean $\gamma^*$ is equivalent to $\gamma$.

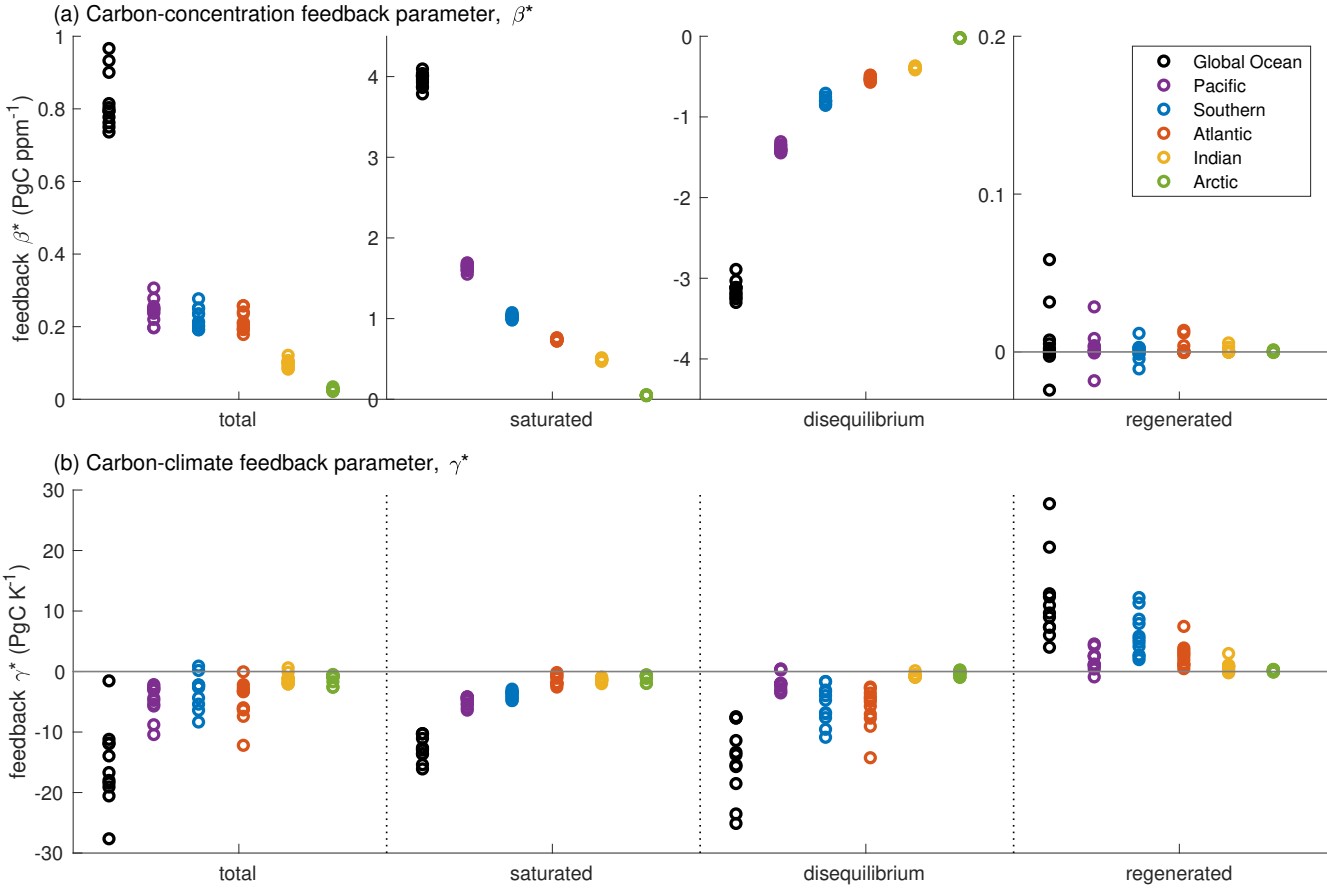

**Figure 5.** Carbon cycle feedback parameters based on carbon storage, along with the contribution from the saturated, disequilibrium and regenerated carbon pools, for the global ocean and the different ocean basins in 11 CMIP6 Earth system models (Table 1): (a) carbon-concentration feedback parameter, $\beta^*$ (PgC ppm$^{-1}$); and (b) carbon-climate feedback parameter $\gamma^*$ (PgC K$^{-1}$). The estimates are based on the fully coupled simulation (COU) and the biogeochemically coupled simulation (BGC) under the 1% yr$^{-1}$ increasing $CO_2$ experiment. Diagnostics are from years 121 to 140 (the 20 years up to quadrupling of atmospheric $CO_2$). For the inter-model mean and standard deviation of these estimates see supplement Table C1. Note that for the global ocean $\beta^*$ and $\gamma^*$ are equivalent to $\beta$ and $\gamma$.

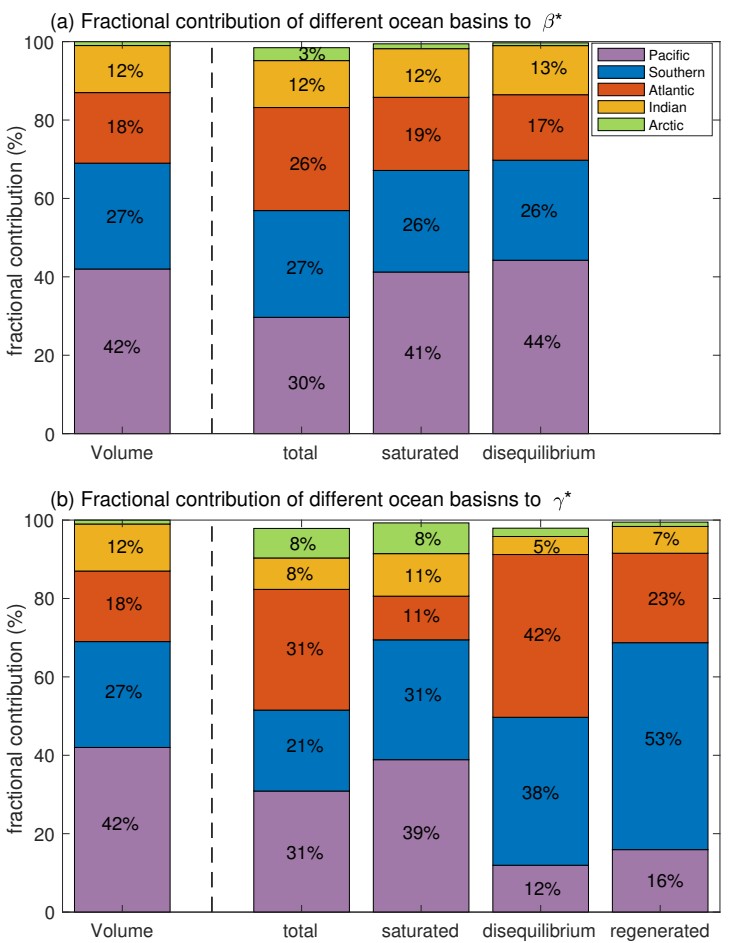

**Figure 6.** The fractional contribution (in %) of different ocean basins to the total volume of the ocean and to the ocean carbon cycle feedback parameters based on carbon storage, along with the contribution from the saturated, disequilibrium and regenerated carbon pools based on the inter-model mean of 11 CMIP6 models (Table 1): (a) carbon-concentration feedback parameter, $\beta^*$; and (b) carbon-climate feedback parameter, $\gamma^*$. The estimates are based on the fully coupled simulation (COU) and the biogeochemically coupled simulation (BGC) under the 1% yr$^{-1}$ increasing $CO_2$ experiment. Diagnostics are from years 121 to 140 (the 20 years up to quadrupling of atmospheric $CO_2$). The regenerated part of $\beta^*$ is omitted as its contribution is negligible in all basins (Fig. 5). The Arctic Ocean has a fractional volume of $\sim$1% of the global ocean, and a fractional contribution of 1.2%, 0.7%, 2% and 1% to $\beta_{sat}$, $\beta_{dis}$, $\gamma_{dis}$ and $\gamma_{reg}$, respectively (not explicitly shown in the figure). The combined fractional contribution of the 5 ocean basins to $\beta^*$ and $\gamma^*$ is less than 100% due to a small contribution, <2%, from semi-enclosed seas with a fractional volume of less than 0.5% of the global ocean (supplement Fig. C1).

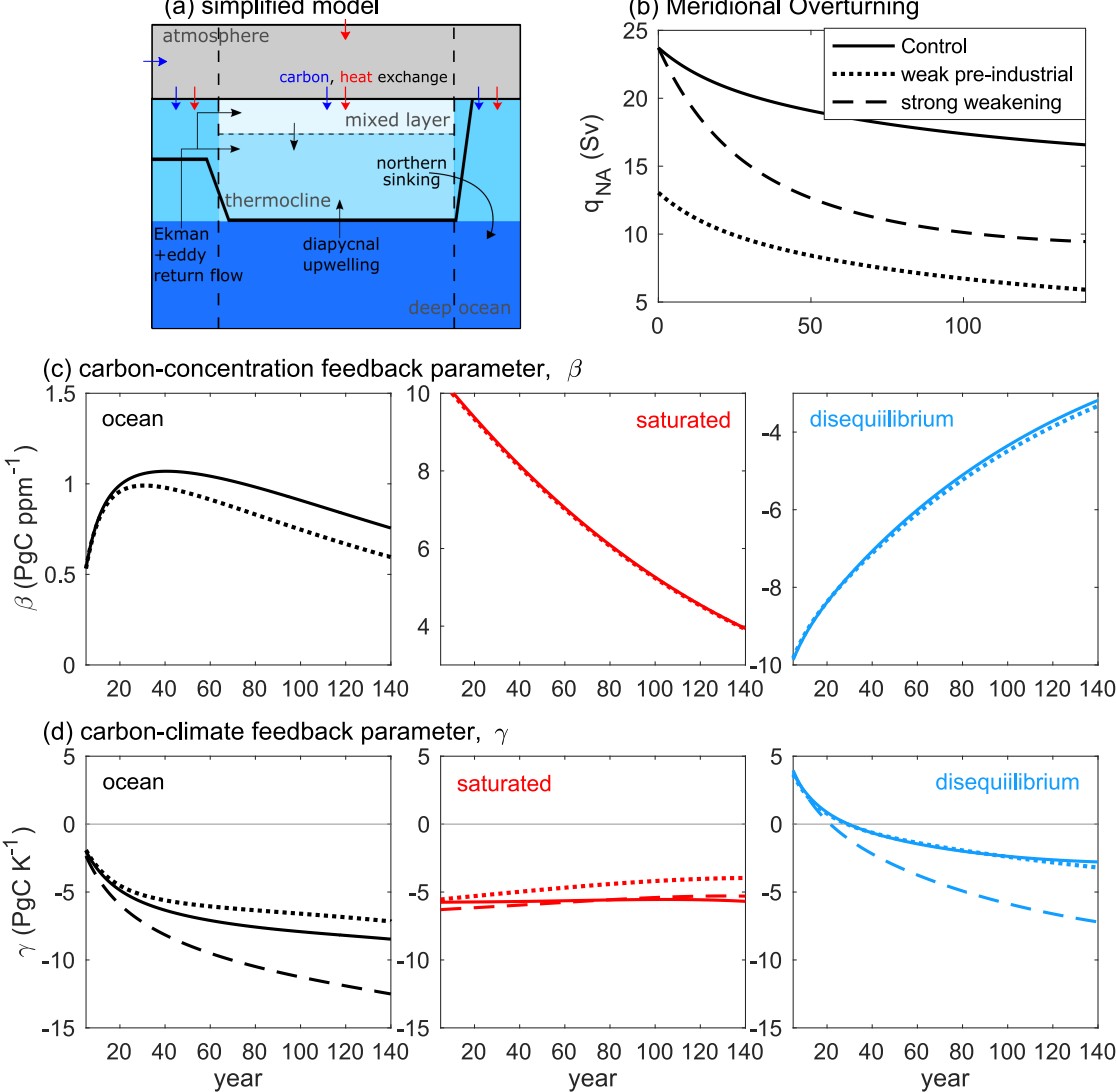

**Figure 7.** (a) A simplified climate model with overturning circulation, including a slab atmosphere, an upper layer of light water consisting of a thermocline layer and a surface mixed layer in the low and mid latitudes, two upper layers at southern and northern high latitudes, and a lower layer of dense water. (b) Meridional overturning in the three experiments forced by 1% yr$^{-1}$ increase in atmospheric $CO_2$: (i) control experiment (solid line), (ii) weaker pre-industrial overturning (dotted line), and (iii) larger reduction in the overturning with climate change (dashed line). (c) The ocean carbon-concentration feedback parameter, $\beta$; and (d) the ocean carbon-climate feedback parameter, $\gamma$, along with their saturated and disequilibrium parts, in the experiments with different meridional overturning. The estimates of $\beta$ and $\gamma$ are based on the fully coupled simulations (COU) and the biogeochemically coupled simulations (BGC). Note that for the "global" ocean of the simplified model $\beta$ and $\gamma$ are equivalent to $\beta^*$ and $\gamma^*$.

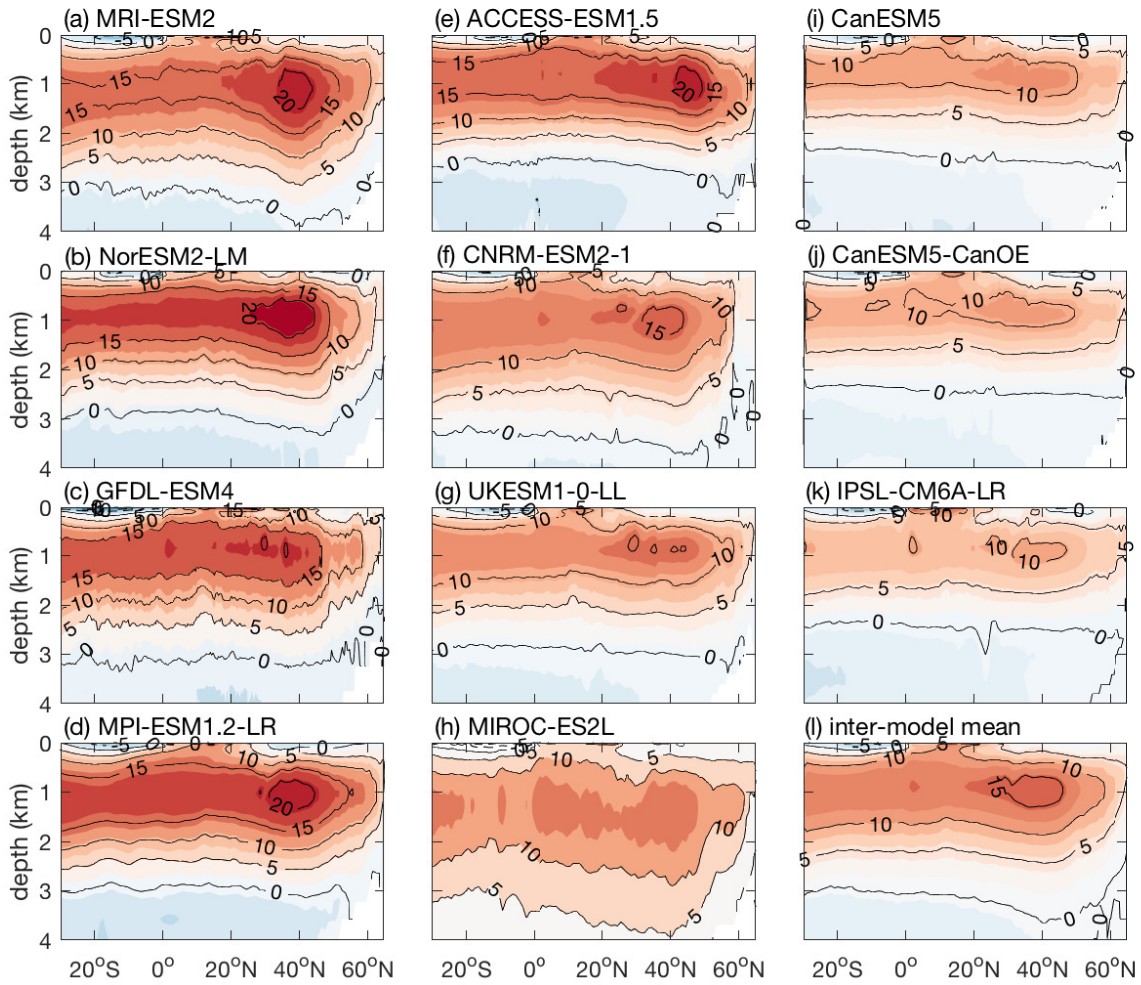

**Figure 8.** The Atlantic Meridional Overturning Circulation, AMOC ($Sv$), at the pre-industrial: (a)-(k) 11 CMIP6 Earth system models and (f) their inter-model mean. The estimates are based on pre-industrial control simulation (piControl) for years 121 to 140.

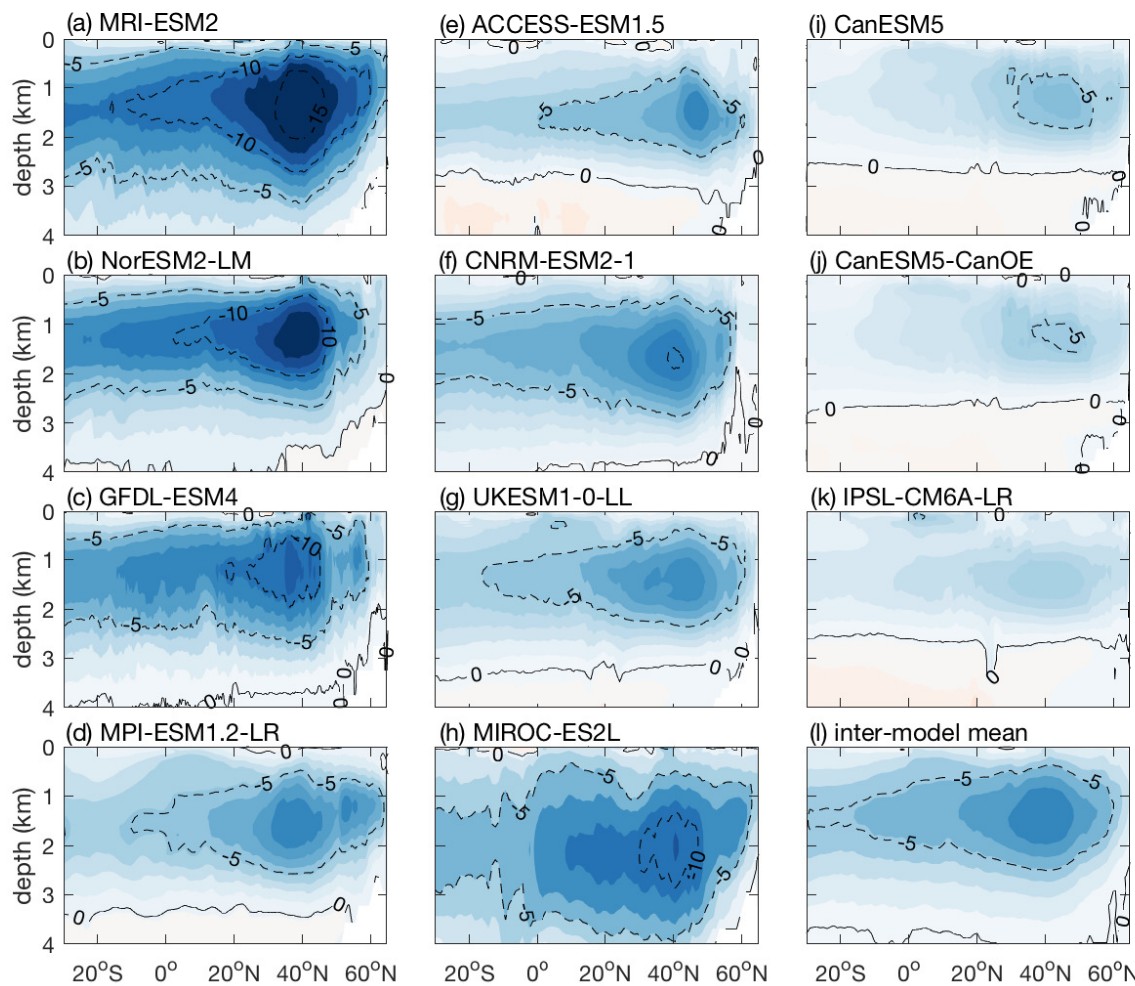

**Figure 9.** Weakening of the Atlantic Meridional Overturning Circulation, AMOC ($Sv$), relative to the pre-industrial: (a)-(k) 11 CMIP6 Earth system models and (f) their inter-model mean. The estimates are based on the fully coupled simulation (COU) under the 1% $yr^{-1}$ increasing $CO_2$ experiment minus the pre-industrial control simulation (piControl). Diagnostics are from years 121 to 140 (the 20 years up to quadrupling of atmospheric $CO_2$).

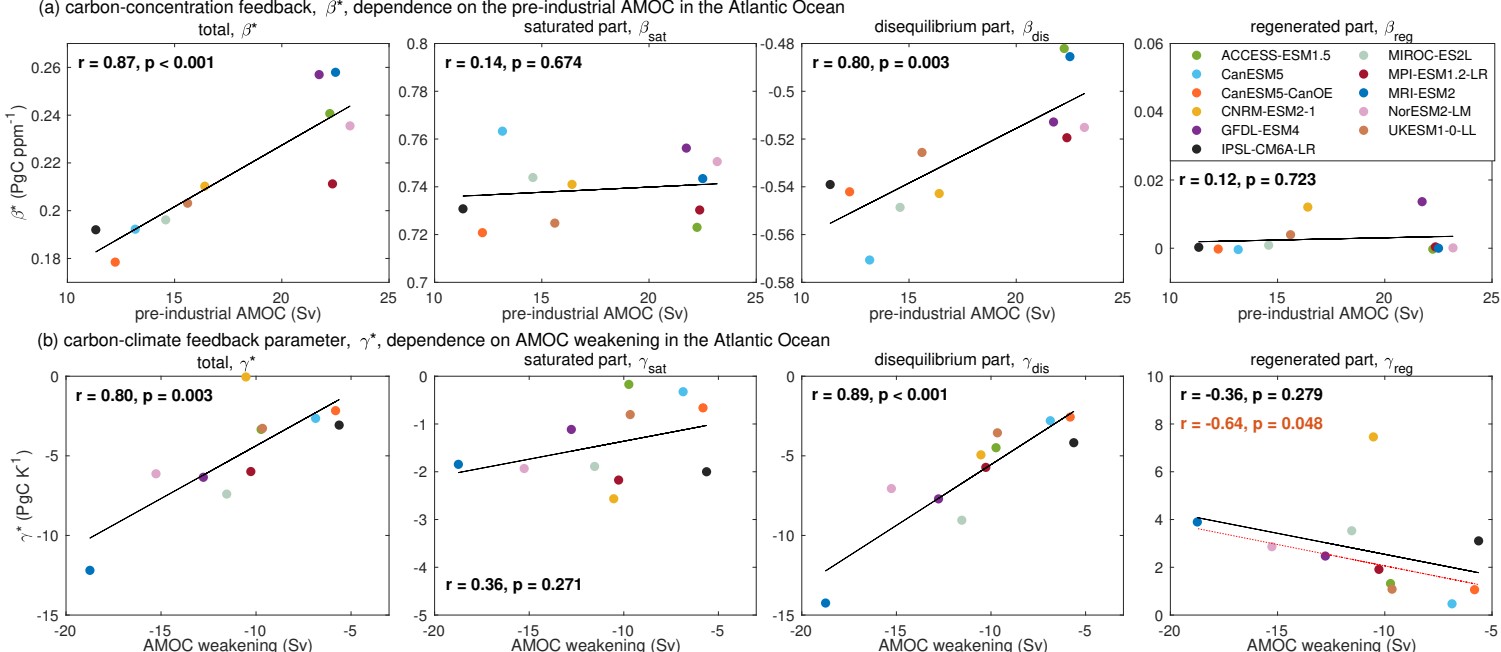

**Figure 10.** (a) Dependence of the carbon-concentration feedback parameter based on carbon storage, $\beta^*$ (PgC ppm$^{-1}$), on the pre-industrial strength of the Atlantic Meridional Overturning Circulation, AMOC ($Sv$), in the Atlantic Ocean; and (b) dependence of the carbon-climate feedback parameter based on carbon storage, $\gamma^*$ (PgC K$^{-1}$), on the AMOC weakening with climate change in the Atlantic Ocean in 11 CMIP6 models (Table 1), along with the contribution from the saturated, disequilibrium and regenerated carbon pools. The black lines correspond to the regression line based on an ordinary least squares regression, with the corresponding correlation coefficient, $r$, and $p$-value shown in each panel. The dashed red, line and $r$ and $p$-value in red color for $\gamma_{reg}$ correspond to the regression line and the correlation coefficient estimated by excluding CNRM-ESM2-1, the model with the largest deviation from the rest of the Earth system models in terms of the regenerated carbon pool. The strength and weakening of the AMOC are estimated as the maximum pre-industrial AMOC and maximum AMOC change between $30^o$N and $50^o$N, respectively (Table 3). Diagnostics are from years 121 to 140 (the 20 years up to quadrupling of atmospheric $CO_2$).

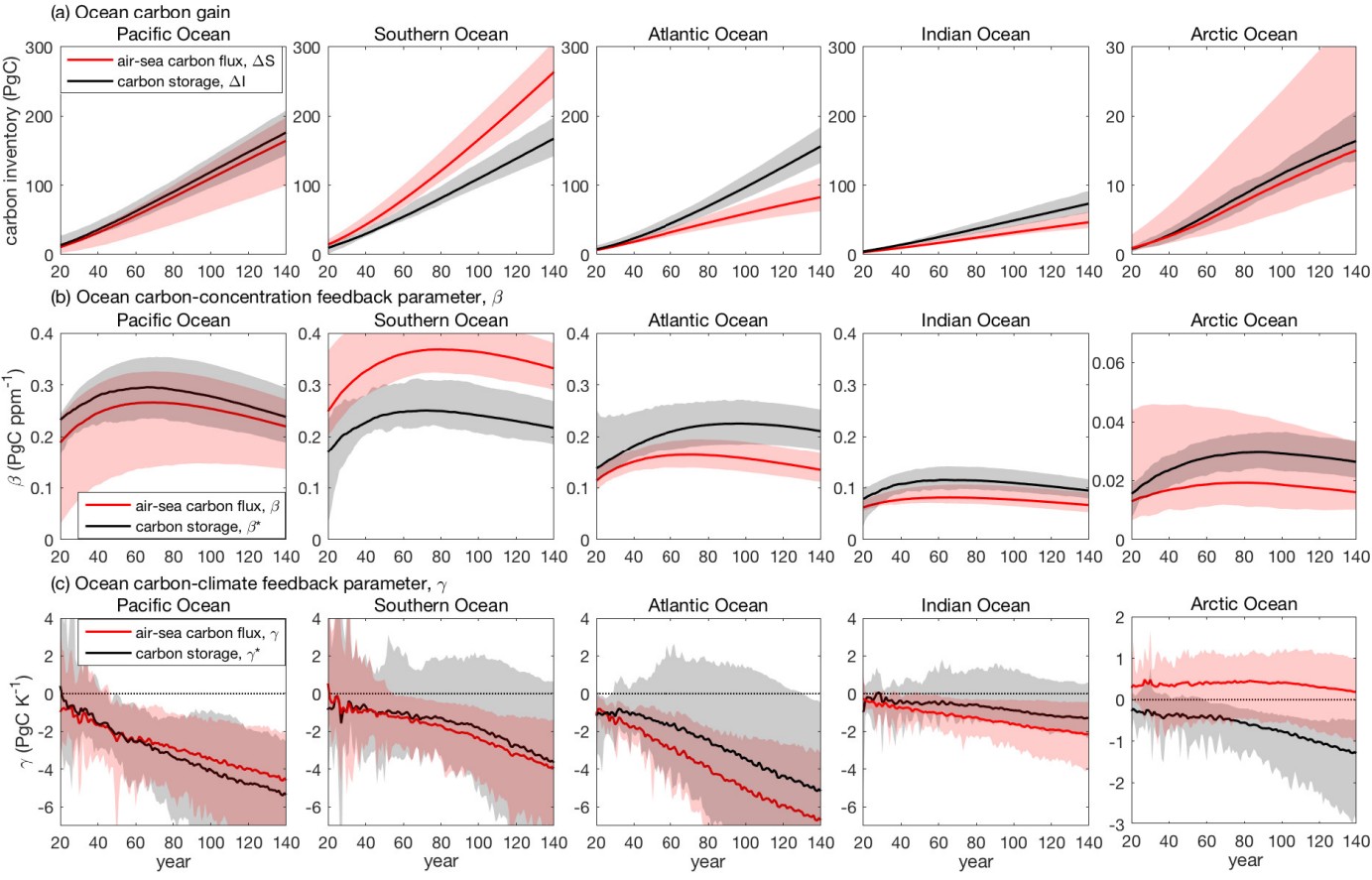

**Figure A1.** Ocean carbon storage and cumulative air-sea carbon flux in the Pacific, Southern, Atlantic, Indian and Arctic Oceans in CMIP6 Earth system models, along the ocean carbon cycle feedback parameters: (a) changes in the ocean carbon storage, $\Delta I$, and cumulative ocean carbon uptake, $\Delta S$, in PgC, in the fully coupled simulations (COU) relative to the pre-industrial; (b) carbon-concentration feedback parameter based on carbon storage, $\beta^*$ (PgC ppm$^{-1}$), and based on the cumulative carbon uptake, $\beta$ (PgC ppm$^{-1}$); and (c) carbon-climate feedback parameter based on carbon storage, $\gamma^*$ (PgC K$^{-1}$), and based on the cumulative carbon uptake, $\gamma$ (PgC K$^{-1}$). The solid lines show the model mean and the shading the model range based on the the 1 % yr$^{-1}$ increasing $CO_2$ experiment over 140 years in 11 CMIP6 models (Table 1). The estimates for the feedback parameters are based on the fully coupled simulation (COU) and the biogeochemically coupled simulation (BGC).

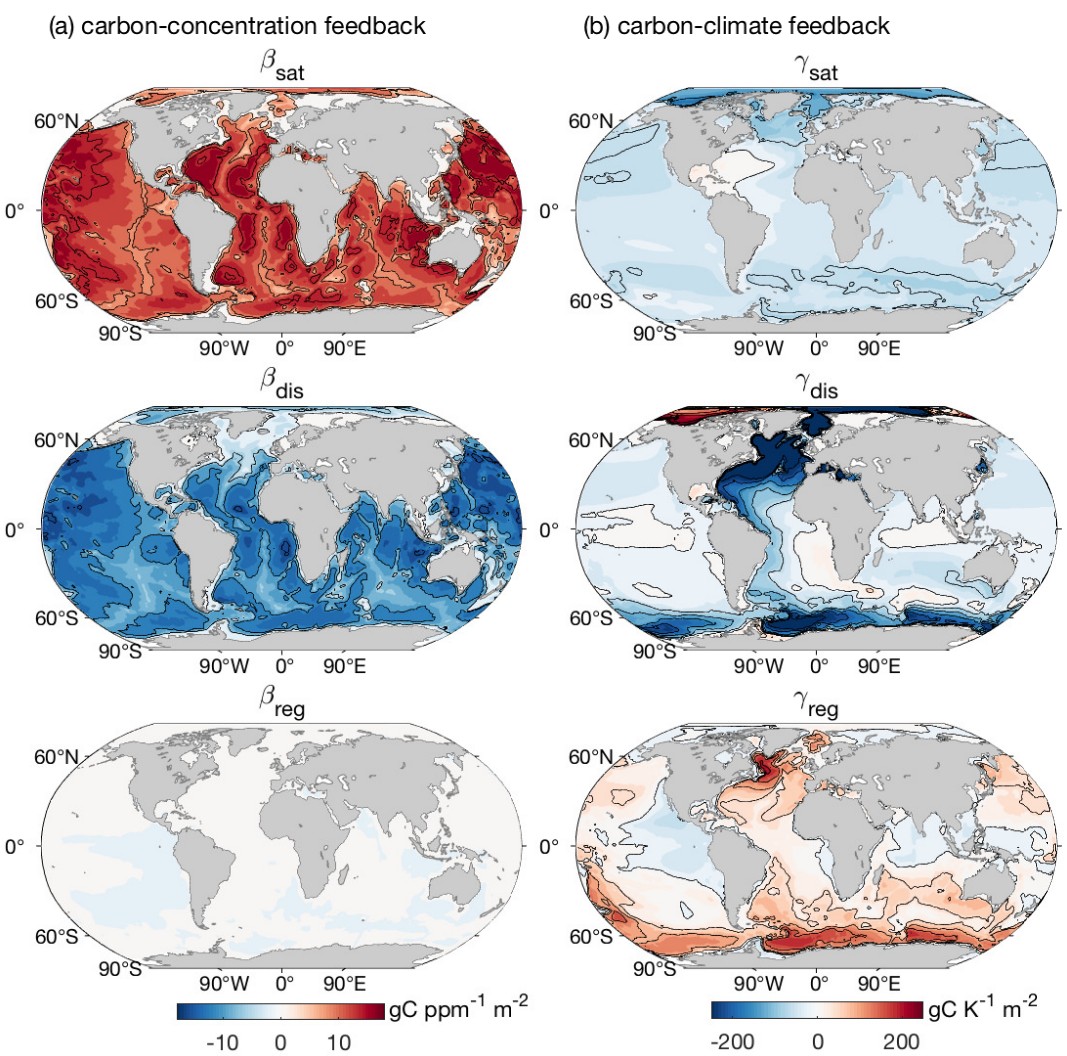

**Figure B1.** Geographical distribution of the CMIP6 inter-model mean contribution from the saturated, disequilibrium and regenerated carbon pools to the carbon cycle feedback parameters estimated based on carbon storage, $\beta^*$ and $\gamma^*$, normalised by area: (a) $\beta_{sat}$, $\beta_{dis}$ and $\beta_{reg}$ (gC ppm$^{-1}$ m$^{-2}$); and (b) $\gamma_{sat}$, $\gamma_{dis}$ and $\gamma_{reg}$ (gC K$^{-1}$ m$^{-2}$). The estimates are based on the fully coupled simulation (COU) and the biogeochemically coupled simulation (BGC) under the 1% yr$^{-1}$ increasing $CO_2$ experiment. Diagnostics are from years 121 to 140 (the 20 years up to quadrupling of atmospheric $CO_2$). The inter-model mean is based on the models in Table 1.