# Peer review of "Ocean carbon cycle feedbacks in CMIP6 models: contributions from different basins"

_Biogeosciences, 2020_

## Author Comment (AC1)

Reply to review by Jörg Schwinger

We thank Dr Jörg Schwinger for his constructive comments.

The authors present a detailed analysis of ocean carbon cycle feedbacks, which is a useful extension of the recently published global CMIP6 carbon cycle feedback paper (Arora et al. 2020). The manuscript goes beyond the Arora et al. study in that it focuses on the contribution of different ocean basins to the feedbacks and also explores circulation changes (using AMOC strength as a proxy) and their relation the feedbacks. The authors diagnose the basin wide contributions of preformed (saturated/disequilibrium) and regenerated carbon pools to the feedbacks. The description of the methodology is more detailed and contains additional diagnostics compared to the Arora et al. study. This manuscript is clearly within the scope of Biogeosciences, and I believe it will be of great interest to ocean carbon cycle community. The manuscript is generally well written (with some exceptions pointed out below) and I recommend it for publication in Biogeosciences after a few points detailed below have been addressed by the authors.

Thank you for your overall positive view.

Main points

1) Title: I don't think "controls of feedbacks from different ocean basins" is a good title. I am not a native speaker, but this sounds a bit odd to me. What the authors present is the "contribution of different ocean basins to carbon cycle feedback", and feedbacks are also attributed to different processes (including AMOC). As also noted further down, I don't think that the wording "control of AMOC on feedbacks in CMIP6 models" is appropriate. For CMIP6 models, the authors show a correlation between pre-industrial AMOC and AMOC weakening. "Control" implies a detailed mechanistic explanation, in my opinion. This is beyond the scope of this study, but therefore I would avoid using the word "control" here.

Agreed. The title will be changed to: 'Ocean carbon cycle feedbacks in CMIP6 models: contributions from different basins.'

2) The authors base their definition of feedbacks on changes in DIC-inventories, which, as they note, makes only very little difference at the global scale. However, at the regional scale the difference can be large (see Fig. 3), and therefore I suggest to use a different symbol for the feedbacks based on DIC-inventories. Since the feedbacks derived from carbon fluxes are the standard definition, I would use something like beta*/gamma* for the feedback estimate based on inventory changes. This would also simplify the discussion at the beginning of Section 3, where the two beta/gamma definitions are compared (the authors could then just write "beta*" instead of "beta, estimated from the regional ocean carbon storage").

Agreed, we will introduce new equations for estimates of gamma and beta from the cumulative flux along with new notation to distinct them from estimates using the carbon inventory.

Also, it seems the authors point out that the feedback definition based on inventory changes makes more sense than the "traditional" one based on accumulated fluxes (line 290-292: "...to gain more mechanistic insight, so as (i) to account explicitly for the ocean transport of carbon..."). Here I would disagree: From the feedback perspective, the flux at the air-sea interface (and changes to it) is the process we are interested in. Transport of carbon below the ocean surface leads to a disconnect between the actual feedback process at the surface, and where DIC-inventory changes are diagnosed (nicely illustrated by Fig. 3). Don't get me wrong here: I think the method the authors use is extremely useful to gain a global to large-scale

regional understanding of ocean carbon cycle feedbacks, but there is a price to pay. In lines 290-292 it sounds like the authors are selling this "price to pay" as an advantage. Maybe the authors can re-consider their wording here?

We will rewrite this part to clarify that the definition based on carbon inventory is different than the definition based on the air-sea flux and explicitly discuss strengths and limitations of both definitions.

3) Figure 3 and related discussion at the beginning of Section 3: This is one of the core figures of the manuscript, but it doesn't account for model uncertainty. I think it would strengthen the manuscript if the authors could expand this figure by 4 panels visualising the model spread (or standard deviation) for the 2 beta/gamma pairs, and add a brief discussion of where the main model uncertainty lies (and how and why this is different for the two definitions of regional feedbacks). Also for beta/gamma based on the DIC inventory, it would be great to split this further into the components (sat/diss/reg; this Figure could go into the Appendix).

Agreed. We will expand this figure to include the model uncertainty (please see Figure R1 below) and discuss this model uncertainty in the main text. We will also introduce a supplement figure that shows maps for the separation of beta and gamma to the different components (saturated, disequilibrium and regenerated) and relevant text.

4) Figure 5 and related discussion: This Figure seems to be flawed:

   -why does the total ocean volume add up to only 99%?

   -why do the different contributions to beta/gamma add up to different percentages (between 91 and 99%)?

In retrospect we recognise that not explicitly defining our ocean basins (related to comment 6 below) may lead to confusion. The volume for the combined Atlantic, Pacific, Indian and Southern Ocean does not add up to the total volume of the global ocean (100%) as 1.4% corresponds to the volume of the Arctic and semi-enclosed seas like the Mediterranean Sea.

The regional contribution to beta and gamma and their different components is not proportional to the regional volume, and each is controlled by different processes as discussed in section 3.2. Hence, the fractional contribution from the combined Atlantic, Pacific, Indian and Southern Oceans (excluding the Arctic and semi-enclosed basins) to beta and gamma are different to each other, and are different to their combined fractional volume. This distinction can be equivalently viewed in terms of the Arctic and semi-enclosed basins having different contributions to beta and gamma, which are also different from their fractional volume. This distinction is also the case for the saturated, disequilibrium and regenerated components of beta and gamma.

We recognise that the Arctic has a high contribution, and we will revise the manuscript to introduce a separate Arctic basin (and modify all figures, tables and text to include this additional basin). We will keep only the semi-enclosed seas in the 'other regions'.

[Figure]

Figure R1. Geographical distribution of the CMIP6 inter-model mean and standard deviation for the carbon cycle feedback parameters normalised by area, β and γ as estimated: (a) based on the regional ocean carbon inventory changes; and (b) based on the regional cumulative ocean carbon uptake from the atmosphere. The estimates now include the NorESM2-LM. Note that this figure will be updated to use different notation for beta and gamma estimated from the carbon inventory vs the cumulative carbon uptake as suggested by the reviewer.

5) The discussion in lines 332-337 is unclear to me: "In the well ventilated Atlantic Ocean, the additional heat penetrates into the ocean interior and is not confined to the ocean surface, which limits the effect of the reduction in solubility with warming". But the definition of gamma_sat

doesn't care whether a water parcel is at the surface or not. I don't think that this is can be the explanation (same in the next paragraph for the Pacific). Please clarify this

Agreed, the reviewer is correct. This point is associated with the non-linearity of gamma as described in equation (16) and discussed in lines 180-189 of the original manuscript. We will update the text in the new manuscript along the lines:

The Pacific and Indian Oceans have somewhat lower contribution to gamma_sat than expected from their fractional volumes as they experience less warming than the other basins (Figure 1.d). The Atlantic Ocean has a smaller contribution to gamma_sat than expected from its fractional volume despite experiencing the largest warming among the basins (Figure 1.d), which is related to the non-linearity of the carbonate system. Specifically, the Atlantic Ocean has the largest increase in DIC (Figure 1.c) which acts to significantly reduce the magnitude of the negative gamma_sat driven solely by the effect of warming on solubility (see equation 16).

[Figure]

Figure R2. Geographical definition for the different basins used in our analysis. Other regions include semi-enclosed seas that are not included in any of the main 5 ocean basins.

6) Nowhere in the manuscript it is stated how the ocean basins are defined. Where is the delineation between the Southern Ocean and the other basins? What about the Arctic Ocean? Is it included in the Atlantic or omitted? What about marginal seas? Also, the definition of AMOC strength is not given. Please add this information.

Agreed. We will provide bounds in terms of latitude and/or characteristic regional features (e.g., south edge of Africa) for the basin separation in the main text and add a figure in the supplement with a map showing our definition of the Atlantic, Pacific, Indian, Southern and Arctic Oceans (please see Figure R2 above). We will clarify that semi-enclosed seas like the Mediterranean Sea, the Red Sea and Hudson Bay are not included in the definition of the 5 main basins. The Arctic will be treated as separate basin in the new manuscript.

We will add the definition of the AMOC strength and weakening in the main text, which could only be found in the caption of Table 3 of the previous manuscript: The strength and weakening

of the AMOC are defined as the maximum pre-industrial AMOC and maximum AMOC change relative to the pre-industrial between 30oN and 50oN, respectively.

Minor points

7) lines 1-3: This sentence is complicated, and the wording isn't very precise (it is not a "competition between the increase in atmospheric CO2" but "a competition between the response to the increase in atmospheric CO2..."). Please consider rewording this sentence and maybe splitting it into two.

Agreed. We will re-write this text.

8) Equation 9: "Delta f" is not defined. Here, I would find it worthwhile writing the equation first in terms of DIC_sat, and state that DIC_sat=f(CO2,....). Then write down explicitly what Delta f means.

Delta f is an iterative algorithm, rather than a simple function, that estimates DIC_sat without prior knowledge of the [H+]_sat. This algorithm follows an iteration based on a first guess of [H+] or pH and is explicitly described in Follows et al., 2009 along with a coding example. In the limit that [H+]_sat is known then Delta f function becomes equivalent to equation (15) for gamma. We will reorganise and rewrite the text to better describe what Delta f represents.

9) Equation 10 and 11: Here, I think, it would be easier and shorter to just express beta and gamma in terms of I_sat (without writing out the f-terms explicitly - since I_sat is already defined in terms of f just a few lines above).

Agreed. We will re-organise these equations.

10) Equation 18 and 19: Here, I also think it is much easier to understand if beta and gamma are just expressed in terms of I_reg, which is defined just a few lines above in Eq. 17.

Agreed. We will reorganise these equations.

11) line 84-85: This is an assumption, not a conclusion, so starting the sentence with "Hence..." is not appropriate. Maybe use "In the carbon cycle feedback framework introduced by Friedlingstein et al. (2003,2006) it is assumed that..." or similar.

Agreed. We will change the text.

12) line 266: "...where gamma_n includes the non-linearity of ocean carbon cycle feedbacks". This is confusing, it sounds like the definition of gamma_n would be different from that of gamma, which is not the case. I suggest to delete this.

Agreed. We will reorganise this text and modify equations 22 and 23 following exactly the description of the global carbon cycle feedbacks in equations 4 and 5 to avoid any confusion.

13) Figure 3: To make this figure consistent with all other results, please add NorESM2-LM (fgco2 for the BGC run is available).

Agreed. We will add NorESM2-LM (please see Figure R1 above).

14) line 286: What do the authors mean by "asymmetries"? Please clarify.

Agreed, spatial asymmetries here refer to the regional differences/pattern in gamma (regions of positive and negative gamma) driven by the regional air-sea carbon exchange. The transport effect redistributes this regional ocean carbon loss/gain from the atmosphere due to changes in climate, such that the changes in gamma estimated based on the carbon storage are more uniform and less negative in the Southern Ocean, less negative in the North Atlantic high latitudes and switch to being negative in the Arctic relative to the gamma estimated based on

the cumulative carbon uptake. We will rewrite this text without using the word asymmetry for clarity.

15) line 323-324: "By definition, the contribution of each basin to beta_sat and beta_dis is approximately proportional to the ocean volume contained in each basin...". I see that this is the case for beta_sat, but for beta_dis this depends on ventilation which is not related to the volume. Maybe delete beta_dis here?

Agreed. We will delete beta_dis.

16) line 445-447: Please check and reword this sentence (consider splitting in two).

Agreed. We will re-write this text

17) line 476-477: "...which is mainly due to the disequilibrium carbon pool and the reduction in the physical ventilation with climate change." The second part of this sentence is a conclusion, isn't it? Then it would be more appropriate to write: "...which is mainly due to the disequilibrium carbon pool, indicating that the Atlantic has the strongest reduction in the physical ventilation with climate change."

Agreed. We will rephrase this text.

18) line 487: This is also seen in Schwinger et al. 2014

Thanks. We will reference Schwinger et al. 2014 here.

19) line 491-492: "The inter-model variability in gamma amongst CMIP6 models is relatively large compared with beta...". I think it is worth mentioning that this is not true in terms of the absolute feedback strength: In terms of PgC taken up by the ocean, it is still the uncertainty in beta that plays the dominant role.

Agreed. We will clarify that the uncertainty of gamma is larger than the uncertainty of beta in relative terms (variability in relation to the mean as described by the coefficient of variation) but that beta has more uncertainty in absolute terms for the carbon storage (variability in PgC).

20) line 513: "...controlled by the AMOC weakening..." As pointed out above, the authors find a correlation, so in my opinion the term "control" should be avoided here. Please consider rewording.

Agreed. We will only use control when we refer to results from the sensitivity experiment for the control of the AMOC with the idealised model. When we refer to CMIP6 and the results based on the correlation we will use something along the lines of 'dependence'.

Technical

21) lines 26-28: Please check the grammar and logic of this sentence.

Agreed. We will rephrase this text.

22) line 35: modes -> models

Agreed

23) line 43: "defined on" -> "defined based on"

We will rephrase this to 'defined in terms of …'

24) line 83: "..such as for example leading to..." please check grammar

Agreed, we will rephrase this text.

25) line 121: "at the surface" is confusing. Maybe better: "is the part of DIC that has been transferred from the surface into the ocean interior..."

Agreed

26) line 129: "is a unit conversion" please spell out from which to which unit.

Agreed, we will change to 'a unit conversion from moles to Pg of carbon'.

27) line 180: "The term inside the first {} brackets..." -> "The first term in curly brackets..."

Agreed.

28) line 199: "to the alkalinity" -> "to alkalinity"

Agreed

29) line 285 South -> Southern

Agreed

30) line 328: necessary -> necessarily

Agreed (I think the reviewer means in line 318).

31) line 349: "by the ocean ventilation" -> "by ocean ventilation"

Agreed

32) line 356: delete "now"

Agreed

33) line 372-373: "is the preindustrial" -> "denotes the preindustrial state" (or similar)

Agreed

34) line 470: "...of 26% to 30%..." -> "...between 26% and 30%..."

Agreed, we will rephrase to 'of between 26% and 30% …'.

---

## Author Comment (AC2)

Reply to anonymous reviewer

**General comments**

The paper is a valuable extension of the global analysis of carbon cycle feedbacks (Arora et al., 2020) and presents a useful framework that uses changes to the interior ocean carbon pools to help infer the first-order mechanisms contributing the magnitudes of regional and, thus, global ocean carbon cycle feedback parameters. The focus is primarily on how and why the strengths of the ocean carbon cycle feedback parameters vary at the basin-scale. The authors use diagnostics of the saturated, regenerated, disequilibrium carbon to help interpret these basin-scale variations. The also include a detailed mechanistic analysis of the relationship between AMOC on the feedback parameters. I particularly appreciated i) the thorough theoretical explanation for the evolution of the global ocean feedback parameters based on the contribution from the diagnosed saturated, disequilibrium, and regenerated carbon pools, and ii) the use of the box models to explore the response of these carbon pools to different AMOC scenarios.

We thank the reviewer for their positive view and constructive comments.

1) The global analysis of the evolution of the carbon cycle feedback parameters based on contributions from the different carbon pools (figure 2 with explanations rooted in fundamental chemistry)  is rigorous and informative, but it  would be easier to follow if this analysis were shifted and consolidated in a dedicated results section (details below).

Agreed, we will re-write/re-organise sections 2 and 3 (please see reply to comment 5 below).

2) I would have liked to see a basin-scale analysis similar to the global analysis of the evolution of the carbon cycle feedback parameters based on relative contributions from the different carbon pools. Why did you not use the same approach and figure presentation? Was it too difficult to interpret presented this way?

We agree and we will add this information for the different ocean basins (Atlantic, Pacific, Indian, Southern, and now Arctic ocean). Specifically, we will extend Figure 2 to include the 5 basins and split it into 2 figures: Figures R1 and R2 below. These figures and the evolution of the carbon cycle feedback parameters based on relative contributions from the different carbon pools for the global ocean and the different ocean basins will be presented and discussed in the updated section 3.2 (please see proposed structured in comment 5 below).

[Figure]

Figure R1. Ocean carbon inventory changes relative to the pre-industrial in the biogeochemically coupled simulation (BGC) (left panels) and ocean carbon-concentration feedback parameter, β (right panels), along with the contribution from the saturated, disequilibrium and regenerated carbon pools in CMIP6 Earth system models: (a) global, (b) Arctic, (c) Atlantic, (d) Indian, (e) Pacific and (d) Southern Oceans. The solid lines show the model mean and the shading the model range. Note this is an extension of Figure 2 in the original manuscript.

Figure R2. Ocean carbon inventory changes in the fully coupled simulation (COU) relative to the biogeochemically coupled simulation (BGC) (left panels) and ocean carbon-climate feedback parameter, γ (right panels), along with the contribution from the saturated, disequilibrium and regenerated carbon pools in CMIP6 Earth system models: (a) global, (b) Arctic, (c) Atlantic, (d) Indian, (e) Pacific and (d) Southern Oceans. The solid lines show the model mean and the shading the model range. Note this is an extension of Figure 2 in the original manuscript.

3) This paper focuses mostly on the regional feedback parameters calculated from the changes to carbon storage in the ocean interior. The ocean carbon cycle feedback parameters derived from air-sea CO2 fluxes are not analysed much here.  Globally the feedback parameters based on the air-sea flux and storage approaches are very similar, but—as the authors note—differences emerge at regional scales, largely because of the influence of transport. I think it is important to introduce the reader to the strengths and limitations of these complementary approaches. For example, basin-scale feedback parameter calculated using changes in the inventories of interior carbon pools,  aggregates many spatially and temporally varying processes over large ocean basins, which has it's limitations in terms of deciphering  the driving mechanisms since it is integrating  DIC changes that are driven by different and often compensating processes within the region and from outside the region itself  (particularly in the Southern Ocean). But, on the other hand, even if the exact sources of the changes are more difficult to pinpoint, the interior carbon pools provide an integrated picture of the first order controls on changes in carbon storage in the different basins. Furthermore, we can use the diagnosis of the various carbon pools to help interpret the mechanisms behind these changes.

We will add discussion about the differences, the strength and limitation calculating feedbacks from carbon storage vs carbon flux in the proposed new section 2.2. We will also introduce new equations and symbols for the regional beta and gamma estimated based on cumulative flux versus estimated based on carbon inventory in the new section 2.1. Please see reply to comment 5 below for the new sections.

4) Here the analysis of the processes (ocean carbonate chemistry, physical ventilation, and biological processes) driving changes in the carbon cycle feedback parameters are  diagnosed from changes to the saturated, disequilibrium and regenerated DIC pools.  So that the reader, can better evaluate the results presented from this part of the study an honest appraisal of the strengths and limitations of this approach would also be a valuable addition.

i) For example, interpreting changes in the distributions of the different carbon pools is not always straightforward as you might hope (see spatial complexity in Arora et al., 2020). Nevertheless,  basin-scale changes in carbon storage provide a natural integration of  a spatially complex air-sea CO2 flux pattern.

We agree with this point and this complexity is reflected in the regional maps for beta and gamma (Figure 3). We will introduce and discuss a new supplement figure with regional maps for the saturated, disequilibrium and regenerated component of beta and gamma in the spirit of Figure 3.

ii) Even using this approach it is not possible to completely separate the various processes. For example,  the regenerated component combines both biological  and ventilation changes because the strength of the biological pump is also dependent on the ventilation.   It seems that describing the changes in terms of their impacts on the different carbon pumps could be helpful.

We agree that the regenerated component depends on changes in the ventilation as we describe in lines 214-217 and lines 253-255, and this dependence leads to a significant correlation between gamma_regenerated and AMOC in the Atlantic Ocean as described in lines 438-440. We incorporate this effect of the changes in the ventilation on the biological carbon pool to biological processes (lines 214-217 and 253-255), since in the absence of biology this effect would be zero. We will add text to clarify that the regenerated component corresponds to the biological pump and the preformed component to the solubility pump. We then go on to split

the preformed component into two idealised pools: a saturated and a disequilibrium pool, such that we split the global solubility pump into a chemical and physical component in terms of the carbon uptake and transfer. We prefer to discuss the changes in terms of physical, chemical, and biological processes rather than in terms of biological and solubility pump, but we will clarify that the biological processes include the effect of physical ventilation to the biological carbon pool.

iii) Basin-scale variations in the change in carbon storage can't be interpreted directly as the importance of the region to the global carbon cycle feedbacks, because the source (and mechanism) of change could have been from outside the region.

The regional carbon cycle feedbacks estimated based on regional carbon inventory includes the effect from the source and the effect from the transport on carbon storage, as illustrated in Figure 3, Figure A1 and discussed in lines 270-289.  We will extend the discussion in lines 270-289 to clarify this issue (please see reply to comment 3). Even for the feedbacks defined by the air-sea fluxes, separating the local from the far-field control is still problematic as the local air sea fluxes include influence from transport of carbon or other tracers like temperature and nutrients from outside the region. Here, we are using a framework to provide insight on the basin-length effect of different processes (biology, chemistry, physical ventilation) on the carbon cycle feedbacks no matter if those processes are of local or remote origin. As we briefly discuss in lines 527-530, but we will further extend in the revised manuscript, estimates of the contribution from the different carbon pools in a dynamically based density space rather the geographical space may provide further insight.

5) My main recommendation is to reorganise several sections of the paper to help clarify the methodology and distil some of the main results—in particular for the analysis of the contribution of the various carbon pools to the regional carbon cycle feedback parameters (detailed suggestions are given below).

We agree with the recommendation to restructure part of the manuscript. Following the reviewer's comments, we plan to re-organise the paper in the following manner:

1. **Introduction**
2. **Regional ocean carbon cycle feedbacks analysis in CMIP6 models**
   **2.1 Methodology**
   Derivation of global and regional beta and gamma (equations 1-5, equations 22-23, new equations for the distinction between feedback estimates from air-sea fluxes vs ocean carbon inventory and relevant text)
   **2.2  Regional beta and gamma in CMIP6 models: estimates based on carbon storage vs carbon uptake**
   Differences due to carbon transport: maps in Figure 3, new supplement figure showing explicitly the effect of transport (see Figure R3), expanded discussion in lines 272-293 of the original manuscript, strength/limitation of the two approaches. Refer to Appendix A for the net effect of carbon transport to the carbon storage on the different ocean basins.
3. **Processes controlling the ocean carbon cycle feedbacks in CMIP6 models**
   **3.1 Methodology**
   Derivation of the contribution from different carbon pools on the carbon cycle feedbacks (equations 6-11, equations 17-21).

**3.2 Contribution from different carbon pools to global and regional beta and gamma**

**3.2.1 Insight from theory**

Material/text linked to equations 12-16 and Figure 2 but now modified to include results for the global ocean and the different basins (please see response to comment 2 and Figures R1 and R2).

**3.2.2 Quantitative contribution from the different carbon pools**

Material/text on the relative contribution from the different carbon pools to the feedbacks in terms of magnitude and uncertainty for a quadrupling of atmospheric CO2 (Figure 4 and Table 2). A new supplement figure with regional maps for the saturated, regenerated and disequilibrium component in the spirit of Figure 3.

**3.3 Contribution from different ocean basins to the global ocean beta and gamma**

Material associated with Figure 5 and subsection 3.2 of the original manuscript.

**4. Dependence of the carbon cycle feedbacks on the Atlantic Meridional Overturning circulation**

**4.1 Insight from an idealised climate model with a meridional overturning**

Material from section 4.1 of the original manuscript

**4.2 Results from CMIP6 models**

Material from section 4.2. of the original manuscript

**5. Discussion and Summary**

**5.1 Regional ocean carbon cycle feedbacks**

Material from section 5.1 of the original manuscript

**5.2 Effect of the Atlantic Meridional Overturning Circulation**

Material from section 5.2 of the original manuscript

**Appendix A: Effect of ocean transport on the carbon storage in different basins** (Figure A1 and material from Appendix A of the original manuscript)

Our new structure draws on the reviewer's recommendations, but sections 2 and 3 are separated in terms of themes rather than in terms of methods and results, which we think is more intuitive and easier to follow.

6) It is difficult to see the relative contribution of saturated, disequilibrium and regenerated carbon pools to the basin-scale feedback parameters. I would suggest adding (or modifying) a figure dedicated to the relative contributions of saturated, disequilibrium and regenerated carbon pools to the basin-scale feedback parameters. Although, Figure 4 contains this information, it can be difficult to read this information off the figure.

We will add and discuss Figures R1 and R2 to complement Figure 4 (please see reply to comment 2 above).

7) Related to the last comment, the author often presents the volume integrated quantities, which makes it more difficult to appreciate:

i) where a feedback is stronger or weaker than expected based on volume alone (Figure 5 shows this well, but sometimes it is lost in the text),

Agreed, we will re-write/re-organise section 3 and text related to Figure 5.

and ii) what processes (diagnosed using the changes to the carbon pools) dominate the magnitudes of each carbon cycle feedback parameter in each basin (not always easy to see in the Figures and tables).

We will add and discuss Figures R1 and R2 (please see reply to comment 2 above) to complement Figure 4.

8) The authors suggest they will account for the impact of carbon transport. This influence is transport is not presented in much detail, and would require a more thorough comparison of regional air-sea CO2 flux and storage feedback parameters. Therefore I would either i) elaborate on this in a dedicated section or ii) simply remove and include some reference to the impact of transport in the methodology where the relative merits of flux and storage approaches are discussed.

The direct influence of the carbon transport to the feedbacks as defined by the carbon storage is captured by the difference between the feedbacks estimated from the cumulative carbon fluxes (Figure 3.b) and the carbon inventory (Figure 3.a), and by the deficit between the black and red lines in Figure A1. We will add a supplement figure (see Figure R3 below) that explicitly shows the difference between panels a and b in Figure 3 and further discuss the regional distribution of this effect of the carbon transport on the carbon storage. We will also clarify that here we discuss the effect of the carbon transport to the carbon storage, but we do not separately estimate and isolate the effect of the transport of carbon and other tracers (e.g., temperature, salinity, nutrients) on the regional source of carbon itself, which will need further analysis.

[Figure]

Figure R3: Geographical distribution of the effect of carbon transport on the beta and gamma based on carbon storage, as estimated from the difference between the CMIP6 intermodel mean regional ocean carbon inventory changes and the CMIP6 intermodal mean regional cumulative ocean carbon uptake from the atmosphere.

9) Several time the authors comment on the beta having less uncertainty (intermodel variability) than the gamma. This is misleading, because the uncertainty in each is mapped disproportionately onto the uncertainty in carbon storage (i.e. gamma is multipled by only  > 4 deg C , while beta is multiplied by the   > 700 ppm change in atmospheric CO2) .

We will clarify that the uncertainty of gamma is larger than the uncertainty of beta in relative terms (variability in relation to the mean as described by the coefficient of variation) but that beta has more uncertainty in absolute terms for the carbon storage (variability in PgC).

**Specific comments**
**Abstract**

10) On beta: "The Atlantic, Pacific and Southern Oceans contribute equally to the carbon-concentration feedback, despite their different size." I think a better emphasis is that the Atlantic storage increase more and Pacific storage increases less than expected in relation to their size. Then mention why. Also the Southern Ocean beta is low despite the strong air-sea CO2 flux here, because it transported out of the region. what about the Indian? Summarise on the controls on beta for each basin

Agreed, we will modify the abstract.

11) On gamma: Similarly, I would summarise the dominant controls for each basin, merging the conclusions from the carbon pools and AMOC parts of the analysis.

We will modify the abstract to summarise dominant controls for each basin. We prefer to keep the AMOC part of the analysis separate rather than merging it with the controls for each basin in the abstract.

**Introduction**

12) lines 25 –28: Given the central importance of these feedbacks in this papers, this introduction to carbon cycle feedback parameters could be a little more comprehensive. It also a bit awkward to read. Perhaps merge the name and description of each feedback. And then finish with "These two carbon cycle feedbacks have been extensively used to...."

Agreed, we will re-write this part.

13) lines 74–79: rework section description to reflect any changes made to the sections.

 Agreed, we will re-write to reflect the changes to the sections (please see reply to comment 5 for the proposed new sections).

**Ocean carbon cycle feedbacks and their control by different processes**

14) The beginning of this section reads like part of the introduction.  I think it should be clear to the reader from the onset that this is where you will present the approach used in this paper.

I would start this section introducing your methodology and how the methodology compares to previous studies, particularly Arora et al., 2020.  Something along the lines of  "here we extend the analysis of the carbon cycle feedbacks diagnosed in the CMIP6 models (Arora et al., 2020)…." and then introduce what part of the analysis is identical and what parts have been modified or extended.

The beginning of the section will be modified to reflect changes/re-organisation of the section (please see response to comment 5 above). We will also clarify that our methodology is a regional extension of the methodology for the effect of different carbon pools on the carbon cycle feedbacks introduced in Williams et al., 2019 and used in Arora et al., 2020 for CMIP6.

15) I recommend separating the methodology into two parts: 2.1  Ocean carbon cycle feedback analysis, 2.2 Diagnostics of processes controlling carbon cycle feedbacks.

We will re-organise section 2 (please see reply to comment 5 above).

16) Move the results and discussion of the contribution of the global saturated, disequilibrium, and regenerated carbon pools (and the associated Figure 2)  to the global carbon cycle

feedback parameters to section 3.  It is very interesting  and thorough and deserves a dedicated results section.

Agreed, we will move this discussion in the new section 3 (please see reply to comment 5).

17)  For improved clarity, the ocean cycle carbon cycle feedbacks (e.g. recommended Section 2.1) should consolidate all the feedback methodology  from other sections and include some missing elements:

We will re-organise section 2 (please see reply to comment 5 above)

(i) Clearly present and give distinguish between the  feedback parameters and the carbon inventories calculated from the  i) air-sea CO2 fluxes and ii) interior carbon inventories. In regional analyses this distinction becomes very important (as the authors explain). For example, you could use 'betaf'  and 'If' and 'betas'  and 'Is' for beta and carbon inventories calculated from carbon  fluxes  and carbon storage respectively.

Agreed, we will introduce terms and equations to distinguish the feedbacks estimated based on cumulative air sea flux and based on the carbon storage in the new section 2.1.

(ii) Please include a discussion of the merits/limitations of the flux vs storage approaches and their complementarity.

Agreed, we will discuss limitation and strength of the two approaches, particularly in terms of regional feedbacks in the new section 2.2 (please see response to comments 3 and 4.iii above).

(iii) Move the regional calculation of the feedback parameters (263–274) to this section.

Agreed.

(iv) The basins have not been defined. Please provide the precise boundaries used for the basin-scale analysis show boundaries on the maps in Figure 3.

Agreed, we will provide bounds in terms of latitude and/or characteristic regional features (e.g., south edge of Africa) for the basin separation in the main text and include a map that shows this separation in a supplement figure (see Figure R4 below). Note that we will also update the manuscript (figures, tables, and text) to include a separate Arctic ocean basin.

[Figure]

Figure R4. Geographical definition for the different basins used in our analysis. Other regions include semi-enclosed seas that are not included in any of the 5 main ocean basins.

18) For the section (e.g. recommended section 2.2) presenting the methodology for diagnosing the ocean processes contributing the carbon cycle feedbacks ( based on calculations of saturated, disequilibrium, regenerated carbon pools)

(i) Start this section with something like your line 115. "To gain insight …"

(ii) Include detail here on the how the various carbon pools are used to diagnose ocean processes (ventilation, biology…) and the limitations of this approach.

(iii) I assume methodology used to calculate the carbon pools is identical to that used in Arora et al., 2020? Please elaborate. If so, it would be useful to inform the reader.

(iv) Include here only the methodology used to calculate the changes in the carbon pools and leave the theoretical explanations that are used to better understand their evolution to section 3.

We will restructure sections 2 and 3 (please see response to comment 5 above). The methodology for separating the contribution into different carbon pools follows the methodology of Williams et al., (2019) that was then used in CMIP6 analysis in Arora et al., (2020) (lines 115-117 of the original manuscript) (please see reply to comment 14 above).

19) line 81–84: this essentially repeats introductory section above (lines 25–28 ) and doesn't need to be repeated here.

We disagree. Repeating this text in the methodology summarises the fundamental view of the carbon cycle feedback framework and is helpful for readers not familiar with this framework

20) line 106: this does not "exclude" all model biases because the initial biases continue to impact their evolution, but it does help reduce them. Either replace "To exclude" by "To reduce" or "To partially exclude".

 Agreed, we will revise the text to 'reduce'.

21) line 251: replace "suggesting" with "indicating"

 Agreed.

22) line 253: replace "act to reduce" with "reduce"

 Agreed.

**Regional carbon cycle feedbacks in CMIP6 Earth system models**

23) To make the paper easier to follow, this section should consolidate the global and basin-scale analyses of the contribution of the carbon pools to the carbon cycle feedback parameters

Agreed, we will re-write and re-organise section 3. Please see reply to comment 5 above for the new structure and Figures R1 and R2.

24) You could rename the title of this section "3 Processes controlling the carbon cycle feedback parameters" and subsections "3.1 Global analysis" and "3.2 Basin-scale analysis". Maybe there is no need for subheadings.

We plan to restructure section 3. Please see reply to comment 5 above for our new structure and title of sections and subsections.

25) The section starting on line 275 touches on the impact of carbon transport on the feedback parameters and on line 291 mentions that the study will account explicitly for the ocean

transport of carbon. Since there there is little discussion of the impact of transport on the feedbacks in the paper and the main results are added to an Appendix, these paragraphs seems out of place here. Two suggestions:

(i) Either include a more detailed examination of the impact of transport on regional betas and gammas in a dedicated results& discussion section including the figures in the Appendix. This would be interesting if it extends our understanding beyond what is presented in Frolicher et al., (2015).

(ii) Or, the spatial distribution of the ocean carbon-cycle feedbacks in figure 3 could be moved into and discussed in the methodology section where the air-sea $CO_2$ flux and carbon storage feedback parameters are introduced and compared. Here the figures could be discussed in terms of the complementarity of the flux and storage approaches.

We will move this part of the study into the new section 2.2 and include discussion about the differences, the strength and limitation calculating feedbacks from carbon storage vs carbon flux. Please see reply to comment 5 (for the proposed new structure), and reply to comments 3 and 8, and Figure R3. We still prefer to keep Figure A1 and discuss it in an appendix such that this information is available to the reader but the new section 2.2. is kept sharp.

26) There are some places—particularly in the interpretation of the changes in the global and regional carbon pools and their contributions to the feedback parameters—where the language used to associate the changes to mechanisms, reads as if these are mechanisms have been determined rather than diagnosed. I think care needs to be taken to word the results and conclusions in light of an honest appraisal of the confidence we have in the mechanisms diagnosed using this approach (probably in the methodology).

Agreed, we will update the text to reflect that the mechanisms are diagnosed rather than determined.

27) lines 263 –271: As mentioned earlier, move this methodological detail to section 2.

Agreed, please sea reply to comment 5 above for the updated structure of the manuscript

**Global analysis**

28) For the Global analysis I would recommend moving the global analysis of the contributions of the various ocean carbon pools to the carbon cycle feedback parameters (I.e. associated with Figure 2) into a dedicated section, which includes the following text: the presentation of the theoretical analysis that helps us interpret the evolution of the global feedback parameters: saturated component (lines 148–196), regenerated (lines 208 – 217), and disequilibrium (lines 225–238 starting with the rise in atmospheric CO2….).

Agreed, subsection 3.2.1 will be dedicated to this insight from the theoretical analysis (please see reply to comment 5 for the updated structure).

**Basin-scale analysis**

29) I feel this section of the paper needs some polishing. The figures and the discussion don't always clearly separate the impact of the strength of the feedback (i.e. volume-normalised) from the volume-integrated impact. Consequently, it can be difficult to clearly see the relative contributions of the saturated, disequilibrium and regenerated carbon pools to the basin-scale parameters. This was done well for the global analysis.

We will re-organise section 3 and the text associated with the basin scale analysis (please see reply to comment 5 for the updated structure). We will also extend Figure 2 into two figures and include information for the basin-length analysis (please see reply to comment 2 above and Figures R1 and R2).

30) I find the separation between section 3.1 and 3.2 confusing. Only a short discussion of the contribution of different basins to global carbon cycle feedback parameters is needed here (the main results are already in Figure 1). The main focus should be i) the relative strengths of the carbon cycle feedbacks relative to their volumes and ii) on the analysis of the mechanisms controlling the differences between the basin-scale carbon cycle feedback parameters (volume-normalised) by accounting for the contributions from the different carbon pools.

We will re-organise this section (please see reply to comment 5 for the updated structure).

31) I would start section on the relative strengths of the feedbacks by discussing Figure 5, which is better suited for this than figure 4. For example, for beta, It is easy to see that the Atlantic takes up more and the Pacific less relative to their volumes

In the new manuscript (see reply to comment 5 for the new structure) Figure 5 will be discussed in terms of contribution of each of the basins to the global carbon cycle feedbacks. Figure 4 along with Figures R1 and R2 (please see reply to comment 2 above) will be discussed in terms of contribution of the different carbon pools to the basin-length feedbacks:

(i)     the opposing or complementary effect of the saturated, disequilibrium and regenerated part on the basin-length carbon cycle feedbacks.

(ii)    the magnitude and uncertainty of the contribution of the different carbon pools to the basin-length feedbacks in the CMIP6 models,

32) A figure is missing that is dedicated to the relative contributions of saturated, disequilibrium and regenerated carbon pools to the basin-scale feedback parameters. For example, you want to be able to easily see what mechanisms is responsible for the Atlantic taking up more and Pacific taking up less carbon relative to it's volume or what process dominates the carbon-climate feedback in each basin. Although, Figure 4 contains this information, since the feedbacks parameters are i) not volume-normalised and ii) the components of the feedback parameter for each region are not presented side-by-side, it can be awkward to read this information form the figure. Maybe Figure 4 could be reworked by :

We will introduce Figures R1 and R2 showing the contribution of the different carbon pools to the basin-length feedbacks as described in the reply to comment 2 above, to complement Figure 4.

(i) either grouping the contributions from each carbon pool into a plot for each basin rather than for each component.

The grouping in terms of basins rather than carbon pool components does not work well for beta in Figure 4 that focus on magnitude and range, as the total, the saturated, the disequilibrium and regenerated are different orders of magnitude (see Figures 4 and R1). However, we now introduce figures R1 and R2 that in our opinion show well the contribution from each carbon pool for each basin.

(ii) or presenting the contribution of each component to the basin-scale feedback parameters as percentage contributions, so that it is easy to clearly see how the relative contributions of the different components to the total feedback vary between the regions.

The contribution for the saturated, disequilibrium, and regenerated components to each basin or to the global ocean cannot be meaningfully presented as percentage as these different components have a positive or negative contribution (opposing each other, see Figures 4 and R2).

In contrast, the contribution of each basin to the different components (saturated, disequilibrium and regenerated) can be presented as percentages, since on average the different components are of the same sign for all the basins and for the global ocean: e.g., the contributions of the regenerated component to gamma is on average positive for all basins for a quadrupling of atmospheric CO2, while the contribution of the saturated and disequilibrium components to gamma is on average negative for all basins for a quadrupling of atmospheric CO2 (see Figures 4, R1 and R2).

33) It would be informative to see the geographical distributions of the saturated, disequilibrium, and regenerated components to help with the interpretation of the basin-scale changes and to complement the depth sections presented in Arora et al., (2020).

Agreed, we will add a supplement figure with the maps/geographical distribution for the saturated, disequilibrium and regenerated component and relevant discussion in the new section 3.2 (please see reply to comment 5 for the new structure).

**Discussion and Summary**

34) line 163: In "The transport effect acts to decrease the carbon-concentration feedback parameter" it is important to mention increase relative to what. That is, the carbon-concentration parameter calculated from the air-sea CO2 fluxes.

Agreed, we will clarify following the reviewer's suggestion.

35) line 473: again the intermodel variability in beta may seem small, but the impact of this intermodel variability on carbon storage is not.

Agreed, please see reply to comment 9 above.

36) line 479: "consistent with previous studies" here mention consistent with the spatial patterns of gamma diagnosed from the air-sea CO2 fluxes in the last two generations of ESMs (i.e. spatial patterns of the feedback parameters in Roy et al., 2011 and in the CMIP5 models in the IPCC WG1 assessment report Ciais et al., 2013).

Agreed, we will reference these studies here.

37) line 490: It would be useful to reference consistency with previous analyses of the drivers of the carbon-concentration parameter distributions here even if they were based on the CO2 air-sea flux carbon cycle feedbacks "as was shown in a CMIP5-generation model…. and is consistent with analyses of the carbon-climate feedback distributions from previous generation models (Roy et al., and Ciais et al., 2013)"

Agreed, we will reference these studies here.

**Figures**

38) Figure 4: Does not include the multimodel mean (or median). I think it would be most informative to overlay the median and IQR.

Agreed, we will add the mean (rather than the median to be consistent with the estimates in our tables), and the interquartile range in Figure 4.

39) Figure 5: It looks like some of the parts of the ocean have not been included here. I suppose there is meant to be an 'Other regions' category (inland seas and arctic?) to make each column sum up to 100%. Please explain in the caption.

Yes, this is meant to be other regions including the Arctic.

Why were these other regions left out? They contribute a substantial amount to global gamma.

The Arctic has a high contribution, and we will revise the manuscript to now introduce a separate Arctic ocean basin. We will keep only the semi-enclosed seas like the Mediterranean Sea, Hudson Bay and Red Sea in the 'other regions'.

**40) ## Technical comments**

There are a few places where the English needs a little work. I have picked up a bunch of them, but someone should give the document another once over.

Agreed.

**Abstract**

line 4: Change ". The contribution from different ocean basins to the carbon cycle feedbacks and its control by the ocean carbonate chemistry, physical ventilation and biological processes is explored in diagnostics of 10 CMIP6 Earth …." to ". The contribution from different ocean basins to the carbon cycle feedbacks and the processes that control them are explored using diagnostics of ocean carbonate chemistry, physical ventilation and biological processes in10 CMIP6 Earth …."

We will re-write this text.

line 5: Change "mechanist" to "mechanistic"

Agreed.

line 15: change "on global scale" to "at the global scale"

Agreed.

**Introduction**

Line 21: change "refer" to "referred"

Agreed

Line 22: change "lapse rate" to "tropospheric lapse rate"

Agreed

Line 24: change "land and ocean" to "land and ocean reservoirs".

Agreed

Line 34: change "feedback is about 3 times stronger over land than the ocean on centennial " to "feedback is about three times stronger over ::the:: land than the ocean on centennial "

Agreed

Line 55: I would remove "hence" because it implies you are going to focus on both heat and carbon in this study.

Agreed

line 56: Replace "Hence, our motive is to explore the mechanisms that lead to this regional variation in the carbon storage and the carbon cycle feedbacks for the different ocean basins…" with "Our ::motivation:: is to explore the mechanisms that lead ::these:: regional variations in the carbon storage and the carbon cycle feedbacks ::in:: the different ocean basins…"

Agreed

line 71: changes "insight for" to "insight into"

Agreed

line 72: This sounds odd. Please replace "Our aim is to provide insight for the relative contribution from different ocean basins to the ocean carbon cycle feedbacks, and the processes that drive this relative contribution and its uncertainty amongst CMIP6 " by "Our aim is to provide insight ::into:: the relative contribution  of different ocean basins to the ocean carbon cycle feedbacks  and the processes that drive this regional partitioning  in the CMIP6 models".

Agreed, we will re-write this text.

line 73: it reads as if you will exploring the controls of the AMOC. Please replace by something like "the control of the AMOC on the carbon cycle feedbacks".

Agreed, we will re-write this text.

Line 76: replace "processes" by "diagnostics of processes".

 Agreed.

**Ocean carbon cycle feedbacks and their control by different processes**

Line 81: comma between "CO2 which"

Agreed.

Line 82: change"At the same time the increase in atmospheric CO2  modifies the physical climate system, such as for example leading to ocean  warming and increase in stratification" change to "At the same time the increase in atmospheric CO2  modifies the physical climate system,  leading to changes such as ocean warming and increased stratification…. "

Agreed.

Line 84: change "ocean carbon uptake" "change in ocean carbon uptake"

We will rephrase to 'the ocean carbon gain due to anthropogenic carbon emissions …'.

line 110: you don't need the subscript ocean for Equation 5. I would save the subscript position for discriminating between feedback parameters calculated using air-sea CO2 fluxes vs carbon storage.

We prefer to keep the subscript ocean for clarity as we refer to ocean carbon cycle feedback parameters rather than including the contribution from land. We will however introduce notation to separate the calculations based on air-sea CO2 vs carbon storage.

line 111: choose to either capitalise or not the word "earth" in the document.

Agreed, we will use Earth consistently.

Line 115: "To gain insight for the driving mechanisms of the carbon cycle feedbacks and their uncertainty " "To gain insight ::into:: the driving mechanisms of the carbon cycle feedbacks and their uncertainty "

Agreed

line 125: replace "extend" by "extent".

Agreed

line 125: replace "contemporary CO2" with "contemporary CO2 concentration".

Rather than use contemporary that has led to confusion, we will explicitly define DICsat and DICdis in terms of the departure from a full chemical disequilibrium with the given atmospheric CO2 at the same time.

Line 126: include symbol and units for carbon inventory (I.e. deltaI, PgC) for consistency with unit conversion listed below. (Odd to have unit conversion when no units have been listed as yet).

Agreed

line 127: for consistency, include units of DIC pools.

Agreed

line 128:  Again you don't need the subscript ocean here. Save it for inventory calculated using carbon storage.

We prefer to keep the subscript ocean but we will introduce notation for the difference between estimates using carbon storage vs air-sea flux.

line 132: Again no ocean subscript needed, use symbol to symbol to specify we are talking about betas calculated from carbon storage.

We prefer to keep the subscript ocean but we will introduce notation for the difference between estimates using carbon storage vs air-sea flux.

line 135: Rather than "can be expressed as" it would be more direct to write "were diagnosed" to clarify that this is what you do in this study.

Agreed

line 155: "contemporary atmospheric CO2". Maybe I have misunderstood something here. But, I would have thought this refers to projected atmospheric CO2? Could you please clarify.

'contemporary atmospheric CO2' will be altered to 'atmospheric CO2 taken at the same time'.

**Regional carbon cycle feedbacks in CMIP6 Earth system models**

line 263: replace "into contribution" with "into contributions"

Agreed

line 266: replace "non-linearity of ocean carbon cycle feedbacks" with "non-linearity of ocean carbon cycle feedbacks (see Equation 4)"

Agreed. We will re-organise this text and modify equations 22 and 23 following the description of the global carbon cycle feedbacks in equations 4 and 5 to avoid any confusion.

line 266: replace "n notes" by "n denotes"

Agreed.

line 305: or that the region dominating the the carbon cycle feedback differs between the models.

Agreed, we will add the suggested text.

**Control of the Atlantic Overturning circulation to the carbon cycle feedbacks**

The title seems back-to-front. Suggestions: "Relationship between carbon cycle feedbacks and the Atlantic Meridional overturning circulation" or "Control of the carbon cycle feedbacks by the Atlantic overturning circulation"

We will change the title for this section to 'Dependence of the carbon cycle feedbacks on the Atlantic Meridional Overturning Circulation'

**Discussion and Summary**

line 494: Similarly "control of the Atlantic Meridional Overturning circulation" is not the right title. You are talking here about the control of the feedbacks by the AMOC, not the controls on the AMOC itself.

We will change the title for this subsection to 'Effect of the Atlantic Meridional Overturning Circulation'

---

## Author Response (AR1)

Dear Editor Prof. Joos,

I would like to submit the revised manuscript entitled "Ocean carbon cycle feedbacks in CMIP6 models: contributions from different basins" for consideration for publication in Biogeosciences.

In this revised manuscript we addressed all the reviewers' concerns, as described below in our reply to comments. We re-organised sections 2 and 3 drawing on the second reviewer's suggestions, which we believe that further improved the presentation of our study. However, we decided to keep the manuscript structured in terms of themes, rather than in terms of methods and results (as suggested by the reviewer), which we think is more intuitive.

In the main text, we introduced and discussed the Arctic Ocean as a separate basin and split the old Figure 2 into two figures (Figs 3 and 4) that now show results for the global ocean and different ocean basins. We introduced additional material and figures less central to the main message to an appendix (Appendix B) and a supplement. We also added the MIROC-ES2L Earth system model to our analysis after personal communication with the Japan Agency for Marine-Earth Science and Technology.

Thank you for your consideration and I look forward to hearing from you.

Yours sincerely,

Dr Anna Katavouta

**Reply to review by Jörg Schwinger (reviewer 1)**

We thank Dr Jörg Schwinger for his constructive comments.

The authors present a detailed analysis of ocean carbon cycle feedbacks, which is a useful extension of the recently published global CMIP6 carbon cycle feedback paper (Arora et al. 2020). The manuscript goes beyond the Arora et al. study in that it focuses on the contribution of different ocean basins to the feedbacks and also explores circulation changes (using AMOC strength as a proxy) and their relation the feedbacks. The authors diagnose the basin wide contributions of preformed (saturated/disequilibrium) and regenerated carbon pools to the feedbacks. The description of the methodology is more detailed and contains additional diagnostics compared to the Arora et al. study. This manuscript is clearly within the scope of Biogeosciences, and I believe it will be of great interest to ocean carbon cycle community. The manuscript is generally well written (with some exceptions pointed out below) and I recommend it for publication in Biogeosciences after a few points detailed below have been addressed by the authors.

Thank you for your overall positive view.

Main points

1) Title: I don't think "controls of feedbacks from different ocean basins" is a good title. I am not a native speaker, but this sounds a bit odd to me. What the authors present is the "contribution of different ocean basins to carbon cycle feedback", and feedbacks are also attributed to different processes (including AMOC). As also noted further down, I don't think that the wording "control of AMOC on feedbacks in CMIP6 models" is appropriate. For CMIP6 models, the authors show a correlation between pre-industrial AMOC and AMOC weakening. "Control" implies a detailed mechanistic explanation, in my opinion. This is beyond the scope of this study, but therefore I would avoid using the word "control" here.

Agreed. The title was changed to: 'Ocean carbon cycle feedbacks in CMIP6 models: contributions from different basins.'

2) The authors base their definition of feedbacks on changes in DIC-inventories, which, as they note, makes only very little difference at the global scale. However, at the regional scale the difference can be large (see Fig. 3), and therefore I suggest to use a different symbol for the feedbacks based on DIC-inventories. Since the feedbacks derived from carbon fluxes are the standard definition, I would use something like beta*/gamma* for the feedback estimate based on inventory changes. This would also simplify the discussion at the beginning of Section 3, where the two beta/gamma definitions are compared (the authors could then just write "beta*" instead of "beta, estimated from the regional ocean carbon storage").

Agreed, we introduced new equations and notation for the estimates of beta and gamma based on the carbon storage as suggested by the reviewer (lines 140-154).

Also, it seems the authors point out that the feedback definition based on inventory changes makes more sense than the "traditional" one based on accumulated fluxes (line 290-292: "...to gain more mechanistic insight, so as (i) to account explicitly for the ocean transport of carbon..."). Here I would disagree: From the feedback perspective, the flux at the air-sea interface (and changes to it) is the process we are interested in. Transport of carbon below the ocean surface leads to a disconnect between the actual feedback process at the surface, and where DIC-inventory changes are diagnosed (nicely illustrated by Fig. 3). Don't get me wrong here: I think the method the authors use is extremely useful to gain a global to large-scale

regional understanding of ocean carbon cycle feedbacks, but there is a price to pay. In lines 290-292 it sounds like the authors are selling this "price to pay" as an advantage. Maybe the authors can re-consider their wording here?

Agreed. We rewritten this part to clarify that the carbon cycle feedbacks estimated based on the carbon uptake better describe the atmosphere-ocean interaction, while the estimates from carbon storage instead better describe the response of the ocean carbon budget to emissions. We clarify that we focus on the later as to enable diagnostics in terms of the preformed and regenerated carbon pools (lines 181-184).

3) Figure 3 and related discussion at the beginning of Section 3: This is one of the core figures of the manuscript, but it doesn't account for model uncertainty. I think it would strengthen the manuscript if the authors could expand this figure by 4 panels visualising the model spread (or standard deviation) for the 2 beta/gamma pairs, and add a brief discussion of where the main model uncertainty lies (and how and why this is different for the two definitions of regional feedbacks). Also for beta/gamma based on the DIC inventory, it would be great to split this further into the components (sat/diss/reg; this Figure could go into the Appendix).

Agreed. We expanded this figure to include the model uncertainty (Figure 2, in the updated manuscript) and discussed this model uncertainty in the main text along with the inter-model mean for beta and gamma (lines 161-180). We also introduced a figure with the maps for the split of beta ad gamma into saturated, disequilibrium and regenerated carbon pools (Figure B1) and discussion in Appendix B.

4) Figure 5 and related discussion: This Figure seems to be flawed:

  -why does the total ocean volume add up to only 99%?

  -why do the different contributions to beta/gamma add up to different percentages (between 91 and 99%)?

In retrospect we recognise that not explicitly defining our ocean basins (related to comment 6 below) may lead to confusion. The regional contribution to beta and gamma and their different components is not proportional to the regional volume, and each is controlled by different processes. Hence, the fractional contribution from the combined Atlantic, Pacific, Indian and Southern Oceans (excluding the Arctic and semi-enclosed basins) to beta and gamma are different to each other, and are different to their combined fractional volume. This distinction can be equivalently viewed in terms of the Arctic and semi-enclosed basins having different contributions to beta and gamma, which are also different from their fractional volume. This distinction is also the case for the saturated, disequilibrium and regenerated components of beta and gamma.

In the revised manuscript we included the Arctic Ocean as a separate basin, and all the figures, tables and text were updated to include the Arctic. We do not include the semi-enclosed seas to any of the basins (see supplement Figure C1). These semi-enclosed seas have a fractional volume of less than 0.5% of the global ocean, and a contribution to beta and gamma and their different components less than 2% (Figure 6).

5) The discussion in lines 332-337 is unclear to me: "In the well ventilated Atlantic Ocean, the additional heat penetrates into the ocean interior and is not confined to the ocean surface, which limits the effect of the reduction in solubility with warming". But the definition of gamma_sat doesn't care whether a water parcel is at the surface or not. I don't think that this is can be the explanation (same in the next paragraph for the Pacific). Please clarify this

Agreed, the reviewer is correct. This point is associated with the non-linearity of gamma as described in equation (20). We updated the text as (lines 375-385): *'The Pacific and Indian Oceans contributions to γsat are slightly smaller than expected from their fractional volumes (Fig. 6b), consistent with a low warming per unit volume in these basins (Fig. 1d). … The Atlantic Ocean has a smaller contribution to γsat than expected from its fractional volume (Fig. 6b), despite experiencing large warming (Fig 1d), which suggests that the non-linearity of the carbonate system is important in this basin. Specifically, the Atlantic Ocean has a large increase in DIC (Fig 1c) which acts to significantly reduce the magnitude of the negative γsat driven solely by the effect of warming on solubility (see Eq. (20)).'*

6) Nowhere in the manuscript it is stated how the ocean basins are defined. Where is the delineation between the Southern Ocean and the other basins? What about the Arctic Ocean? Is it included in the Atlantic or omitted? What about marginal seas? Also, the definition of AMOC strength is not given. Please add this information.

Agreed. We now provide bounds in terms of latitude in the main text (lines 186-187), and a map of the different basins is presented in supplement Figure C1. The Arctic Ocean is explicitly discussed in the updated manuscript.

We now provide the definition for the AMOC strength and weakening in lines 457-458: *The strength and weakening of the AMOC are diagnosed as the maximum pre-industrial AMOC between 30oN and 50oN (Fig. 8) and the maximum AMOC change between 30oN and 50oN (Fig. 9), respectively, in 11 CMIP6 Earth system models (Table 3).'*

Note that in the original manuscript the values for the AMOC weakening were wrongly reported in Table 3 as: Maximum(AMOC projected) – Maximum( AMOC preindustrial). Table 3 has been updated to report the AMOC weakening as: Maximum (AMOC projected – AMOC preindustrial), consistent with Figure 9.

Minor points

7) lines 1-3: This sentence is complicated, and the wording isn't very precise (it is not a "competition between the increase in atmospheric CO2" but "a competition between the response to the increase in atmospheric CO2..."). Please consider rewording this sentence and maybe splitting it into two.

Agreed. We re-written this text (lines 1-3)

8) Equation 9: "Delta f" is not defined. Here, I would find it worthwhile writing the equation first in terms of DIC_sat, and state that DIC_sat=f(CO2,....). Then write down explicitly what Delta f means.

We clarified and expanded this description for this function in lines 225-233. Delta f is an iterative algorithm, rather than a simple function, that estimates DIC_sat without prior knowledge of the [H+]_sat and by explicitly considering the contribution from borate, phosphate and silicate to alkalinity. This algorithm follows an iteration based on a first guess of [H+] or pH and is explicitly described in Follows et al., 2009 along with a coding example.

9) Equation 10 and 11: Here, I think, it would be easier and shorter to just express beta and gamma in terms of I_sat (without writing out the f-terms explicitly - since I_sat is already defined in terms of f just a few lines above).

Agreed. We re-organised these equations (lines 221-225).

10) Equation 18 and 19: Here, I also think it is much easier to understand if beta and gamma are just expressed in terms of I_reg, which is defined just a few lines above in Eq. 17.

Agreed. We re-organise these equations (lines 288-296).

11) line 84-85: This is an assumption, not a conclusion, so starting the sentence with "Hence..." is not appropriate. Maybe use "In the carbon cycle feedback framework introduced by Friedlingstein et al. (2003,2006) it is assumed that..." or similar.

Agreed. We changed the text as recommended (lines 86-88).

12) line 266: "...where gamma_n includes the non-linearity of ocean carbon cycle feedbacks". This is confusing, it sounds like the definition of gamma_n would be different from that of gamma, which is not the case. I suggest to delete this.

We re-organised this text and equations (lines 127-139).

13) Figure 3: To make this figure consistent with all other results, please add NorESM2-LM (fgco2 for the BGC run is available).

Agreed. We added NorESM2-LM. Please note that we also added MIROC-ES2L to our analysis.

14) line 286: What do the authors mean by "asymmetries"? Please clarify.

Agreed, spatial asymmetries here refer to the regional differences/pattern in gamma (regions of positive and negative gamma). We re-written this text without using the word asymmetry for clarity (lines 176-180).

15) line 323-324: "By definition, the contribution of each basin to beta_sat and beta_dis is approximately proportional to the ocean volume contained in each basin...". I see that this is the case for beta_sat, but for beta_dis this depends on ventilation which is not related to the volume. Maybe delete beta_dis here?

Agreed. We deleted beta_dis (lines 365-366).

16) line 445-447: Please check and reword this sentence (consider splitting in two).

Agreed. We re-written and split this sentence (501-503).

17) line 476-477: "...which is mainly due to the disequilibrium carbon pool and the reduction in the physical ventilation with climate change." The second part of this sentence is a conclusion, isn't it? Then it would be more appropriate to write: "...which is mainly due to the disequilibrium carbon pool, indicating that the Atlantic has the strongest reduction in the physical ventilation with climate change."

Agreed. We re-written this part (lines 533-535).

18) line 487: This is also seen in Schwinger et al. 2014

Thanks. We referenced this study (lines 547-548).

19) line 491-492: "The inter-model variability in gamma amongst CMIP6 models is relatively large compared with beta...". I think it is worth mentioning that this is not true in terms of the

absolute feedback strength: In terms of PgC taken up by the ocean, it is still the uncertainty in beta that plays the dominant role.

Agreed. We clarify that the inter-model variability for gamma is larger than for beta in relative terms (variability in relation to the mean as described by the coefficient of variation) but that beta has more uncertainty in absolute terms for the carbon storage (variability in PgC). Lines 120-125: *'The variability in β amongst the Earth system models, as described by the coefficient of variation, CV, is relatively small on the global scale (CV=0.09) when compared with the variability in γ (CV=0.43) (Table 2). However, for the uncertainty in the ocean carbon gain due to carbon emissions, the carbon-concentration feedback contributes to a spread of 62 PgC, while the carbon-climate feedback contributes only to a spread of 25 PgC amongst the CMIP6 Earth system models on a global scales and for a quadrupling of atmospheric CO2; where the spread corresponds to one standard deviation.'*

20) line 513: "...controlled by the AMOC weakening..." As pointed out above, the authors find a correlation, so in my opinion the term "control" should be avoided here. Please consider rewording.

Agreed. In the revised manuscript we used 'control' only when we refer to results from the sensitivity experiment with the idealised model. When we refer to CMIP6 and the results based on the correlation we used something along the lines of 'correlation' of 'dependence' (e.g., line 573).

Technical

21) lines 26-28: Please check the grammar and logic of this sentence.

Agreed. We rephrased this text (lines 28-32).

22) line 35: modes -> models

Changed.

23) line 43: "defined on" -> "defined based on"

We rephrase this to *'defined in terms of …'* (line 48).

24) line 83: "..such as for example leading to..." please check grammar

We have deleted this text.

25) line 121: "at the surface" is confusing. Maybe better: "is the part of DIC that has been transferred from the surface into the ocean interior..."

Agreed, we changed the text as recommended (lines 205-206).

26) line 129: "is a unit conversion" please spell out from which to which unit.

Agreed, we changed to '… *is a unit conversion from moles to Pg of carbon'* (line 215).

27) line 180: "The term inside the first {} brackets..." -> "The first term in curly brackets..."

Agreed, we changed as recommended (line 270 and line 272).

28) line 199: "to the alkalinity" -> "to alkalinity"

Changed

29) line 285 South -> Southern

Changed

30) line 328: necessary -> necessarily

Changed

31) line 349: "by the ocean ventilation" -> "by ocean ventilation"

Changed.

32) line 356: delete "now"

Deleted.

33) line 372-373: "is the preindustrial" -> "denotes the preindustrial state" (or similar)

Changed to '… *denotes the pre-industrial state*' (lines 421-422).

34) line 470: "...of 26% to 30%..." -> "...between 26% and 30%..."

Agreed, we rephrased to *'of between 26% and 30% …'* (line 525).

**Reply to anonymous reviewer (reviewer 2)**

**General comments**

The paper is a valuable extension of the global analysis of carbon cycle feedbacks (Arora et al., 2020) and presents a useful framework that uses changes to the interior ocean carbon pools to help infer the first-order mechanisms contributing the magnitudes of regional and, thus, global ocean carbon cycle feedback parameters. The focus is primarily on how and why the strengths of the ocean carbon cycle feedback parameters vary at the basin-scale. The authors use diagnostics of the saturated, regenerated, disequilibrium carbon to help interpret these basin-scale variations. The also include a detailed mechanistic analysis of the relationship between AMOC on the feedback parameters. I particularly appreciated i) the thorough theoretical explanation for the evolution of the global ocean feedback parameters based on the contribution from the diagnosed saturated, disequilibrium, and regenerated carbon pools, and ii) the use of the box models to explore the response of these carbon pools to different AMOC scenarios.

We thank the reviewer for their positive view and constructive comments.

1) The global analysis of the evolution of the carbon cycle feedback parameters based on contributions from the different carbon pools (figure 2 with explanations rooted in fundamental chemistry)  is rigorous and informative, but it  would be easier to follow if this analysis were shifted and consolidated in a dedicated results section (details below).

Agreed, we re-organised the manuscript (please see reply to comment 5 below).

2) I would have liked to see a basin-scale analysis similar to the global analysis of the evolution of the carbon cycle feedback parameters based on relative contributions from the different carbon pools. Why did you not use the same approach and figure presentation? Was it too difficult to interpret presented this way?

Agreed. We introduced this basin-scale analysis for the carbon cycle feedback parameters in Figures 3 and 4 of the revised manuscript. These figures and the evolution of the carbon cycle feedback parameters based on the relative contributions from the different carbon pools, for the global ocean and the different ocean basins, are discussed in the revised section 3.

3) This paper focuses mostly on the regional feedback parameters calculated from the changes to carbon storage in the ocean interior. The ocean carbon cycle feedback parameters derived from air-sea CO2 fluxes are not analysed much here.  Globally the feedback parameters based on the air-sea flux and storage approaches are very similar, but—as the authors note— differences emerge at regional scales, largely because of the influence of transport. I think it is important to introduce the reader to the strengths and limitations of these complementary approaches. For example, basin-scale feedback parameter calculated using changes in the inventories of interior carbon pools,  aggregates many spatially and temporally varying processes over large ocean basins, which has it's limitations in terms of deciphering  the driving mechanisms since it is integrating  DIC changes that are driven by different and often compensating processes within the region and from outside the region itself  (particularly in the Southern Ocean). But, on the other hand, even if the exact sources of the changes are more difficult to pinpoint, the interior carbon pools provide an integrated picture of the first order controls on changes in carbon storage in the different basins. Furthermore, we can use the diagnosis of the various carbon pools to help interpret the mechanisms behind these changes.

We introduced new equations and symbols for the regional beta and gamma estimated based on cumulative flux versus estimated based on carbon storage in the revised section 2, to highlight how these two estimates are different (lines 133-154). We added/re-written the text to clarify that the carbon cycle feedbacks estimated based on the carbon uptake better describe the atmosphere-ocean interaction, while the estimates from carbon storage instead better describe the response of the ocean carbon budget to emissions. We clarify that we focus on the later as to enable diagnostics in terms of the preformed and regenerated carbon pools (lines 181-184).

4) Here the analysis of the processes (ocean carbonate chemistry, physical ventilation, and biological processes) driving changes in the carbon cycle feedback parameters are diagnosed from changes to the saturated, disequilibrium and regenerated DIC pools. So that the reader, can better evaluate the results presented from this part of the study an honest appraisal of the strengths and limitations of this approach would also be a valuable addition.

i) For example, interpreting changes in the distributions of the different carbon pools is not always straightforward as you might hope (see spatial complexity in Arora et al., 2020). Nevertheless, basin-scale changes in carbon storage provide a natural integration of a spatially complex air-sea CO2 flux pattern.

We agree with this point and this complexity is reflected in the regional maps for beta and gamma (Figure 2 in the revised manuscript). We introduced and discussed a new Figure (Figure B1) for the regional maps of the saturated, disequilibrium and regenerated component of beta and gamma in Appendix B.

ii) Even using this approach it is not possible to completely separate the various processes. For example, the regenerated component combines both biological and ventilation changes because the strength of the biological pump is also dependent on the ventilation. It seems that describing the changes in terms of their impacts on the different carbon pumps could be helpful.

We agree that the regenerated component depends on changes in the ventilation as we describe in lines 300-306 and lines 340-341, and this dependence leads to a significant correlation between gamma_regenerated and AMOC in the Atlantic Ocean as described in lines 493-496 in the revised manuscript. We incorporate this effect of the changes in the ventilation on the biological carbon pool to biological processes (lines 300-306), since in the absence of biology this effect would be zero. We introduced text in the revised manuscript to clarify that the regenerated component corresponds to the biological pump and the preformed component to the solubility pump (lines 207-208). We then go on to split the preformed component into two idealised pools: a saturated and a disequilibrium pool, such that we split the global solubility pump into a chemical and physical component in terms of the carbon uptake and transfer. We prefer to discuss the changes in terms of physical, chemical, and biological processes rather than in terms of biological and solubility pump, and clarifying that the biological processes include the effect of physical ventilation to the biological carbon pool (lines 300-306).

iii) Basin-scale variations in the change in carbon storage can't be interpreted directly as the importance of the region to the global carbon cycle feedbacks, because the source (and mechanism) of change could have been from outside the region.

We introduced new equations and text, discussing how the regional carbon cycle feedbacks estimated based on regional carbon storage include the effect from the source and the effect from the transport on carbon storage (lines 133-154, please see reply to comment 3). Even for

the feedbacks defined by the air-sea fluxes, separating the local from the far-field control is still problematic as the local air sea fluxes include influence from transport of carbon or other tracers like temperature and nutrients from outside the region. Here, we are using a framework to provide insight on the basin-scale effect of different processes (biology, chemistry, physical ventilation) on the carbon cycle feedbacks no matter if those processes are of local or remote origin. We briefly discuss this caveat in our analysis in lines 587-592.

5) My main recommendation is to reorganise several sections of the paper to help clarify the methodology and distil some of the main results—in particular for the analysis of the contribution of the various carbon pools to the regional carbon cycle feedback parameters (detailed suggestions are given below).

We agree, and we restructured sections 2 and 3 of the revised manuscript:

1) **Section 2** discusses (i) the global ocean carbon cycle feedback framework, and (ii) the regional ocean carbon cycle feedbacks defined based on cumulative carbon uptake versus based on carbon storage, along with the basin-scale beta and gamma.
2) **Section 3** describes the separation of the carbon cycle feedbacks into contribution from different carbon pools, and discusses the processes that control these feedbacks on a global and basin scale, as well as the contribution from the different basins to the global feedbacks.

Our new structure draws on the reviewer's recommendations, but sections 2 and 3 are separated in terms of themes rather than in terms of methods and results, which we think is more intuitive and easier to follow.

6) It is difficult to see the relative contribution of saturated, disequilibrium and regenerated carbon pools to the basin-scale feedback parameters. I would suggest adding (or modifying) a figure dedicated to the relative contributions of saturated, disequilibrium and regenerated carbon pools to the basin-scale feedback parameters. Although, Figure 4 contains this information, it can be difficult to read this information off the figure.

Agreed, we introduced new Figures 3 and 4 to complement Figure 5 (old Figure 4) in the revised manuscript (please see reply to comment 2 above).

7) Related to the last comment, the author often presents the volume integrated quantities, which makes it more difficult to appreciate:

i) where a feedback is stronger or weaker than expected based on volume alone (Figure 5 shows this well, but sometimes it is lost in the text),

Agreed, we reorganised section 3 and the text associated with Figure 6 (old Figure 5).

and ii) what processes (diagnosed using the changes to the carbon pools) dominate the magnitudes of each carbon cycle feedback parameter in each basin (not always easy to see in the Figures and tables).

Agreed. We introduced and discussed new Figures 3 and 4 (please see reply to comment 2 above) to complement Figure 5 in the revised manuscript.

8) The authors suggest they will account for the impact of carbon transport. This influence is transport is not presented in much detail, and would require a more thorough comparison of regional air-sea CO2 flux and storage feedback parameters. Therefore I would either i) elaborate on this in a dedicated section or ii) simply remove and include some reference to the

impact of transport in the methodology where the relative merits of flux and storage approaches are discussed.

We introduced the supplement Figure C2 that shows explicitly the direct influence of the carbon transport to the feedbacks estimated from the carbon storage, as captured by the difference between the feedbacks estimated from the cumulative carbon fluxes (Figure 2.b) and the carbon storage (Figure 2.a). We also discuss this effect of the carbon transport to the carbon cycle feedbacks estimated from the carbon storage in the dedicated subsection 2.2.1. This direct influence from carbon transport is also shown by the deficit between the black and red lines in Figure A1 and discussed in Appendix A. We clarified that we do not separately estimate and isolate the effect of the transport of carbon and other tracers (e.g., temperature, salinity, nutrients) on the regional source of carbon itself, which will need further analysis (lines 590-592).

9) Several time the authors comment on the beta having less uncertainty (intermodel variability) than the gamma. This is misleading, because the uncertainty in each is mapped disproportionately onto the uncertainty in carbon storage (i.e. gamma is multipled by only > 4 deg C , while beta is multiplied by the   > 700 ppm change in atmospheric CO2) .

Agreed. We clarified that the inter-model variability for gamma is larger than for beta in relative terms (variability in relation to the mean as described by the coefficient of variation) but that beta has more uncertainty in absolute terms for the carbon storage (variability in PgC). Lines 120-125: *'The variability in β amongst the Earth system models, as described by the coefficient of variation, CV, is relatively small on the global scale (CV=0.09) when compared with the variability in γ (CV=0.43) (Table 2). However, for the uncertainty in the ocean carbon gain due to carbon emissions, the carbon-concentration feedback contributes to a spread of 62 PgC, while the carbon-climate feedback contributes only to a spread of 25 PgC amongst the CMIP6 Earth system models on a global scale and for a quadrupling of atmospheric CO2; where the spread corresponds to one standard deviation.'*

**Specific comments**
**Abstract**

10) On beta: "The Atlantic, Pacific and Southern Oceans contribute equally to the carbon-concentration feedback, despite their different size." I think a better emphasis is that the Atlantic storage increase more and Pacific storage increases less than expected in relation to their size. Then mention why. Also the Southern Ocean beta is low despite the strong air-sea CO2 flux here, because it transported out of the region. what about the Indian? Summarise on the controls on beta for each basin

We rewritten parts of the abstract as to clarify that we refer to contribution to the carbon cycle feedbacks when estimated in terms of carbon storage (e.g., lines 6-8). We also added that '*The Southern Ocean has a large anthropogenic carbon uptake from the atmosphere, but its contribution to the carbon storage is relatively small due to a large carbon transport to the other basins.*' (lines 9-11) and '*The more poorly-ventilated Indo-Pacific Ocean provides a small contribution to the carbon cycle feedbacks relative to its size*' (lines 16-17).

11) On gamma: Similarly, I would summarise the dominant controls for each basin, merging the conclusions from the carbon pools and AMOC parts of the analysis.

We added discussion on the different basins (lines 11-17). We prefer to keep the AMOC part of the analysis discussed separately in the abstract.

**Introduction**

12) lines 25 –28: Given the central importance of these feedbacks in this papers, this introduction to carbon cycle feedback parameters could be a little more comprehensive. It also a bit awkward to read. Perhaps merge the name and description of each feedback. And then finish with "These two carbon cycle feedbacks have been extensively used to...."

Agreed, we re-written the text as recommended (lines 29-34).

13) lines 74–79: rework section description to reflect any changes made to the sections.

Agreed, we updated the text following the changes in sections 2 and 3 (lines 77-83)

**Ocean carbon cycle feedbacks and their control by different processes**

14) The beginning of this section reads like part of the introduction. I think it should be clear to the reader from the onset that this is where you will present the approach used in this paper.

I would start this section introducing your methodology and how the methodology compares to previous studies, particularly Arora et al., 2020. Something along the lines of "here we extend the analysis of the carbon cycle feedbacks diagnosed in the CMIP6 models (Arora et al., 2020)…." and then introduce what part of the analysis is identical and what parts have been modified or extended.

We re-organised sections 2 and 3, please see reply to comment 5. We clarify that we follow the carbon cycle feedback framework introduced by Friedlingstein et al. (2003, 2006) (lines 86-88). We also clarify that we follow the framework of Williams et al. (2019) and Arora et al. (2020) to separated beta and gamma into contribution from the regenerated, the saturated and the disequilibrium ocean carbon pools (lines 199-201).

15) I recommend separating the methodology into two parts: 2.1 Ocean carbon cycle feedback analysis, 2.2 Diagnostics of processes controlling carbon cycle feedbacks.

We re-organised sections 2 and 3, please see reply to comment 5 above.

16) Move the results and discussion of the contribution of the global saturated, disequilibrium, and regenerated carbon pools (and the associated Figure 2) to the global carbon cycle feedback parameters to section 3. It is very interesting and thorough and deserves a dedicated results section.

We moved these results and discussion to section 3, along with the methodology for separating these feedbacks into contribution from the different carbon pools. Please see reply to comment 5 above.

17) For improved clarity, the ocean cycle carbon cycle feedbacks (e.g. recommended Section 2.1) should consolidate all the feedback methodology from other sections and include some missing elements:

We prefer to organise the manuscript in terms of themes rather than in terms of methods and results, such as the methodology for the global and regional ocean carbon cycle framework is discussed in section 2, and the methodology for splitting the feedbacks intro contributions from different carbon pools is discussed in section 3. Please see reply to comment 5 above.

(i) Clearly present and give distinguish between the feedback parameters and the carbon inventories calculated from the i) air-sea $CO_2$ fluxes and ii) interior carbon inventories. In

regional analyses this distinction becomes very important (as the authors explain). For example, you could use 'betaf' and 'If' and 'betas' and 'Is' for beta and carbon inventories calculated from carbon fluxes and carbon storage respectively.

Agreed. We introduced equations and notation to clearly present and distinguish the feedbacks estimated based on cumulative air sea flux and based on the carbon storage (lines 133-154).

(ii) Please include a discussion of the merits/limitations of the flux vs storage approaches and their complementarity.

Agreed. We introduced equations that explicitly show how the feedbacks estimated based on the carbon storage relate to the feedbacks estimated based on carbon uptake (150-154). We also clarified that the carbon cycle feedbacks estimated based on the carbon uptake better describe the atmosphere-ocean interaction, while the estimates from carbon storage instead better describe the response of the ocean carbon storage to emissions (lines 181-184).

(iii) Move the regional calculation of the feedback parameters (263–274) to this section.

Agreed. We moved and expanded this text to section 2.2 (lines 127-154).

(iv) The basins have not been defined. Please provide the precise boundaries used for the basin-scale analysis show boundaries on the maps in Figure 3.

Agreed. We now provide bounds in terms of latitude in the main text (lines 186-187), and a map of the different basins is presented in supplement Figure C1.

18) For the section (e.g. recommended section 2.2) presenting the methodology for diagnosing the ocean processes contributing the carbon cycle feedbacks ( based on calculations of saturated, disequilibrium, regenerated carbon pools)

(i) Start this section with something like your line 115. "To gain insight …"

(ii) Include detail here on the how the various carbon pools are used to diagnose ocean processes (ventilation, biology…) and the limitations of this approach.

(iii) I assume methodology used to calculate the carbon pools is identical to that used in Arora et al., 2020? Please elaborate. If so, it would be useful to inform the reader.

(iv) Include here only the methodology used to calculate the changes in the carbon pools and leave the theoretical explanations that are used to better understand their evolution to section 3.

We re-organised sections 2 and 3. We prefer to organise the manuscript in terms of themes rather than in terms of methods, such that the methodology for splitting the feedbacks intro contributions from different carbon pools is discussed in section 3, along with the theoretical explanation and results. We clarify that we follow the framework of Williams et al. (2019) and Arora et al. (2020) to separated beta and gamma into contribution from the regenerated, the saturated and the disequilibrium ocean carbon pools (lines 199-201).

19) line 81–84: this essentially repeats introductory section above (lines 25–28 ) and doesn't need to be repeated here.

Agreed. We removed this text.

20) line 106: this does not "exclude" all model biases because the initial biases continue to impact their evolution, but it does help reduce them. Either replace "To exclude" by "To reduce" or "To partially exclude".

Agreed, we revised the text to '*reduce*' (lines 109-110).

21) line 251: replace "suggesting" with "indicating"

Changed. (line 119)

22) line 253: replace "act to reduce" with "reduce"

Changed. (line 339).

**Regional carbon cycle feedbacks in CMIP6 Earth system models**

23) To make the paper easier to follow, this section should consolidate the global and basin-scale analyses of the contribution of the carbon pools to the carbon cycle feedback parameters

We re-organised sections 2 and 3, please see reply to comment 5 above. Section 3 is now dedicated to the processes controlling the global and basin-scale carbon cycle feedbacks, including related methodology/diagnostics, theory and results.

24) You could rename the title of this section "3 Processes controlling the carbon cycle feedback parameters" and subsections "3.1 Global analysis" and "3.2 Basin-scale analysis". Maybe there is no need for subheadings.

We renamed section 3 to 'Processes controlling the carbon cycle feedbacks in CMIP6 models'.

25) The section starting on line 275 touches on the impact of carbon transport on the feedback parameters and on line 291 mentions that the study will account explicitly for the ocean transport of carbon. Since there there is little discussion of the impact of transport on the feedbacks in the paper and the main results are added to an Appendix, these paragraphs seems out of place here. Two suggestions:

(i) Either include a more detailed examination of the impact of transport on regional betas and gammas in a dedicated results& discussion section including the figures in the Appendix. This would be interesting if it extends our understanding beyond what is presented in Frolicher et al., (2015).

(ii) Or, the spatial distribution of the ocean carbon-cycle feedbacks in figure 3 could be moved into and discussed in the methodology section where the air-sea CO2 flux and carbon storage feedback parameters are introduced and compared. Here the figures could be discussed in terms of the complementarity of the flux and storage approaches.

We moved this part of the study into the new sub-section 2.2, and discussed the differences and similarities calculating feedbacks from carbon storage vs carbon flux in 2.2.1 (lines 155-184). Please see reply to comments 3 and 8 above. We still prefer to keep Figure A1 and discuss it in an appendix such that this information is available to the reader but the new section 2.2. is kept focused on the different ways to evaluate the feedbacks.

26) There are some places—particularly in the interpretation of the changes in the global and regional carbon pools and their contributions to the feedback parameters—where the language used to associate the changes to mechanisms, reads as if these are mechanisms have been determined rather than diagnosed. I think care needs to be taken to word the results and

conclusions in light of an honest appraisal of the confidence we have in the mechanisms diagnosed using this approach (probably in the methodology).

Agreed, we updated the text to reflect that the mechanisms are diagnosed rather than determined.

27) lines 263 –271: As mentioned earlier, move this methodological detail to section 2.

Agreed, this part was moved to sub-section 2.2.

**Global analysis**

28)  For the Global analysis  I would recommend moving the global analysis of the contributions of the various ocean carbon pools to the carbon cycle feedback parameters  (I.e. associated with Figure 2) into a dedicated section, which  includes the following text:  the presentation of the theoretical analysis  that helps us interpret the evolution of the global feedback parameters: saturated component  (lines 148–196), regenerated (lines 208 – 217), and disequilibrium (lines 225–238 starting with the rise in atmospheric CO2….).

We tried different structures for the manuscript as recommended by the reviewer. However, we decided to keep the methodology for diagnosing the contributions from each of the carbon pools together with the theoretical insight and the discussion of new Figures 3 and 4 in subsections 3.1-3.3. We think that this structure is more intuitive for understanding these contributions.

**Basin-scale analysis**

29) I feel this section of the paper needs some polishing. The figures and the discussion don't always clearly separate the impact of the strength of the feedback (i.e. volume-normalised) from the volume-integrated impact. Consequently, it can be difficult to clearly see the relative contributions of the saturated, disequilibrium and regenerated carbon pools to the basin-scale parameters. This was done well for the global analysis.

We re-organised sections 2 and 3. We also have re-written the text associated with the basin scale analysis and introduced Figures 3 and 4.

30) I find the separation between section 3.1 and 3.2 confusing. Only a short discussion of the contribution of different basins to global carbon cycle feedback parameters is needed here (the main results are already in Figure 1). The main focus should be i) the relative strengths of the carbon cycle feedbacks relative to their volumes and ii) on the analysis of the mechanisms controlling the differences between the basin-scale carbon cycle feedback parameters (volume-normalised) by accounting for the contributions from the different carbon pools.

Agreed. We have re-organised and re-written this text into

1) Section 2.2.2: short discussion on the contributions from different basins to beta and gamma in terms of their magnitude and inter-model spread in CMIP6.
2) Section 3.4: discussion on the combined effect from saturated, disequilibrium and regenerated carbon pools to beta and gamma, on a global and basin scale.
3) Section 3.5: discussion on the contribution from different basins to the carbon cycle feedbacks relative to their volumes and the processes that control this contribution.

31) I would start section on the relative strengths of the feedbacks by discussing Figure 5, which is better suited for this than figure 4.  For example, for beta, It is easy to see that the Atlantic takes up more and the Pacific less relative to their volumes

In the new section 3.4, Figure 5 (old Figure 4) is discussed in terms of contribution of the different carbon pools to the basin-scale feedbacks. In the new section 3.5, the contribution of the different basins to the global carbon cycle feedbacks relative to their volume is discussed in terms of Figure 6 (old Figure 5), as suggested by the reviewer.

32) A figure is missing that is dedicated to the relative contributions of saturated, disequilibrium and regenerated carbon pools to the basin-scale feedback parameters. For example, you want to be able to easily see what mechanisms is responsible for the Atlantic taking up more and Pacific taking up less carbon relative to it's volume or what process dominates the carbon-climate feedback in each basin. Although, Figure 4 contains this information, since the feedbacks parameters are i) not volume-normalised and ii) the components of the feedback parameter for each region are not presented side-by-side, it can be awkward to read this information form the figure. Maybe Figure 4 could be reworked by :

Agreed. We introduced Figures 3 and 4 showing the contribution of the different carbon pools to the basin-scales feedbacks (see reply to comment 2 above), to complement Figure 5 (old Figure 4).

(i) either grouping the contributions from each carbon pool into a plot for each basin rather than for each component.

The grouping in terms of basins rather than carbon pool components does not work well for beta in Figure 5 (old Figure 4) that focuses on magnitude and range, as the total, the saturated, disequilibrium and regenerated parts are different orders of magnitude. However, we now introduce Figures 3 and 4 to complement Figure 5.

(ii) or presenting the contribution of each component to the basin-scale feedback parameters as percentage contributions, so that it is easy to clearly see how the relative contributions of the different components to the total feedback vary between the regions.

The contribution for the saturated, disequilibrium, and regenerated components to each basin or to the global ocean cannot be meaningfully presented as percentage as these different components have a positive or negative contribution (opposing each other, see Figures 3, 4 and 5).

In contrast, the contribution of each basin to the different components (saturated, disequilibrium and regenerated) can be presented as percentages, since on average the different components are of the same sign for all the basins and for the global ocean: e.g., the contributions of the regenerated component to gamma is on average positive for all basins for a quadrupling of atmospheric CO2, while the contribution of the saturated and disequilibrium components to gamma is on average negative for all basins for a quadrupling of atmospheric CO2 (see Figures 4 and 5).

33) It would be informative to see the geographical distributions of the saturated, disequilibrium, and regenerated components to help with the interpretation of the basin-scale changes and to complement the depth sections presented in Arora et al., (2020).

Agreed. We introduced and discussed maps for the saturated, disequilibrium and regenerated component in Appendix B (Figure B1).

**Discussion and Summary**

34) line 163: In "The transport effect acts to decrease the carbon-concentration feedback parameter" it is important to mention increase relative to what. That is, the carbon-concentration parameter calculated from the air-sea CO2 fluxes.

Agreed, we have re-written this text following the reviewer's suggestion (lines 522-524).

35) line 473: again the intermodel variability in beta may seem small, but the impact of this intermodel variability on carbon storage is not.

Agreed. We removed this text from the Discussion and Summary section, but we clarified this point in lines 120-125 (see reply to comment 9 above).

36) line 479: "consistent with previous studies" here mention consistent with the spatial patterns of gamma diagnosed from the air-sea CO2 fluxes in the last two generations of ESMs (i.e. spatial patterns of the feedback parameters in Roy et al., 2011 and in the CMIP5 models in the IPCC WG1 assessment report Ciais et al., 2013).

Agreed, we referenced these studies (lines 535-537).

37) line 490: It would be useful to reference consistency with previous analyses of the drivers of the carbon-concentration parameter distributions here even if they were based on the CO2 air-sea flux carbon cycle feedbacks "as was shown in a CMIP5-generation model…. and is consistent with analyses of the carbon-climate feedback distributions from previous generation models (Roy et al., and Ciais et al., 2013)"

Agreed, we referenced these studies (lines 551-552).

**Figures**

38) Figure 4: Does not include the multimodel mean (or median). I think it would be most informative to overlay the median and IQR.

Adding the mean and interquartile range crowds this figure and makes it hard to read. However, we reported the inter-model mean and standard deviation for the fields in Figure 5 (old Figure 4) in the supplement Table C1.

39) Figure 5: It looks like some of the parts of the ocean have not been included here. I suppose there is meant to be an 'Other regions' category (inland seas and arctic?) to make each column sum up to 100%. Please explain in the caption.

Yes, this is meant to be other regions including the Arctic. In the revised manuscript we included a separate Arctic Ocean, and clarified that the deviation from 100% is due to the contribution from semi-enclosed seas not included in any of the 5 basins in the caption of Figure 6.

Why were these other regions left out? They contribute a substantial amount to global gamma.

Agreed, the Arctic has a high contribution and in the revised manuscript we introduced a separate Arctic ocean basin (updated all tables, Figures and text to include the Arctic Ocean). The semi-enclosed seas (like the Mediterranean Sea, Hudson Bay and Red Sea) contribute less than 2% in the carbon cycle feedbacks and are not included is any of the main 5 ocean basins.

40) ## Technical comments

There are a few places where the English needs a little work. I have picked up a bunch of them, but someone should give the document another once over.

Agreed.

**Abstract**

line 4: Change ". The contribution from different ocean basins to the carbon cycle feedbacks and its control by the ocean carbonate chemistry, physical ventilation and biological processes is explored in diagnostics of 10 CMIP6 Earth …." to ". The contribution from different ocean basins to the carbon cycle feedbacks and the processes that control them are explored using diagnostics of ocean carbonate chemistry, physical ventilation and biological processes in10 CMIP6 Earth …."

Agreed, we have re-written this part (lines 3-5).

line 5: Change "mechanist" to "mechanistic"

Changed.

line 15: change "on global scale" to "at the global scale"

We removed this text.

**Introduction**

Line 21: change "refer" to "referred"

Changed.

Line 22: change "lapse rate" to "tropospheric lapse rate"

Changed.

Line 24: change "land and ocean" to "land and ocean reservoirs".

Changed

Line 34: change "feedback is about 3 times stronger over land than the ocean on centennial " to "feedback is about three times stronger over ::the:: land than the ocean on centennial "

Changed.

Line 55: I would remove "hence" because it implies you are going to focus on both heat and carbon in this study.

Changed.

line 56: Replace "Hence, our motive is to explore the mechanisms that lead to this regional variation in the carbon storage and the carbon cycle feedbacks for the different ocean basins…" with "Our ::motivation:: is to explore the mechanisms that lead ::these:: regional variations in the carbon storage and the carbon cycle feedbacks ::in:: the different ocean basins…"

Changed.

line 71: changes "insight for" to "insight into"

Changed.

line 72: This sounds odd. Please replace "Our aim is to provide insight for the relative contribution from different ocean basins to the ocean carbon cycle feedbacks, and the processes that drive this relative contribution and its uncertainty amongst CMIP6 " by "Our aim is to provide insight ::into:: the relative contribution of different ocean basins to the ocean

carbon cycle feedbacks  and the processes that drive this regional partitioning  in the CMIP6 models".

Agreed, changed as recommended by the reviewer.

line 73: it reads as if you will exploring the controls of the AMOC. Please replace by something like "the control of the AMOC on the carbon cycle feedbacks".

We removed this text.

Line 76: replace "processes" by "diagnostics of processes".

We have re-written this text.

**Ocean carbon cycle feedbacks and their control by different processes**

Line 81: comma between "CO2 which"

We removed this text.

Line 82: change"At the same time the increase in atmospheric CO2  modifies the physical climate system, such as for example leading to ocean  warming and increase in stratification" change to "At the same time the increase in atmospheric CO2  modifies the physical climate system,  leading to changes such as ocean warming and increased stratification…. "

We removed this text, as suggested by the reviewer in comment 19 above.

Line 84: change "ocean carbon uptake" "change in ocean carbon uptake"

We rephrased the text: '*the ocean carbon gain due to anthropogenic carbon emissions* …' (lines 86-87).

line 110: you don't need the subscript ocean for Equation 5. I would save the subscript position for discriminating between feedback parameters calculated using air-sea CO2 fluxes vs carbon storage.

We prefer to keep the subscript ocean, at least for $\Delta I_{ocean}$, for clarity. We did, however, introduce notation to separate the calculations based on air-sea CO2 vs carbon storage (lines 136-151).

line 111: choose to either capitalise or not the word "earth" in the document.

Agreed, we used Earth consistently in the revised manuscript.

Line 115: "To gain insight for the driving mechanisms of the carbon cycle feedbacks and their uncertainty " "To gain insight ::into:: the driving mechanisms of the carbon cycle feedbacks and their uncertainty "

Changed (line 199)

line 125: replace "extend" by "extent".

Changed.

line 125: replace "contemporary CO2" with "contemporary CO2 concentration".

Changed.

Line 126: include symbol and units for carbon inventory (I.e. deltaI, PgC) for consistency with unit conversion listed below. (Odd to have unit conversion when no units have been listed as yet).

Done, line 214.

line 127: for consistency, include units of DIC pools.

Done, line 214

line 128: Again you don't need the subscript ocean here. Save it for inventory calculated using carbon storage.

We prefer to keep the subscript ocean for ΔIocean, for clarity. We did, however, introduce notation to separate the calculations based on air-sea CO2 vs carbon storage (lines 136-151).

line 132: Again no ocean subscript needed, use symbol to symbol to specify we are talking about betas calculated from carbon storage.

Agreed, we used beta* and gamma* to clarify that we are talking about feedbacks calculated from carbon storage.

line 135: Rather than "can be expressed as" it would be more direct to write "were diagnosed" to clarify that this is what you do in this study.

Changed (line 224).

line 155: "contemporary atmospheric CO2". Maybe I have misunderstood something here. But, I would have thought this refers to projected atmospheric CO2? Could you please clarify.

Agreed, we updated the text to *'… is the ocean buffer factor for the increasing atmospheric CO2, but with no climate change …'* (lines 244-245).

**Regional carbon cycle feedbacks in CMIP6 Earth system models**

line 263: replace "into contribution" with "into contributions"

Changed (line 127).

line 266: replace "non-linearity of ocean carbon cycle feedbacks" with "non-linearity of ocean carbon cycle feedbacks (see Equation 4)"

Agreed. We re-organised this text and modified equation 6 (lines 129-139).

line 266: replace "n notes" by "n denotes"

Changed.

line 305: or that the region dominating the the carbon cycle feedback differs between the models.

Agreed and this is included in the '*different basins compensate each other*'.

**Control of the Atlantic Overturning circulation to the carbon cycle feedbacks**

The title seems back-to-front. Suggestions: "Relationship between carbon cycle feedbacks and the Atlantic Meridional overturning circulation" or "Control of the carbon cycle feedbacks by the Atlantic overturning circulation"

We changed the title for this section to 'Dependence of the carbon cycle feedbacks on the Atlantic Meridional Overturning Circulation'

**Discussion and Summary**

line 494: Similarly "control of the Atlantic Meridional Overturning circulation" is not the right title. You are talking here about the control of the feedbacks by the AMOC, not the controls on the AMOC itself.

We changed the title for this subsection to 'Effect of the Atlantic Meridional Overturning Circulation'.

---

## Author Response (AR2)

Dear Editor Prof. Joos,

I would like to submit the revised manuscript entitled '*Ocean carbon cycle feedbacks in CMIP6 models: contributions from different basins*' for consideration for publication in Biogeosciences.

In this revised version, we provided information on the timescale of the study in the summary, as suggested (line 593 of the revised manuscript), as well as in the abstract (line 4). To highlight the dependence of these feedback parameters on the timescale, we also moved the text *'Finally, for a quadrupling of atmospheric CO2 and on centennial time scale, as considered in this study, the carbon-concentration feedback is substantially larger than the carbon-climate feedback; however, this will not necessarily be the case after the emissions cease and the system adjusts towards equilibrium'* at the end of this final summary (lines 602-605).

Thank you for your consideration and I look forward to hearing from you.

Yours sincerely,

Dr Anna Katavouta